# The LOTUS initiative for open knowledge management in natural products research

Adriano Rutz[1,2], Maria Sorokina[3], Jakub Galgonek[4], Daniel Mietchen[5,6,7], Egon Willighagen[8], Arnaud Gaudry[1,2], James G Graham[9,10], Ralf Stephan[11], Roderic Page[12], Jiří Vondrášek[4], Christoph Steinbeck[3], Guido F Pauli[9,10], Jean-Luc Wolfender[1,2], Jonathan Bisson[9,10]*, Pierre-Marie Allard[1,2,13]*

[1]School of Pharmaceutical Sciences, University of Geneva, Geneva, Switzerland; [2]Institute of Pharmaceutical Sciences of Western Switzerland, University of Geneva, Geneva, Switzerland; [3]Institute for Inorganic and Analytical Chemistry, Friedrich-Schiller-University Jena, Jena, Germany; [4]Institute of Organic Chemistry and Biochemistry of the CAS, Prague, Czech Republic; [5]Ronin Institute, Montclair, United States; [6]Leibniz Institute of Freshwater Ecology and Inland Fisheries, Berlin, Germany; [7]School of Data Science, University of Virginia, Charlottesville, United States; [8]Department of Bioinformatics-BiGCaT, Maastricht University, Maastricht, Netherlands; [9]Center for Natural Product Technologies and WHO Collaborating Centre for Traditional Medicine (WHO CC/TRM), Pharmacognosy Institute; College of Pharmacy, University of Illinois at Chicago, Chicago, United States; [10]Department of Pharmaceutical Sciences, College of Pharmacy, University of Illinois at Chicago, Chicago, United States; [11]Ontario Institute for Cancer Research (OICR), University Ave Suite, Toronto, Canada; [12]University of Glasgow, Glasgow, United Kingdom; [13]Department of Biology, University of Fribourg, Fribourg, Switzerland

*For correspondence:
bjo@uic.edu (JB);
pierre-marie.allard@unifr.ch (P-MA)

**Abstract** Contemporary bioinformatic and chemoinformatic capabilities hold promise to reshape knowledge management, analysis and interpretation of data in natural products research. Currently, reliance on a disparate set of non-standardized, insular, and specialized databases presents a series of challenges for data access, both within the discipline and for integration and interoperability between related fields. The fundamental elements of exchange are referenced structure-organism pairs that establish relationships between distinct molecular structures and the living organisms from which they were identified. Consolidating and sharing such information via an open platform has strong transformative potential for natural products research and beyond. This is the ultimate goal of the newly established LOTUS initiative, which has now completed the first steps toward the harmonization, curation, validation and open dissemination of 750,000+ referenced structure-organism pairs. LOTUS data is hosted on Wikidata and regularly mirrored on https://lotus.naturalproducts.net. Data sharing within the Wikidata framework broadens data access and interoperability, opening new possibilities for community curation and evolving publication models. Furthermore, embedding LOTUS data into the vast Wikidata knowledge graph will facilitate new biological and chemical insights. The LOTUS initiative represents an important advancement in the design and deployment of a comprehensive and collaborative natural products knowledge base.

## Editor's evaluation

Rutz et al. describe the LOTUS initiative, an open science database that contains over 750,000 referenced structure-organism pairs. Present both the data that they have made available in Wikidata, as well as an interactive web portal, LOTUS provides a powerful platform for mining literature

for published data on structure-organism pairs. The strength of this initiative lies in the effort the authors had put in creating a database that is both reproducible and usable. The result is thus a complete and user-friendly product that will respond to people's needs.

## Introduction

### Evolution of electronic natural products resources

Natural Products (NP) research is a transdisciplinary field with wide-ranging interests: from fundamental structural aspects of naturally occurring molecular entities to their effects on living organisms and extending to the study of chemically mediated interactions within entire ecosystems. Defining the 'natural' qualifier is a complex task (*Ducarme and Couvet, 2020*; *All natural, 2007*). We thus adopt here a broad definition of a NP as any chemical entity *found in* a living organism, hereafter refered to as a *structure-organism pair*. An additional and fundamental element of a structure-organism pair is a reference to the experimental evidence that establishes the linkages between the chemical structure and the biological organism. A future-oriented electronic NP resource should contain fully-referenced structure-organism pairs.

Reliance on data from the NP literature presents many challenges. The assembly and integration of NP occurrences into an inter-operative platform relies primarily on access to a heterogeneous set of databases (DB) whose content and maintenance status are critical factors in this dependency (*Tsugawa, 2018*). A tertiary inter-operative NP platform is thus dependent on a secondary set of data that has been selectively annotated into a DB from primary literature sources. The experimental data itself reflects a complex process involving collection or sourcing of natural material (and establishment of its identity), a series of material transformation and separation steps and ultimately the chemical or spectral elucidation of isolates. The specter of human error and the potential for the introduction of biases are present at every phase of this journey. These include publication biases (*Lee et al., 2013*), such as emphasis on novel and/or bioactive structures in the review process, or, in DB assembly stages, with selective focus on a specific compound class or a given taxonomic range, or disregard for annotation of other relevant evidence that may have been presented in primary sources. Temporal biases also exist: a technological 'state-of-the-art' when published can eventually be recast as anachronistic.

The advancement of NP research has always relied on the development of new technologies. In the past century alone, the rate at which unambiguous identification of new NP entities from biological matrices can be achieved has been reduced from years to days and in the past few decades, the scale at which new NP discoveries are being reported has increased exponentially. Without a means to access and process these disparate NP data points, information is fragmented and scientific progress is impaired (*Balietti et al., 2015*). To this extent, contemporary bioinformatic tools enable the (re-)interpretation and (re-)annotation of (existing) datasets documenting molecular aspects of biodiversity (*Mongia and Mohimani, 2021*; *Jarmusch et al., 2020*).

While large, well-structured and freely accessible DB exist, they are often concerned primarily with chemical structures (e.g. PubChem (*Kim et al., 2019*), with over 100 M entries) or biological organisms (e.g. GBIF (*GBIF, 2020*), with over 1900 M entries), but scarce interlinkages limit their application for documentation of NP occurrence(s). Currently, no open, cross-kingdom, comprehensive and computer-interpretable electronic NP resource links NP and their containing organisms, along with referral to the underlying experimental work. This shortcoming breaks the crucial evidentiary link required for tracing information back to the original data and assessing its quality. Even valuable commercially available efforts for compiling NP data, such as the Dictionary of Natural Products (DNP), can lack proper documentation of these critical links.

Pioneering efforts to address such challenges led to the establishment of KNApSAck (*Shinbo et al., 2006*), which is likely the first public, curated electronic NP resource of referenced structure-organism pairs. KNApSAck (*Afendi et al., 2012*) currently contains 50,000+ structures and 100,000+ structure-organism pairs. However, the organism field is not standardized and access to the data is not straightforward. Another early-established electronic NP resources is the NAPRA-LERT dataset (*Graham and Farnsworth, 2010*), which was compiled over five decades from the NP literature, gathering and annotating data derived from over 200,000 primary literature sources. This dataset contains 200,000+ distinct compound names and structural elements, along with 500,000+ records of distinct, fully-cited structure-organism pairs. In total, NAPRALERT contains over 900,000

such records, due to equivalent structure-organism pairs reported in different citations. However, NAPRALERT is not an open platform and employs an access model that provides only limited free searches of the dataset. Finally, the NPAtlas (*van Santen et al., 2019*; *van Santen et al., 2022*) is a more recent project complying with the FAIR (Findability, Accessibility, Interoperability, and Reuse) guidelines for digital assets (*Wilkinson et al., 2016*) and offering convenient web access. While the NPAtlas allows retrieval and encourages submission of compounds with their biological source, it focuses on microbial NP and ignores a wide range of biosynthetically active organisms found in the Plantae kingdom.

The LOTUS initiative seeks to address the aforementioned shortcomings. Building on the experience gained through the establishment of the recently published COlleCtion of Open NatUral producTs (COCONUT) (*Sorokina et al., 2021a*) regarding the aggregation and curation of NP structural databases, this *savoir-faire* was expanded to accommodate biological organisms and scientific references in the equation. After extensive data curation and harmonization of over 40 electronic resources, pairs characterizing a NP occurrence were standardized at the chemical, biological and reference levels. At its current stage of development, LOTUS disseminates 750,000+ referenced structure-organism pairs. These efforts and experiences represent an intensive preliminary curatorial phase and the first major step towards providing a high-quality, computer-interpretable knowledge base capable of transforming NP research data management from a classical (siloed) database approach to an optimally shared resource.

## Accommodating principles of FAIRness and TRUSTworthiness for natural products knowledge management

In awareness of the multi-faceted pitfalls associated with implementing, using and maintaining classical scientific DBs (*Helmy et al., 2016*), and to enhance current and future sharing options, the LOTUS initiative selected the Wikidata platform for disseminating its resources. The idea of using wikis to disseminate databases is not new, with multiple underlying advantages (*Finn et al., 2012*). Since its creation, Wikidata has focused on cross-disciplinary and multilingual support. Wikidata is curated and governed collaboratively by a global community of volunteers, about 20,000 of which are contributing monthly. Wikidata currently contains more than 1 billion statements in the form of subject-predicate-object triples. Triples are machine-interpretable and can be enriched with qualifiers and references. Within Wikidata, data triples correspond to approximately 100 million entries, which can be grouped into classes as diverse as countries, songs, disasters, or chemical compounds. The statements are closely integrated with Wikipedia and serve as the source for many of its infoboxes. Various workflows have been established for reporting such classes, particularly those of interest to life sciences, such as genes, proteins, diseases, drugs, or biological taxa (*Waagmeester et al., 2020*).

Building on the principles and experiences described above, the present report introduces the development and implementation of the LOTUS workflow for NP occurrence curation and dissemination, which applies both FAIR and TRUST (Transparency, Responsibility, User focus, Sustainability and Technology) principles (*Lin et al., 2020*). LOTUS data upload and retrieval procedures ensure optimal accessibility by the research community, allowing any researcher to contribute, edit and reuse the data with a clear and open CC0 license (Creative Commons 0).

Despite many advantages, Wikidata hosting has some notable, yet manageable drawbacks. While its SPARQL query language offers a powerful way to query available data, it can also appear intimidating to the less experienced user. Furthermore, some typical queries of molecular electronic NP resources such as structural or spectral searches are not yet available in Wikidata. To address these shortcomings, LOTUS is hosted in parallel at https://lotus.naturalproducts.net (LNPN) within the naturalproducts.net ecosystem. The Natural Products Online website is a portal for open-source and open-data resources for NP research. In addition to the generalistic COCONUT and LNPN databases, the portal will enable hosting of arbitrary and skinned collections, themed in particular by species or taxonomic clade, by geographic location or by institution, together with a range of cheminformatics tools for NP research. LNPN is periodically updated with the latest LOTUS data. This dual hosting provides an integrated, community-curated and vast knowledge base (via Wikidata), as well as a NP community-oriented product with tailored search modes (*via* LNPN). The multiple data interaction options should establish the basis for the transparent and sustainable access, sharing and creation of knowledge on NP occurrence.

The LOTUS initiative was initially launched to answer our need to access the most comprehensive compilation of biological occurrences of NP. Indeed, we recently highlighted the interest of considering the taxonomic dimension when annotating metabolites (*Rutz et al., 2019*). This being said, many other concrete applications can result from an access by the scientific community to the LOTUS initiative data. For example, such a resource will facilitate the exploration of eco-evolutionary mechanisms at the molecular level (*Defossez et al., 2021*). In terms of drug discovery, this resource is extremely valuable to orient and guide researchers interested in a structure of interest. On the same theme, LOTUS is expected to be the perfect place to encounter 'molecular arguments' for biodiversity conservation (*Campbell, 2003*). Researchers interested in the history of science will be able, through this kind of resource, to gain a preliminary view of the temporal evolution of disciplines such as pharmacognosy. More generally, the objective of the LOTUS initiative is to prepare the ground for an electronic and globally accessible resource that would be the counterpart, at the metabolite level, of established databases linking proteins to biological organisms (e.g. Uniprot) and genes to biological organisms (Genbank). Once such an objective is reached, it will be possible to interconnect the three central objects of the living, that is metabolites, proteins and genes, through the common entity of these resources, the biological organism. Such an interconnection, fostering cross-fertilization of the fields of chemistry, biology and associated disciplines is desirable and necessary to advance towards a better understanding of Life.

## Results and discussion

This section is structured as follows: first, we present an overview of the LOTUS initiative at its current stage of development. The central curation and dissemination elements of the LOTUS initiative are then explained in detail. The third section addresses the interaction modes between end-users and LOTUS, including *data retrieval*, *addition,* and *editing*. Some examples on how LOTUS data can be used to answer research questions or develop hypothesis are given. The final section is dedicated to the interpretation of LOTUS data and illustrates the dimensions and qualities of the current LOTUS dataset from chemical and biological perspectives.

### Blueprint of the LOTUS initiative

Building on the standards established by the related WikiProjects on Wikidata (Chemistry, Taxonomy and Source Metadata), a NP chemistry-oriented subproject was created (Chemistry/Natural products). Its central data consists of three minimal sufficient objects:

- A *chemical structure object*, with associated Simplified Molecular Input Line Entry System (SMILES) (*Weininger, 1988*), International Chemical Identifier (InChI) (*Heller et al., 2013*) and InChIKey (a hashed version of the InChI).
- A *biological organism object*, with associated taxon name, the taxonomic DB where it was described and the taxon ID in the respective DB.
- A *reference object* describing the structure-organism pair, with the associated article title and a Digital Object Identifier (DOI), a PubMed (PMID), or PubMed Central (PMCID) ID.

As data formats are largely inhomogeneous among existing electronic NP resources, fields related to chemical structure, biological organism and references are variable and essentially not standardized. Therefore, LOTUS implements multiple stages of harmonization, processing, and validation (*Figure 1*, stages 1–3). LOTUS employs a Single Source of Truth (SSOT, Single_source_of_truth) to ensure data reliability and continuous availability of the latest curated version of LOTUS data in both Wikidata and LNPN (*Figure 1*, stage 4). The SSOT approach consists of a PostgreSQL DB that structures links and data schemes such that every data element has a single place. The LOTUS processing pipeline is tailored to efficiently include and diffuse novel or curated data directly from new sources or at the Wikidata level. This iterative workflow relies both on data addition and retrieval actions as described in the Data Interaction section. The overall process leading to referenced and curated structure-organisms pairs is illustrated in *Figure 1* and detailed hereafter.

By design, this iterative process fosters community participation, essential to efficiently document NP occurrences. All stages of the workflow are described on the git sites of the LOTUS initiative at https://github.com/lotusnprod and in the methods. At the time of writing, 750,000+ LOTUS entries contained a curated chemical structure, biological organism and reference and were available on both

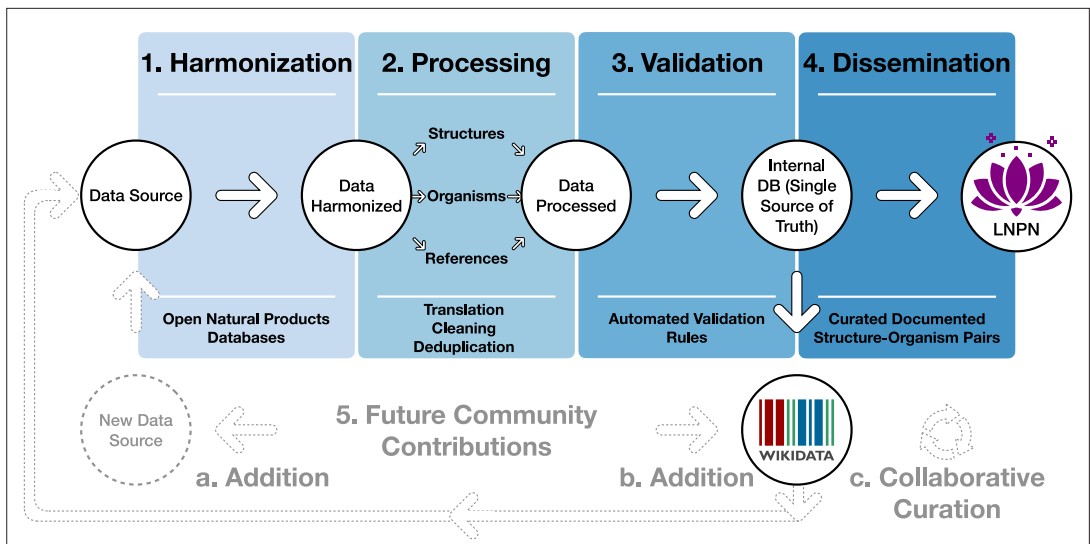

**Figure 1.** Blueprint of the LOTUS initiative. Data undergo a four-stage process: (1) Harmonization, (2) Processing, (3) Validation, and (4) Dissemination. The process was designed to incorporate future contributions (5), either by the addition of new data from within Wikidata (**a**) or new sources (**b**) or via curation of existing data (**c**). The figure is available under the CC0 license at https://commons.wikimedia.org/wiki/File:Lotus_initiative_1_blueprint.svg.

Wikidata and LNPN. As the LOTUS data volume is expected to increase over time, a frozen (as of 2021-12-20), tabular version of this dataset with its associated metadata is made available at https://doi.org/10.5281/zenodo.5794106 (*Rutz et al., 2021a*).

## Data harmonization

Multiple data sources were processed as described hereafter. All publicly accessible electronic NP resources included in COCONUT that contain referenced structure-organism pairs were considered as initial input. The data were complemented with COCONUT's own referenced structure-organism pairs (*Sorokina and Steinbeck, 2020a*), as well as the following additional electronic NP resources: Dr. Duke (*U.S. Department of Agriculture, 1992*), Cyanometdb (*Jones et al., 2021*), Datawarrior (*Sander et al., 2015*), a subset of NAPRALERT, Wakankensaku (*Wakankenaku, 2020*) and DiaNat-DB (*Madariaga-Mazón et al., 2021*).

The contacts of the electronic NP resources not explicitly licensed as open were individually reached for permission to access and reuse data. A detailed list of data sources and related information is available as Appendix 1. All necessary scripts for data gathering and harmonization can be found in the lotus-processor repository in the src/1_gathering directory and processed is detailed in the corresponding methods section gathering section. All subsequent iterations including new data sources, either updated information from the same data sources or new data, will involve a comparison with the previously gathered data at the SSOT level to ensure that the data is only curated once.

## Data processing and validation

As shown in *Figure 1*, data curation consisted of three stages: harmonization, processing, and validation. Thereby, after the harmonization stage, each of the three central objects – chemical compounds, biological organisms, and reference – were processed, as described in related methods section. Given the data size (2.5M+ initial entries), manual validation was unfeasible. Curating the references was a particularly challenging part of the process. Whereas organisms are typically reported by at least their vernacular or scientific denomination and chemical structures via their SMILES, InChI, InChIKey or image (not covered in this work), references suffer from largely insufficient reporting standards. Despite poor standardization of the initial reference field, proper referencing remains an indispensable way to establish the validity of structure-organism pairs. Better reporting practices, supported by tools such as Scholia (*Blomqvist et al., 2017*; *Rasberry et al., 2019*) and relying on Wikidata, Fatcat, or Semantic Scholar should improve reference-related information retrieval in the future.

**Table 1.** Example of a referenced structure-organism pair before and after curation.

|  | Structure | Organism | Reference |
|---|---|---|---|
| Before curation | Cyathocaline | Stem bark of Cyathocalyx zeylanica CHAMP. ex HOOK. f. & THOMS. (Annonaceae) | Wijeratne E. M. K., de Silva L. B., Kikuchi T., Tezuka Y., Gunatilaka A. A. L., Kingston D. G. I., J. Nat. Prod., 58, 459–462 (1995). |
| After curation | VFIIVOHWCNHINZ-UHFFFAOYSA-N | Cyathocalyx zeylanicus | 10.1021 /NP50117A020 |

In addition to curating the entries during data processing, 420 referenced structure-organism pairs were selected for manual validation. An entry was considered as valid if: (*i*) the structure (in the form of any structural descriptor that could be linked to the final sanitized InChIKey) was described in the reference (*ii*) the containing organism (as any organism descriptor that could be linked to the accepted canonical name) was described in the reference and (*iii*) the reference was describing the occurrence of the chemical structure in the biological organism. More details are available in the related methods section. This process allowed us to establish rules for automatic filtering and validation of the entries. The parameters of the automatic filtering are available as a function (filter_dirty.R) and are further described in the related methods section. The automatic filtering was then applied to all entries. To confirm the efficacy of the filtering process, a new subset of 100 diverse, automatically curated and automatically validated entries was manually checked, yielding a rate of 97% of true positives. The detailed results of the two manual validation steps are reported in Appendix 2. The resulting data is also available in the dataset shared at https://doi.org/10.5281/zenodo.5794106 (*Rutz et al., 2021a*). *Table 1* shows an example of a referenced structure-organism pair before and after curation. This process resolved the structure to an InChIKey, the organism to a valid taxonomic name and

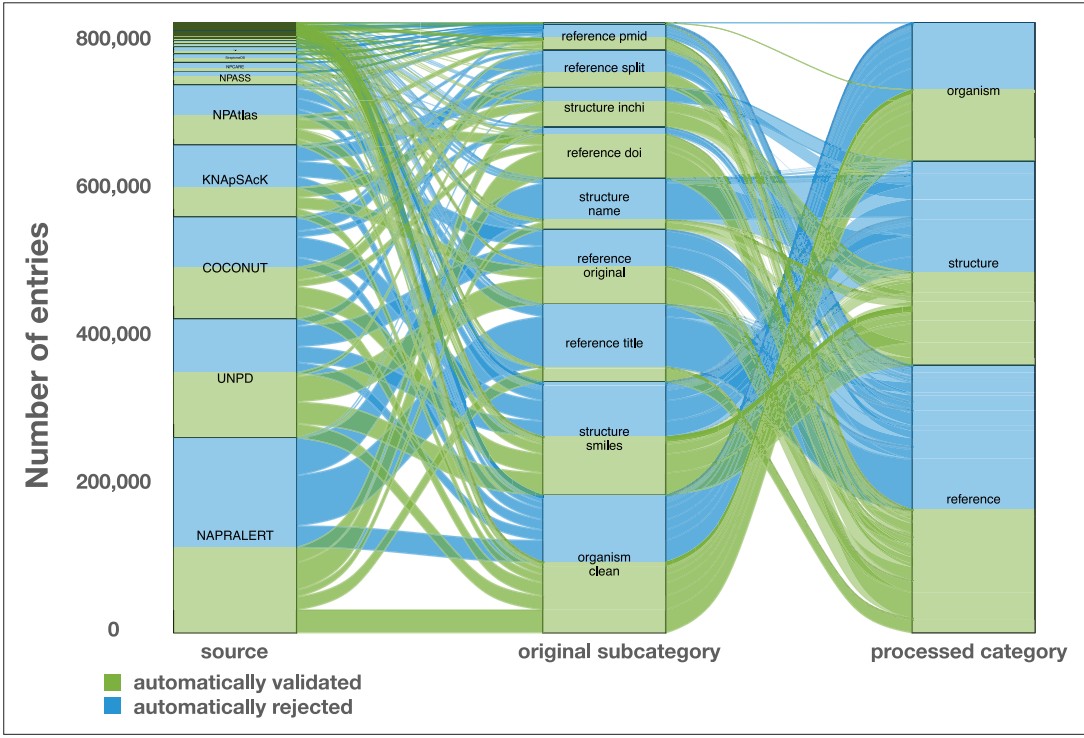

**Figure 2.** Alluvial plot of the data transformation flow within LOTUS during the automated curation and validation processes. The figure also reflects the relative proportions of the data stream in terms of the contributions from the various sources ('source' block, left), the composition of the harmonized subcategories ('original subcategory' block, middle) and the validated data after curation ('processed category' block, right). Automatically validated entries are represented in green, rejected entries in blue. The figure is available under the CC0 license at https://commons.wikimedia.org/wiki/File:Lotus_initiative_1_alluvial_plot.svg.

the reference to a DOI, thereby completing the establishment of the essential referenced structure-organism pair.

Challenging examples encountered during the development of the curation process were compiled in an edge cases table (tests/tests.tsv) to allow for automated unit testing. These tests allow a continuous revalidation of any change made to the code, ensuring that corrected errors will not reappear. The alluvial plot in *Figure 2* illustrates the individual contribution of each *source* and *original subcategory* that led to the *processed categories*: structure, organism, and reference.

The figure highlights, for example, the essential contribution of the reference DOI category to the final validated entries. A similar pattern can be seen concerning structures, where the validation rate of structural identifiers is higher than chemical names. The combination of the results of the automated curation pipeline and the manually curated entries led to the establishment of four categories (manually validated, manually rejected, automatically validated and automatically rejected) of the referenced structure-organism pairs that formed the processed part of the SSOT. Out of a total of 2.5M+ initial pairs, the manual and automatic validation retained 750,000+ pairs (approximately 30%), which were then selected for dissemination on Wikidata. Among validated entries, multiple ones were redundant among the source databases, thus also explaining the decrease of entries between the initial pairs and validated ones. Moreover, because data quality was favored over quantity, the number of rejected entries is high. Among them, multiple correct entries were certainly falsely rejected, but still not disseminated. All rejected entries were kept aside for later manual inspection and validation. These are publicly available at https://doi.org/10.5281/zenodo.5794597 (*Rutz et al., 2021b*). In the end, the disseminated data contained 290,000+ unique chemical structures, 40,000+ distinct organisms and 75,000+ references.

## Data dissemination

Research worldwide can benefit the most when all results of published scientific studies are fully accessible immediately upon publication (*Agosti and Johnson, 2002*). This concept is considered the foundation of scientific investigation and a prerequisite for the effective direction of new research efforts based on prior information. To achieve this, research results have to be made publicly available and reusable. As computers are now the main investigation tool for a growing number of scientists, all research data including those in publications should be disseminated in computer-readable format, following the FAIR principles. LOTUS uses Wikidata as a repository for referenced structure-organism pairs, as this allows documented research data to be integrated with a large, pre-existing and extensible body of chemical and biological knowledge. The dynamic nature of Wikidata encourages the continuous curation of deposited data through the user community. Independence from individual and institutional funding represents another major advantage of Wikidata. The Wikidata knowledge base and the option to use elaborate SPARQL queries allow the exploration of the dataset from a sheer unlimited number of angles. The openness of Wikidata also offers unprecedented opportunities for community curation, which will support, if not guarantee, a dynamic and evolving data repository. At the same time, certain limitations of this approach can be anticipated. Despite (or possibly due to) their power, SPARQL queries can be complex and potentially require an in-depth understanding of the models and data structure. This involves a steep learning curve which can discourage some end-users. Furthermore, traditional ways to query electronic NP resources such as structural or spectral searches are currently not within the scope of Wikidata and, are thus addressed in LNPN. Using the pre-existing COCONUT template, LNPN hosting allows the user to perform structural searches by directly drawing a molecule, thereby addressing the current lack of such structural search possibilities in Wikidata. Since metabolite profiling by Liquid Chromatography (LC) - Mass Spectrometry (MS) is now routinely used for the chemical composition assessment of natural extracts, future versions of LOTUS and COCONUT are envisioned to be augmented by predicted MS spectra and hosted at https://naturalproducts.net to allow mass and spectral-based queries. Note that such spectral database is already available at https://doi.org/10.5281/zenodo.5607264 (*Allard et al., 2021*). To facilitate queries focused on specific taxa (e.g. 'return all molecules found in the Asteraceae family'), a unified taxonomy is paramount. As the taxonomy of living organisms is a complex and constantly evolving field, all the taxon identifiers from all accepted taxonomic DB for a given taxon name were kept. Initiatives such as the Open Tree of Life (OTL) (*Rees and Cranston, 2017*) will help to gradually reduce these discrepancies, and the Wikidata platform can and does support such developments.

OTL also benefits from regular expert curation and new data. As the taxonomic identifier property for this resource did not exist in Wikidata, its creation was requested and obtained. The property is now available as 'Open Tree of Life ID' (P9157).

Following the previously described curation process, all validated entries have been made available through Wikidata and LNPN. LNPN will be regularly mirroring Wikidata LOTUS through the SSOT as described in *Figure 1*.

## User interaction with LOTUS data

The possibilities to interact with the LOTUS data are numerous. The following gives examples of how to retrieve, add and edit LOTUS data.

### Data retrieval

LOTUS data can be queried and retrieved either directly in Wikidata or on LNPN, both of which have distinct advantages. While Wikidata offers flexible and powerful queries capacities at the cost of potential complexity, LNPN has a graphical user interface with capabilities of drawing chemical structures, simplified structural or biological filtering and advanced chemical descriptors, albeit with a more rigid structure. For bulk download, a frozen version of LOTUS data (timestamp of 2021-12-20) is also available at https://doi.org/10.5281/zenodo.5794106 (*Rutz et al., 2021a*). More refined approaches to the direct interrogation of the up-to-date LOTUS data both in Wikidata and LNPN are detailed hereafter.

### Wikidata

The easiest way to search for NP occurrence information in Wikidata is by typing the name of a chemical structure directly into the 'Search Wikidata' field, which (for left-to-right languages) can be found in the upper right corner of the Wikidata homepage or any other Wikidata page. For example, by typing 'erysodine', the user will land on the page of this compound (Q27265641). Scrolling down to the 'found in taxon' statement will allow the user to view the biological organisms reported to contain this NP (*Figure 3*). Clicking the reference link under each taxon name links to the publication(s) documenting the occurrence.

The typical approach to more elaborated querying involves writing SPARQL queries using the Wikidata Query Service or another direct connection to a SPARQL endpoint. *Table 2* contains some

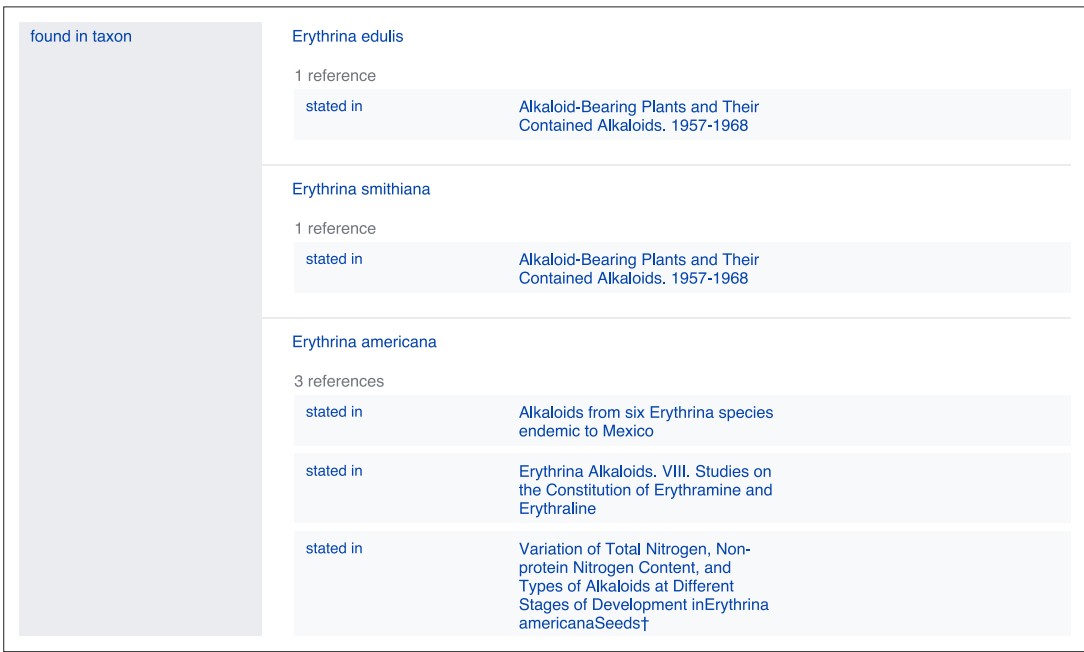

**Figure 3.** Illustration of the 'found in taxon' statement section on the Wikidata page of erysodine Q27265641 showing a selection of erysodine-containing taxa and the references documenting these occurrences.

**Table 2.** Potential questions about structure-organism relationships and corresponding Wikidata queries.

| Question | Wikidata SPARQL query |
| --- | --- |
| What are the compounds present in Mouse-ear cress (*Arabidopsis thaliana*) or its child taxa? | https://w.wiki/4Vcv |
| Which organisms are known to contain *β*-sitosterol? | https://w.wiki/4VFn |
| Which organisms are known to contain stereoisomers of *β*-sitosterol? | https://w.wiki/4VFq |
| Which pigments are found in which taxa, according to which reference? | https://w.wiki/4VFx |
| What are examples of organisms where compounds were found in an organism sharing the same parent taxon, but not in the organism itself? | https://w.wiki/4Wt3 |
| Which *Zephyranthes* species lack compounds known from at least two species in the genus? | https://w.wiki/4VG3 |
| How many compounds are structurally similar to compounds labeled as antibiotics? (grouped by the parent taxon of the containing organism) | https://w.wiki/4VG4 |
| Which organisms contain indolic scaffolds? Count occurrences, group and order the results by the parent taxon. | https://w.wiki/4VG9 |
| Which compounds with known bioactivities were isolated from Actinobacteria, between 2014 and 2019, with related organisms and references? | https://w.wiki/4VGC |
| Which compounds labeled as terpenoids were found in *Aspergillus* species, between 2010 and 2020, with related references? | https://w.wiki/4VGD |
| Which are the available referenced structure-organism pairs on Wikidata? (example limited to 1000 results) | https://w.wiki/4VFh |

examples from simple to more elaborated queries, demonstrating what can be done using this approach. The full-text queries with explanations are included in *Supplementary file 1*.

The queries presented in *Table 2* are only selected examples, and many other ways of interrogating LOTUS can be formulated. Generic queries can be used, for example, for hypothesis generation when starting a research project. For instance, a generic SPARQL query - listed in *Table 2* as "Which are the available referenced structure-organism pairs on Wikidata?" - retrieves all structures, identified by their InChIKey (P235), which contain 'found in taxon' (P703) statements that are stated in (P248) a bibliographic reference: https://w.wiki/4VFh. Data can then be exported in various formats, such as classical tabular formats, json, or html tables (see Download tab on the lower right of the query frame). At the time of writing (2021-12-20), this query (without the LIMIT 1000) returned 951,800 entries; a frozen query result is available at https://doi.org/10.5281/zenodo.5668854 (*Rutz et al., 2021d*).

Targeted queries allowing to interrogate LOTUS data from the perspective of one of the three objects forming the referenced structure-organism pairs can also be built. Users can, for example, retrieve a list of all structures reported from a given organism, such as all structures reported from *Arabidopsis thaliana* (Q158695) or its child taxa (https://w.wiki/4Vcv). Alternatively, all organisms containing a given chemical can be queried via its structure, such as in the search for all organisms where *β*-sitosterol (Q121802) was found in (https://w.wiki/4VFn). For programmatic access, the lotus-wikidata-exporter repository also allows data retrieval in RDF format and as TSV tables.

To further showcase the possibilities, two additional queries were established (https://w.wiki/4VGC and https://w.wiki/4VGD). Both queries were inspired by recent literature review works (*Jose et al., 2021*; *Zhao et al., 2022*). The first work describes compounds found in Actinobacteria, with a biological focus on compounds with reported bioactivity. The second one describes compounds found in *Aspergillus* spp., with a chemical focus on terpenoids. In both cases, in seconds, the queries allow retrieving a table similar to the ones in the mentioned literature reviews. While these queries are not a direct substitute for manual literature review, they do allow researchers to quickly begin such a review process with a very strong body of relevant references.

For a convenient expansion or limitation of the results, certain types of queries such as structure or similarity searches usually exist in molecular electronic resources. As these queries are not natively integrated by SPARQL, they are not readily available for Wikidata exploration. To address such limitation, Galgonek et al. developed an in-house SPARQL engine that allows utilization of Sachem, a high-performance chemical DB cartridge for PostgreSQL for fingerprint-guided substructure and similarity search (*Kratochvíl et al., 2018*). The engine is used by the Integrated Database of Small Molecules

(IDSM) that operates, among other things, several dedicated endpoints allowing structural search in selected small-molecule datasets via SPARQL (*Kratochvíl et al., 2019*). To allow substructure and similarity searches via SPARQL also on compounds from Wikidata, a dedicated IDSM/Sachem endpoint was created for the LOTUS project. The endpoint indexes isomeric (P2017) and canonical (P233) SMILES code available in Wikidata. To ensure that data is kept up-to-date, SMILES codes are automatically downloaded from Wikidata daily. The endpoint allows users to run federated queries and, thereby, proceed to structure-oriented searches on the LOTUS data hosted at Wikidata. For example, the SPARQL query https://w.wiki/4VG9 returns a list of all organisms that produce NP with an indolic scaffold. The output is aggregated at the parent taxa level of the containing organisms and ranked by the number of scaffold occurrences.

Regarding the versioning aspects, some challenges are implied by the dynamic nature of the Wikidata environment. However, tracking of the data evolution can be achieved in multiple ways and at different levels: at the full Wikidata level, dumps are regularly created (https://dumps.wikimedia.org/wikidatawiki/entities) while at the individual entry level the full history of modification can be consulted (see following link for the full edit history of erythromycin for example (https://www.wikidata.org/w/index.php?title=Q213511&action=history)).

We propose to the users a simple approach to document, version and share the output of queries on the LOTUS data at a defined time point. For this, in addition to sharing the short url of a the SPARQL query (which will return results evolving over time), a simple archiving of the returned table to Zenodo or similar platform can be done. In order to gather results of SPARQL queries, we established the LOTUS Initiative Community repository. The following link allows to directly contribute to the community repository https://zenodo.org/deposit/new?c=the-lotus-initiative. For example, the output of this Wikidata SPARQL query https://w.wiki/4N8G realized on the 2021-11-10T16:56 can be easily archived and shared in a publication via its DOI 10.5281/zenodo.5668380.

## Lotus.naturalproducts.net (LNPN)

In the search field of the LNPN interface (https://lotus.naturalproducts.net), simple queries can be achieved by typing the molecule name (e.g. ibogaine) or pasting a SMILES, InChI, InChIKey string, or a Wikidata identifier. All compounds reported from a given organism can be found by entering the organism name at the species or any higher taxa level (e.g. *Tabernanthe iboga*). Compound search by chemical class is also possible.

Alternatively, a structure can be directly drawn in the structure search interface (https://lotus.naturalproducts.net/search/structure), where the user can also decide on the nature of the structure search (exact, similarity, substructure search). Refined search mode combining multiple search criteria, in particular physicochemical properties, is available in the advanced search interface (https://lotus.naturalproducts.net/search/advanced).

Within LNPN, LOTUS bulk data can be retrieved as SDF or SMILES files, or as a complete MongoDB dump via https://lotus.naturalproducts.net/download. Extensive documentation describing the search possibilities and data entries is available at https://lotus.naturalproducts.net/documentation. LNPN can also be queried via the application programming interface (API) as described in the documentation.

## Data addition and evolution

One major advantage of the LOTUS architecture is that every user has the option to contribute to the NP occurrences documentation effort by adding new or editing existing data. As all LOTUS data applies the SSOT mechanism, reprocessing of previously treated elements is avoided. However, at the moment, the SSOT channels are not open to the public for direct write access to maintain data coherence and evolution of the SSOT scheme. For now, the users can employ the following approaches to add or modify data in LOTUS.

### Sources

LOTUS data management involves regular re-importing of both current and new data sources. New and edited information from these electronic NP resources will be checked against the SSOT. If absent or different, data will be passed through the curation pipeline and subsequently stored in the SSOT. Accordingly, by contributing to external electronic NP resources, any researcher has a means

of providing new data for LOTUS, keeping in mind the inevitable delay between data addition and subsequent inclusion into LOTUS.

## Wikidata

The currently favored approach to add new data to LOTUS is to create or edit Wikidata entries directly. Newly created or edited data will then be imported into the SSOT. There are several ways to interact with Wikidata which depend on the technical skills of the user and the volume of data to be uploaded/modified.

## Pre-requisites

While direct Wikidata upload is possible, contributors are encouraged to use the LOTUS curation pipeline as a preliminary step to strengthen the initial data quality. For this, a specific mode of the LOTUS processor can be called (see Custom mode). The added data will therefore benefit from the curation and validation stages implemented in the LOTUS processing pipeline.

## Manual upload

Any researcher interested in reporting NP occurrences can manually add the data directly in Wikidata, without any particular technical knowledge requirement. For this the creation of a Wikidata account and following the general object editing guidelines is advised. Regarding the addition of NP-centered objects (i.e. referenced structure-organisms pairs), users shall refer to the WikiProject Chemistry/Natural products group page.

A tutorial for the manual creation and upload of a referenced structure-organism pair to Wikidata is available in *Supplementary file 2*.

## Batch and automated upload

Through the initial curation process described previously, 750,000+ referenced structure-organism pairs were validated for Wikidata upload. To automate this process, a set of programs were written to automatically process the curated outputs, group references, organisms and compounds, check if they are already present in Wikidata (using SPARQL and direct Wikidata querying) and insert or update the entities as needed (i.e. upserting). These scripts can be used for future batch upload of properly curated and referenced structure-organism pairs to Wikidata. Programs for data addition to Wikidata can be found in the repository lotus-wikidata-interact. The following Xtools page offers an overview of the latest activity performed by our NPimporterBot, using those programs.

## Data editing

Even if correct at a given time point, scientific advances can invalidate or update previously uploaded data. Thus, the possibility to continuously edit the data is desirable and guarantees data quality and sustainability. Community-maintained knowledge bases such as Wikidata encourage such a process. Wikidata presents the advantage of allowing both manual and automated correction. Field-specific robots such as SuccuBot, KrBot, Pi_bot and ProteinBoxBot or our NPimporterBot went through an approval process. The robots are capable of performing thousands of edits without the need for human input. This automation helps reduce the amount of incorrect data that would otherwise require manual editing. However, manual curation by human experts remains irreplaceable as a standard. Users who value this approach and are interested in contributing are invited to follow the manual curation tutorial in *Supplementary file 2*.

The Scholia platform provides a visual interface to display the links among Wikidata objects such as researchers, topics, species or chemicals. It now provides an interesting way to view the chemical compounds found in a given biological organism (see here for the metabolome view of *Eurycoma longifolia*). If Scholia currently does not offer a direct editing interface for scientific references, it still allows users to proceed to convenient batch editing via QuickStatements. The adaptation of such a framework to edit the referenced structure-pairs in the LOTUS initiative could thus facilitate the capture of future expert curation, especially manual efforts that cannot be replaced by automated scripts.

## Data interpretation

To illustrate the nature and dimensions of the LOTUS dataset, some selected examples of data interpretation are shown. First, the distribution of chemical structures among four important NP reservoirs: plants, fungi, animals, and bacteria (Table 3). Then, the distribution of biological organisms according to the number of related chemical structures and likewise the distribution of chemical structures across biological organisms are illustrated (Figure 4). Furthermore, the individual electronic NP resources participation in LOTUS data is resumed using the UpSet plot depiction, which allows the visualization of intersections in data sets (Figure 5). Across these figures we take again the two previous examples, i.e, *β*-sitosterol as chemical structure and *Arabidopsis thaliana* as biological organism because of their well-documented statuses. Finally, a biologically interpreted chemical tree and a chemically-interpreted biological tree are presented (Figures 6 and 7). The examples illustrate the overall chemical and biological coverage of LOTUS by linking family-specific classes of chemical structures to their taxonomic position. Table 3, Figures 4, 6 and 7 were generated using the frozen data (2021-12-20 timestamp), which is available for download at https://doi.org/10.5281/zenodo.5794106 (*Rutz et al., 2021a*). Figure 5 required a dataset containing information from closed resources and the complete data used for its generation is therefore not available for public distribution. All scripts used for the generation of the figures are available in the lotus-processor repository in the src/4_visualizing directory for reproducibility.

## Distribution of chemical structures across reported biological organisms in LOTUS

*Table 3* summarizes the distribution of chemical structures and their chemical classes (according to NPClassifier *Kim et al., 2021*) across the biological organisms reported in LOTUS. For this, biological organisms were grouped into four artificial taxonomic levels (plants, fungi, animals, and bacteria). These were built by combining the two highest taxonomic levels in the OTL taxonomy, namely Domain and Kingdom levels. "Plants" corresponded to "Eukaryota_Archaeplastida", "Fungi" to "Eukaryota_Fungi", "Animals" to "Eukaryota_Metazoa" and "Bacteria" to "Bacteria_NA". The category corresponding to "Eukaryota_NA" mainly contained Algae, but also other organisms such as Amoebozoa and was therefore excluded. This represented less than 1% of all entries. The details of this process are available under src/3_analyzing/structure_taxon_distribution.R. When the chemical structure/class was reported only in one taxonomic grouping, it was counted as 'specific'.

## Distributions of organisms per structure and structures per organism

Readily achievable outcomes from LOTUS show that the depth of exploration of the world of NP is rather limited: as depicted in Figure 4, on average, three organisms are reported per chemical structure and eleven structures per organism. Notably, half of all structures have been reported from a single organism and half of all studied organisms are reported to contain five or fewer structures. Metabolomics studies suggest that these numbers are heavily underrated (*Noteborn et al., 2000*; *Wang et al., 2020*) and indicate that a better reporting of the metabolites detected in the course of NP chemistry investigations should greatly improve coverage.

This incomplete coverage may be partially explained by the habit in classical NP journals to accept only new and/or bioactive chemical structures for publication. Another possible explanation is the fact

**Table 3.** Distribution and specificity of chemical structures across four important NP reservoirs: plants, fungi, animals, and bacteria.
When the chemical structure/class appeared only in one group and not the three others, they were counted as 'specific'. Chemical classes were attributed with NPClassifier.

| Group | Organisms | 2D Structure-Organism pairs | 2D chemical structures | Specific 2D chemical structures | Chemical classes | Specific chemical classes |
|---|---|---|---|---|---|---|
| Plantae | 28,439 | 342,891 | 95,191 | 90,672 (95%) | 545 | 59 (11%) |
| Fungi | 4,003 | 36,950 | 22,594 | 20,194 (89%) | 417 | 19 (5%) |
| Animalia | 2,716 | 24,114 | 15,242 | 11,822 (78%) | 455 | 14 (3%) |
| Bacteria | 1,555 | 23,198 | 15,895 | 14,130 (89%) | 385 | 43 (11%) |

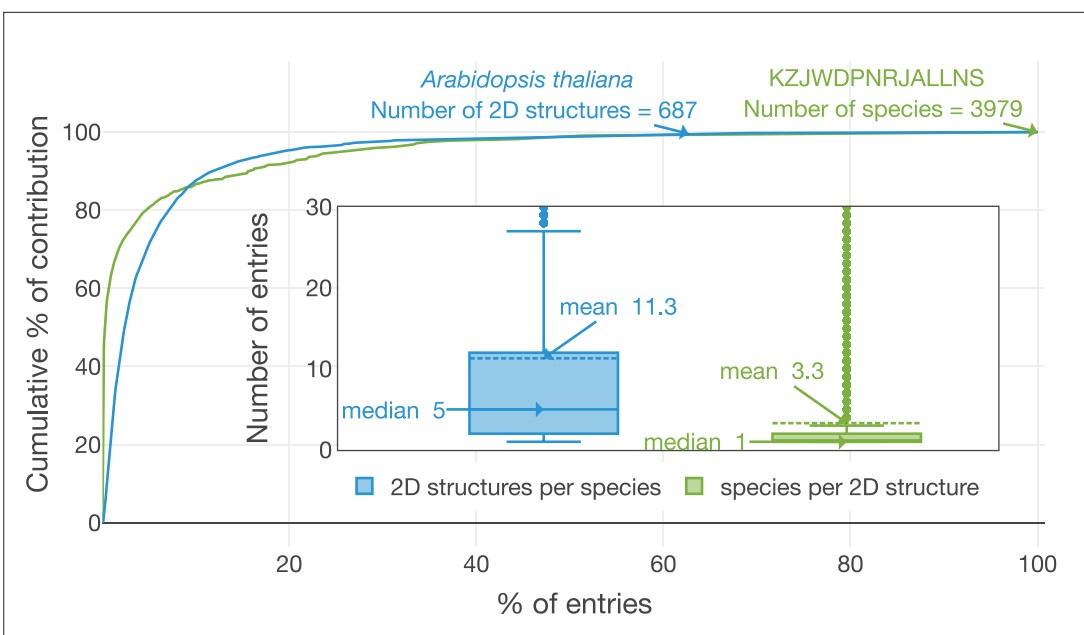

**Figure 4.** Distribution of 'structures per organism' and 'organisms per structure'. The number of organisms linked to the planar structure of β-sitosterol (KZJWDPNRJALLNS) and the number of chemical structures in *Arabidopsis thaliana* are two exemplary highlights. *A. thaliana* contains 687 different short InChIKeys (i.e. 2D structures) and KZJWDPNRJALLNS is reported in 3979 distinct organisms. Less than 10% of the species contain more than 80% of the structural diversity present within LOTUS. In parallel, 80% of the species present in LOTUS are linked to less than 10% of the structures. The figure is available under the CC0 license at https://commons.wikimedia.org/wiki/File:Lotus_initiative_1_structure_organism_distribution.svg.

that specific chemical classes have been under heavier scrutiny by the natural products community than others. For example, alkaloids have three specific characteristics which favor their reporting in the literature. First, they are often endowed with potent biological activities making them a target in the frame of pharmacognosy research. Second, their chemical nature makes them readily accessible from complex biological matrices through acido-basic extraction. Third, they ionize greatly in positive MS mode, which makes their detection even at a very low concentration possible, where other compounds present in much higher concentrations are not detected. It is thus a complex task to answer the following question: "Is the currently observed repartition of alkaloids across the tree of life a reflection of their true biological occurrence or is this repartition biased by the aforementioned characteristics of this chemical class?" While the LOTUS initiative does not allow yet disentangling the bias from the true occurrence, it should offer sound and strong foundations for such challenging research problematic.

Another obvious explanation for the limited coverage (see *Figure 4*) is the fact that most of the chemical structures in LOTUS have been physically isolated and described. This is an extremely time-consuming effort that can obviously not be carried on all metabolites of all biological organisms. Here, the sensitivity of mass spectrometry and the ever-increasing efficiency of computational metabolite annotation solutions could offer a strong take. The documentation of metabolite annotation results obtained on large collections of biological matrices and the associated metadata within knowledge graphs offers exciting perspectives regarding the possibilities to expand both the chemical and biological coverage of the LOTUS data in a feasible manner.

## Contribution of individual electronic NP resources to LOTUS

The added value of the LOTUS initiative to assemble multiple electronic NP resources is illustrated in *Figure 5*: Panel A shows the contributions of the individual electronic NP resources to the ensemble of chemical structures found in one of the most studied vascular plants, *Arabidopsis thaliana* ("Mouse-ear cress"; Q147096). Panel B shows the ensemble of taxa reported to contain the planar structure of the widely occurring triterpenoid *β*-sitosterol (Q121802).

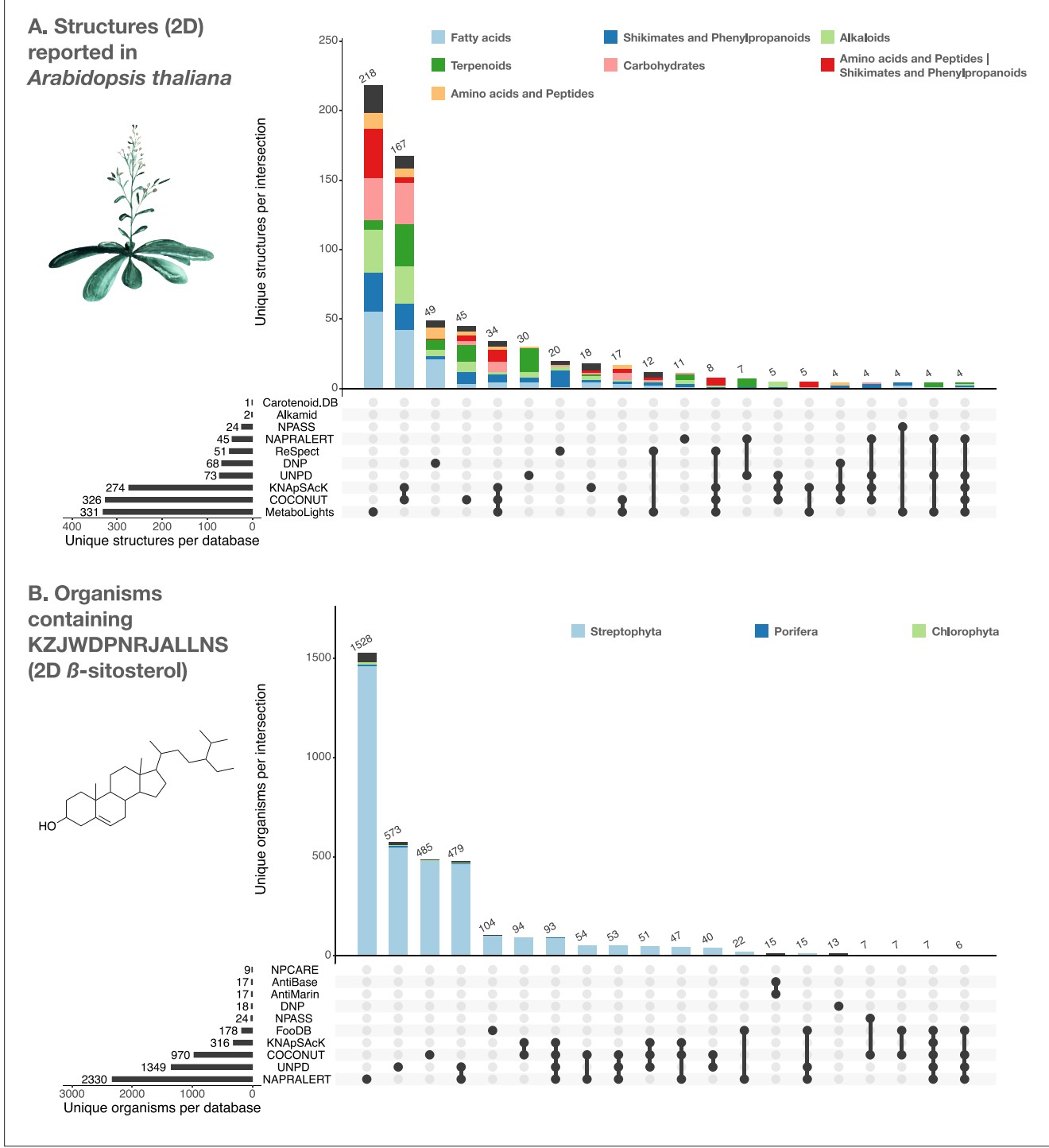

**Figure 5.** UpSet plots of the individual contribution of electronic NP resources to the planar structures found in *Arabidopsis thaliana* (**A**) and to organisms reported to contain the planar structure of β-sitosterol (KZJWDPNRJALLNS) (**B**). UpSet plots are evolved Venn diagrams, allowing to represent intersections between multiple sets. The horizontal bars on the lower left represent the number of corresponding entries per electronic NP resource. The dots and their connecting line represent the intersection between source and consolidate sets. The vertical bars indicate the number of entries at the intersection. For example, 479 organisms containing the planar structure of β-sitosterol are present in both UNPD and NAPRALERT, whereas each of them respectively reports 1349 and 2330 organisms containing the planar structure of β-sitosterol. The figure is available under the CC0 license at https://commons.wikimedia.org/wiki/File:Lotus_initiative_1_upset_plot.svg.

*Figure 5A* also shows that according to NPClassifier, the 'chemical pathway' category distribution across electronic NP resources is not conserved. Note that NPClassifier and ClassyFire (*Djoumbou Feunang et al., 2016*) chemical classification results are both available as metadata in the frozen LOTUS export and LNPN. Both classification tools return a chemical taxonomy for individual structures, thus allowing their grouping at higher hierarchical levels, in the same way as it is done for biological taxonomies. The UpSet plot in *Figure 5* indicates the poor overlap of preexisting electronic NP resources and the added value of an aggregated dataset. This is particularly well illustrated in *Figure 5B.*, where the number of organisms for which the planar structure of β-sitosterol (KZJWDPNR-JALLNS) has been reported is shown for each intersection. NAPRALERT has by far the highest number of entries (2330 in total), while other electronic NP resources complement this well: e.g. UNPD has 573 reported organisms with β-sitosterol that do not overlap with any other resource. Of note, β-sitosterol is documented in only 13 organisms in the DNP, highlighting the importance of a better systematic reporting of ubiquitous metabolites and the interest of multiple data sources agglomeration.

## A biologically interpreted chemical tree

The chemical diversity captured in LOTUS is here displayed using TMAP (*Figure 6*), a visualization library allowing the structural organization of large chemical datasets as a minimum spanning tree (*Probst and Reymond, 2020*). Using Faerun, an interactive HTML file is generated to display metadata and molecule structures by embedding the SmilesDrawer library (*Probst and Reymond, 2018a*; *Probst and Reymond, 2018b*). Planar structures were used for all compounds to generate the TMAP (chemical space tree-map) using MAP4 encoding (*Capecchi et al., 2020*). As the tree organizes structures according to their molecular fingerprint, an anticipated coherence between the clustering of compounds and the mapped NPClassifier chemical class is observed (*Figure 6A.*). For clarity, some of the most represented chemical classes of LOTUS plus quassinoids and stigmastane steroids are mapped, with examples of a quassinoid (NXZXPYYKGQCDRO) (light green star) and a stigmastane steroid (KZJWDPNRJALLNS) (dark green diamond) and their corresponding location in the TMAP.

To explore relationships between chemistry and biology, it is possible to map taxonomical information such as the most reported biological family per chemical compound (*Figure 6B.*) or the biological specificity of chemical classes (*Figure 6C.*) on the TMAP. The biological specificity score at a given taxonomic level for a given chemical class is calculated as a Jensen-Shannon divergence. A score of 1 suggests that compounds are highly specific, 0 that they are ubiquitous. For more details, see 3_analyzing/jensen_shannon_divergence.R. This visualization allows to highlight chemical classes specific to a given taxon, such as the quassinoids in the Simaroubaceae family. In this case, it is striking to see how well the compounds of a given chemical class (quassinoids) (*Figure 6A.*) and the most reported plant family per compound (Simaroubaceae) (*Figure 6B.*) overlap. This is also evidenced on *Figure 6C.* with a Jensen-Shannon divergence of 0.99 at the biological family level for quassinoids. In this plot, it is also possible to identify chemical classes that are widely spread among living organisms, such as the stigmastane steroids, which exhibit a Jensen-Shannon divergence of 0.73 at the biological family level. This means that repartition of stigmastane steroids among families is not specific. *Figure 7—figure supplement 1* further supports this statement.

## A chemically interpreted biological tree

An alternative view of the biological and chemical diversity covered by LOTUS is illustrated in *Figure 7*. Here, chemical compounds are not organized but biological organisms are placed in their taxonomy. To limit bias due to under-reporting in the literature and keep a reasonable display size, only families with at least 50 reported compounds were included. Organisms were classified according to the OTL taxonomy and structures according to NPClassifier. The tips were labeled according to the biological family and colored according to their biological kingdom. The bars represent structure specificity of the most characteristic chemical class of the given biological family (the higher the more specific). This specificity score was a Jaccard index between the chemical class and the biological family. For more details, see 4_visualizing/plot_magicTree.R.

*Figure 7* makes it possible to spot highly specific compound classes such as trinervitane terpenoids in the Termitidae, the rhizoxin macrolides in the Rhizopodaceae, or the quassinoids and limonoids typical, respectively, of Simaroubaceae and Meliaceae. Similarly, tendencies of more generic occurrence of NP can be observed. For example, within the fungal kingdom, Basidiomycotina appear to

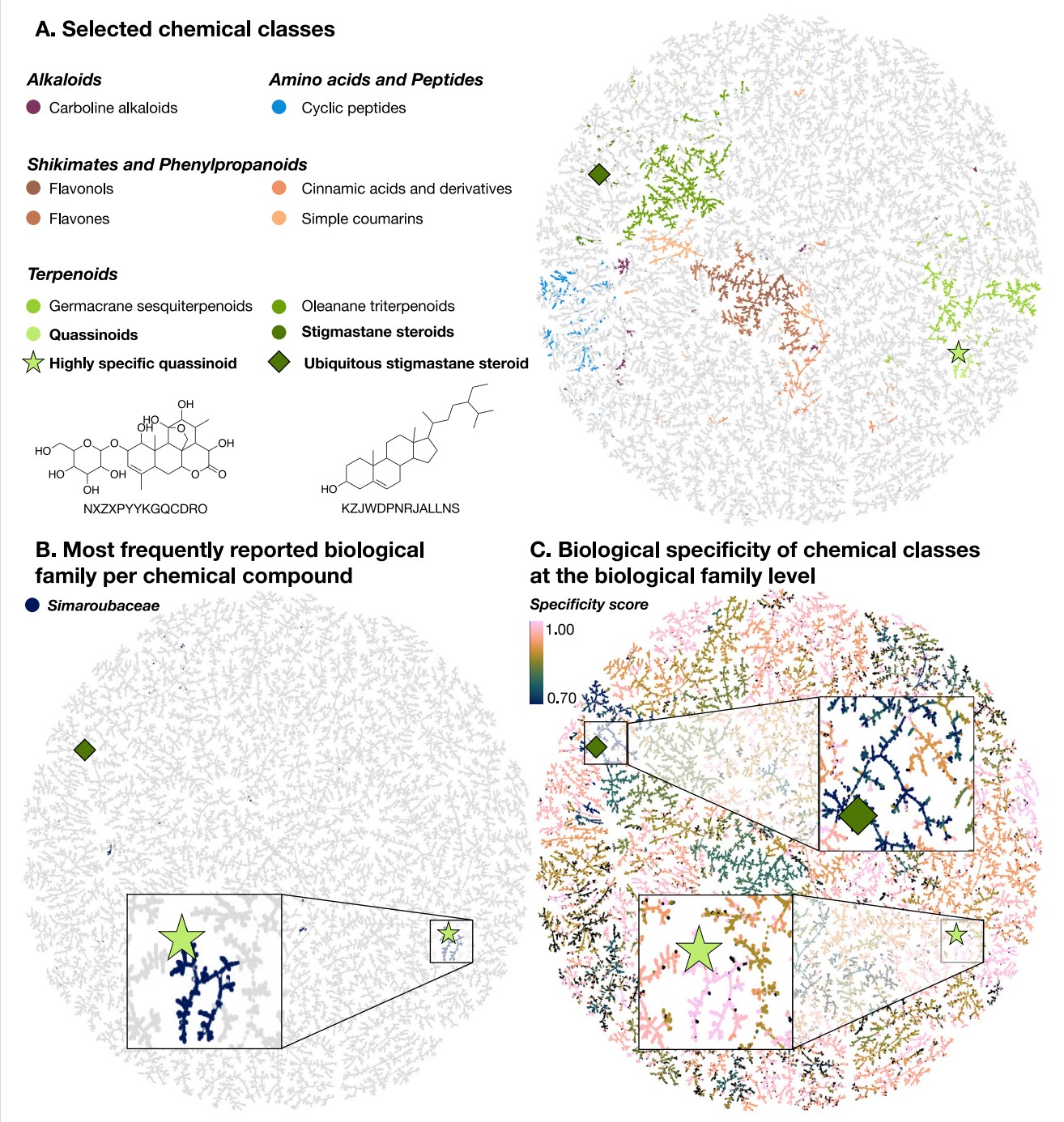

**Figure 6.** TMAP visualizations of the chemical diversity present in LOTUS. Each dot corresponds to a chemical structure. A highly specific quassinoid (NXZXPYYKGQCDRO) (light green star) and an ubiquitous stigmastane steroid (KZJWDPNRJALLNS) (dark green diamond) are mapped as examples in all visualizations. In panel A., compounds (dots) are colored according to the NPClassifier chemical class they belong to. In panel B., compounds that are mostly reported in the Simaroubaceae family are highlighted in blue. Finally, in panel C., the compounds are colored according to the specificity score of chemical classes found in biological organisms. This biological specificity score at a given taxonomic level for a given chemical class is calculated as a Jensen-Shannon divergence. A score of 1 suggests that compounds are highly specific, 0 that they are ubiquitous. Zooms on a group of compounds of high biological specificity score (in pink) and on compounds of low specificity (blue) are depicted. An interactive HTML visualization of the LOTUS TMAP is available at https://lotus.nprod.net/post/lotus-tmap/ and archived at https://doi.org/10.5281/zenodo.5801807 (*Rutz and Gaudry, 2021*). The figure is available under the CC0 license at https://commons.wikimedia.org/wiki/File:Lotus_initiative_1_biologically_interpreted_chemical_tree.svg.

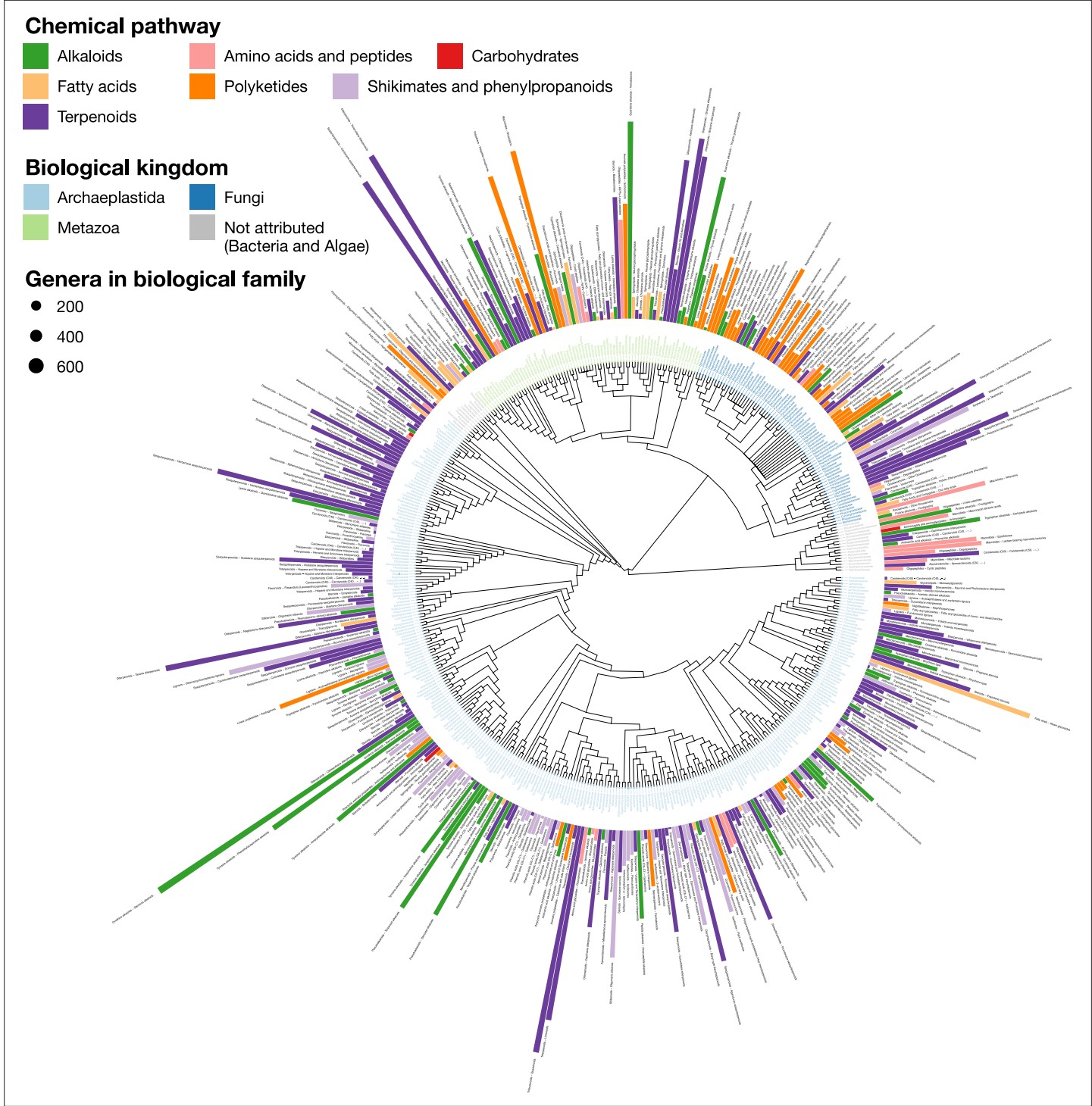

**Figure 7.** LOTUS provides new means of exploring and representing chemical and biological diversity. The tree generated from current LOTUS data builds on biological taxonomy and employs the kingdom as tips label color (only families containing 50+ chemical structures were considered). The outer bars correspond to the most specific chemical class found in the biological family. The height of the bar is proportional to a specificity score corresponding to a Jaccard index between the chemical class and the biological family. The bar color corresponds to the chemical pathway of the most specific chemical class in the NPClassifier classification system. The size of the leaf nodes corresponds to the number of genera reported in the family. The figure is vectorized and zoomable for detailed inspection and is available under the CC0 license at https://commons.wikimedia.org/wiki/File:Lotus_initiative_1_chemically_interpreted_biological_tree.svg.

The online version of this article includes the following figure supplement(s) for figure 7:

**Figure supplement 1.** Distribution of β-sitosterol and related chemical parents among families with at least 50 reported compounds present in LOTUS.

have a higher biosynthetic specificity toward terpenoids than other fungi, which mostly focus on polyketides production. As explained previously, *Figure 7* is highly dependent of the data reported in literature. As also illustrated in *Figure 4*, some compounds can be over-studied among several organisms, and many organisms studied for specific compounds only. This is a direct consequence of the way the NP community reports its data actually. Having this in mind, when observed at a finer scale, down to the structure level, such chemotaxonomic representation can give valuable insights. For example, among all chemical structures, only two were found in all biological kingdoms, namely heptadecanoic acid (KEMQGTRYUADPNZ-UHFFFAOYSA-N) and *β*-carotene (OENHQHLEOONYIE-JLTXGRSLSA-N). Looking at the distribution of *β*-sitosterol (KZJWDPNRJALLNS-VJSFXXLFSA-N) within the overall biological tree, *Figure 7—figure supplement 1* plots its presence/absence *versus* those of its superior chemical classifications, namely the stigmastane, steroid and terpenoid derivatives, over the same tree used in *Figure 7*. The comparison of these five chemically interpreted biological trees clearly highlights the increasing speciation of the *β*-sitosterol biosynthetic pathway in the Archaeplastida kingdom, while the superior classes are distributed across all kingdoms. *Figure 7* is zoomable and vectorized for detailed inspection.

As illustrated, the possibility of data interrogation at multiple precision levels, from fully defined chemical structures to broader chemical classes, is of great interest, for example, for taxonomic and evolution studies. This makes LOTUS a unique resource for the advancement of chemotaxonomy, a discipline pioneered by Augustin Pyramus de Candolle and pursued by other notable researchers (Robert Hegnauer, Otto R. Gottlieb) (*Gottlieb, 1982*; *Hegnauer, 1986a*; *Candolle, 1816*). Six decades after Hegnauer's publication of 'Die Chemotaxonomie der Pflanzen' (*Hegnauer, 1986b*) much remains to be done for the advancement of this field of study and the LOTUS initiative aims to provide a solid basis for researchers willing to pursue these exciting explorations at the interface of chemistry, biology and evolution.

As shown recently in the context of spectral annotation (*Dührkop et al., 2021*), lowering the precision level of the annotation allows a broader coverage along with greater confidence. Genetic studies investigating the pathways involved and the organisms carrying the responsible biosynthetic genes would be of interest to confirm the previous observations. These forms of data interpretation exemplify the importance of reporting not only new structures, but also novel occurrences of known structures in organisms as comprehensive chemotaxonomic studies are pivotal for a better understanding of the metabolomes of living organisms.

The integration of multiple knowledge sources, for example genetics for NP producing gene clusters (*Kautsar et al., 2020*) combined to taxonomies and occurrences DB, also opens new opportunities to understand if an organism is responsible for the *biosynthesis* of a NP or merely *contains* it. This understanding is of utmost importance for the chemotaxonomic field and will help to understand to which extent microorganisms (endosymbionts) play a role in host development and its NP expression potential (*Saikkonen et al., 2004*).

## Conclusion and Perspectives

### Advancing natural products knowledge

At its current development stage, data harmonized and curated throughout the LOTUS initiative remains imperfect and, by the very nature of research, at least partially biased (see Introduction). In the context of bioactive NP research, and due to global editorial practices, it should not be ignored that many publications tend to emphasize new compounds and/or those for which interesting bioactivity has been measured. Near-ubiquitous (primarily plant-based) compounds, if broadly bioactive, tend to be overrepresented in the NP literature, yet the implication of their wide distribution in nature and associated patterns of non-specific activity are often underappreciated (*Bisson et al., 2016b*). Ideally, all characterized compounds independent of structural novelty and/or bioactivity profile should be documented, and the sharing of verified structure-organism pairs is fundamental to the advancement of NP research.

The LOTUS initiative provides a framework for rigorous review and incorporation of new records and already presents a valuable overview of the distribution of NP occurrences studied to date. While current data presents a reasonable approximation of the chemistries of a few well-studied organisms such as *Arabidopsis thaliana*, it remains patchy for many other organisms represented in the dataset. Community participation is the most efficient means of achieving a better documentation of NP

occurrences, and the comprehensive editing opportunities provided within LOTUS and through the associated Wikidata distribution platform open new opportunities for such collaborative engagement. In addition to facilitating the introduction of new data, it also provides a forum for critical review of existing data (see an example of a Wikidata Talk page here), as well as harmonization and verification of existing NP datasets as they come online.

## Fostering FAIRness and TRUSTworthiness

The LOTUS harmonized data and dissemination of referenced structure-organism pairs through Wikidata, enables novel forms of queries and transformational perspectives in NP research. As LOTUS follows the guidelines of FAIRness and TRUSTworthiness, all researchers across disciplines can benefit from this opportunity, whether the interest is in ecology and evolution, chemical ecology, drug discovery, biosynthesis pathway elucidation, chemotaxonomy, or other research fields connected with NP.

Researchers worldwide uniformly acknowledge the limitations caused by the intrinsic unavailability of essential (raw) data (*Bisson et al., 2016a*). In addition to being FAIR, LOTUS data is also open with a clear license, while closed data is still a major impediment to advancement of science (*Murray-Rust, 2008*). The lack of progress in such direction is partly due to elements in the dissemination channels of the classical print and static PDF publication formats that complicate or sometimes even discourage data sharing, for example, due to page limitations and economically motivated mechanisms, including those involved in the focus on and calculation of journal impact factors. In particular raw data such as experimental readings, spectroscopic data, instrumental measurements, statistical, and other calculations are valued by all, but disseminated by only very few. The immense value of raw data and the desire to advance the public dissemination has recently been documented in detail for nuclear magnetic resonance (NMR) spectroscopic data by a large consortium of NP researchers (*McAlpine et al., 2019*). However, to generate the vital flow of contributed data, the effort associated with preparing and submitting content to open repositories as well as data reuse should be better acknowledged in academia, government, regulatory, and industrial environments (*Cousijn et al., 2019*; *Cousijn et al., 2018*; *Pierce et al., 2019*). The introduction of LOTUS provides here a new opportunity to advance the FAIR guiding principles for scientific data management and stewardship (*Wilkinson et al., 2016*).

## Opening new perspectives for spectral data

The possibilities for expansion and future applications of the Wikidata-based LOTUS initiative are significant. For example, properly formatted spectral data (e.g. obtained by MS or NMR) can be linked to the Wikidata entries of the originating chemical compounds. MassBank (*Horai et al., 2010*) and SPLASH (*Wohlgemuth et al., 2010*) identifiers are already reported in Wikidata, and this existing information can be used to report MassBank or SPLASH records for example for *Arabidopsis thaliana* compounds (https://w.wiki/3PJD). Such possibilities will help to bridge experimental data results obtained during the early stages of NP research with data that has been reported and formatted in different contexts. This opens exciting perspectives for structural dereplication, NP annotation, and metabolomic analysis. The authors have previously demonstrated that taxonomically informed metabolite annotation is critical for the improvement of the NP annotation process (*Rutz et al., 2019*). Alternative approaches linking structural annotation to biological organisms have also shown substantial improvements (*Hoffmann et al., 2021*). In this context, the LOTUS initiative offers new opportunities for linking chemical objects to both their biological occurrences and spectral information and should significantly facilitate such applications.

## Integrating chemodiversity, biodiversity, and human health

As shown in *Figure 7—figure supplement 1*, observing the chemical and biological diversity at various granularities offers new insights. Regarding the chemical objects involved, it will be important to document the taxonomies of chemical annotations for the Wikidata entries. However, this is a rather complex task, for which stability and coverage issues will have to be addressed first. Existing chemical taxonomies such as ChEBI, ClassyFire, or NPClassifier are evolving steadily, and it will be important to constantly update the tools used to make further annotations. Promising efforts have been undertaken to automate the inclusion of Wikidata structures into a chemical ontology. Such

approach exploits the SMILES and SMARTS associated properties to infer a chemical classification for the structure. See for example, the entry related to emericellolide B. Repositioning NP within their greater biosynthetic context is another major challenge - and active field of research. The fact that the LOTUS initiative disseminates its data through Wikidata will help facilitate further integration with biological pathway knowledge bases such as WikiPathways and contribute to this complex task (*Martens et al., 2021*; *Slenter et al., 2018*).

In the field of ecology, molecular traits are gaining increased attention (*Kessler and Kalske, 2018*; *Sedio, 2017*; *Taylor and Dunn, 2018*). The LOTUS architecture can help to associate classical plant traits (e.g. leaf surface area, photosynthetic capacities, etc.) with Wikidata biological organisms entries, and, thus, allow their integration and comparison with chemicals that are associated with the organisms. Likewise, the association of biogeography data documented in repositories such as GBIF could be further exploited in Wikidata to pursue the exciting but understudied topic of 'chemodiversity hotspots' (*Defossez et al., 2021*).

Other NP-related information of great interest remains poorly formatted. One example of such a shortcoming relates to traditional medicine (and the field studying it: ethnomedicine and ethnobotany), which is the historical and empiric approach of mankind to discover and use bioactive products from Nature, primarily plants. The amount of knowledge generated in human history on the use of medicinal substances represents a fascinating yet underutilized amount of information. Notably, the body of literature on the pharmacology and toxicology of NP is compound-centric, increases steadily, and relatively scattered, but still highly relevant for exploring the role and potential utility of NP for human health. To this end, the LOTUS initiative represents a potential framework for new concepts by which such information could be valued and conserved in the digital era (*Allard et al., 2018*; *Cordell, 2017a*; *Cordell, 2017b*). This underscores the transformative value of the LOTUS initiative for the advancement of traditional medicine and its interest for drug discovery in health systems worldwide.

## Shortcomings and challenges

Despite these strong advantages, the establishment and functioning of the LOTUS curation pipeline is not devoid of defaults and we list hereafter some of the observed shortcomings and associated challenges.

First, the LOTUS processing pipeline is heavy. It includes many dependencies and is convoluted. We tried to simplify the process and associated programs as much as possible but they remain consequent. This is the consequence of the heterogeneous nature of the source information and the number of successive operations required to process the data.

Second, while the overall objective of the LOTUS processing pipeline is to increase data quality, the pipeline also *transforms* data during the process, and, in some cases, data quality can be degraded or errors can be propagated. For example, regarding the chemical objects, the processing pipeline performs a systematic sanitization step that includes salt removal, uncharging of molecules and dimers resolving. We decided to apply this step systematically after observing a high ratio of artifacts within salts, charged or dimeric molecules. This thus implies that correct salts, charged or dimeric molecules in the input data will suffer an unwanted 'sanitization' step. Also, the LOTUS processing step uses external libraries and tools for the automated 'name to structure' and 'structure to name' translations. These remain challenging as they rely on sets of predefined rules which do not cover all cases and can commonly lead to incorrect translations.

On the biological organisms curation side, we are aware of shortcomings, whether inherent to specific inputs or regarding limitations of the general process. Regarding inputs, some cases are clearly not resolvable except through human curation. For example, the word Lotus can refer both to the genus of a plant of the Fabaceae family (https://www.wikidata.org/wiki/Q3645698) or to the vernacular name of *Nelumbo nucifera* (https://www.wikidata.org/wiki/Q16528). In fact, the name of the LOTUS Initiative comes, in part, from this taxonomic curiosity - and the challenge for its automated curation. To give another striking illustration, *Ficus variegata* corresponds both to a plant (https://www.wikidata.org/wiki/Q5446649) and to a mollusc (https://www.wikidata.org/wiki/Q502030). For specific names coming from traditional Chinese medicine or other sources using vernacular names, translation was dependent on hand curated dictionaries, which are clearly not exhaustive. Additionally, it is noteworthy to remind that the validation of the processed entries relies in part on partly imperfect rules, thus leading to erroneous entries in the output data. However, we

also deliberately kept those rules restrictive in order to overall favor quality over quantity (see *Figure 2*).

Thus, despite our efforts, there is no doubt that incorrect structure-organism pairs have been uploaded on Wikidata (and some correct ones have not). We however expect that the editing facilities offered by this platform and community efforts will, over time, improve data quality.

## Summary and outlook

Despite these challenges, the various facets discussed above connect with ongoing and future developments that the tandem of the LOTUS initiative and its Wikidata integration can accommodate through a broader knowledge base. The information of the LOTUS initiative is already readily accessible by third party projects build on top of Wikidata such as the SLING project (https://github.com/ringgaard/sling, see entry for gliotoxin) or the Plant Humanities Lab project (https://lab.plant-humanities.org, see entry for *Ilex guayusa* in the 'From Related Items' section). LOTUS data has also been integrated to PubChem (https://pubchem.ncbi.nlm.nih.gov/source/25132) to complement the natural products related metadata of this major chemical DB. For an example, see *Gentiana lutea*.

Behind the scenes, all underlying resources represent data in a multidimensional space and can be extracted as individual graphs, which can then be interconnected. The craft of appropriate federated queries allows users to navigate these graphs and fully exploit their potential (*Waagmeester et al., 2020*; *Kratochvíl et al., 2018*). The development of interfaces such as RDFFrames (*Mohamed et al., 2020*) will also facilitate the use of the wide arsenal of existing machine learning approaches to automate reasoning on these knowledge graphs.

Overall, the LOTUS initiative aims to make more and better data available. While we did our best to ensure high data quality, the current processing pipeline still removes a lot of correct entries and misses or induces some incorrect ones. Aware of those imperfections, our project hopefully paves the way for the establishment of an open, durable and expandable electronic NP resource. The design and efforts of the LOTUS initiative reflect our conviction that the integration of NP research results is long-needed and requires a truly open and FAIR approach to information dissemination, with high quality data directly flowing from its source to public knowledge bases. We believe that the LOTUS initiative has the potential to fuel a virtuous cycle of research habits and, as a result, *contribute to a better understanding of Life and its chemistry*.

# Materials and methods

**Key resources table**

| Reagent type (species) or resource | Designation | Source or reference | Identifiers | Additional information |
|---|---|---|---|---|
| Software, algorithm | Lotus-processor code | This work (https://github.com/lotusnprod/lotus-processor, *Rutz, 2022a*) | | Archived at https://doi.org/10.5281/zenodo.5802107 |
| Software, algorithm | Lotus-web code | This work (https://github.com/lotusnprod/lotus-web, *Rutz, 2022b*) | | Archived at https://doi.org/10.5281/zenodo.5802119 |
| Software, algorithm | Lotus-wikidata-interact code | This work (https://github.com/lotusnprod/lotus-wikidata-interact, *Rutz, 2022c*) | | Archived at https://doi.org/10.5281/zenodo.5802113 |
| Software, algorithm | Global Names Architeture | https://globalnames.org | QID:Q65691453 | See Additional executable files |
| Software, algorithm | Java | https://www.java.com | QID:Q251 | |
| Software, algorithm | Kotlin | https://kotlinlang.org | QID:Q3816639 | See Kotlin packages |
| Software, algorithm | Manubot | https://manubot.org | QID:Q96473455 RRID:SCR_018553 | Repository available at https://github.com/lotusnprod/lotus-manuscript |
| Software, algorithm | NPClassifier | https://npclassifier.ucsd.edu | | See https://doi.org/10.1021/acs.jnatprod.1c00399 |
| Software, algorithm | OPSIN | https://github.com/dan2097/opsin | QID:Q26481302 | See Additional executable files |
| Software, algorithm | Python Programming Language | https://www.python.org | QID:Q28865 RRID:SCR_008394 | See Python packages |
| Software, algorithm | R Project for Statistical Computing | https://www.r-project.org | QID:Q206904 RRID:SCR_001905 | See R packages |

*Continued on next page*

*Continued*

| Reagent type (species) or resource | Designation | Source or reference | Identifiers | Additional information |
|---|---|---|---|---|
| Software, algorithm | Molconvert | https://docs.chemaxon.com/display/docs/molconvert.md | QID:Q55377678 | See Chemical structures |
| Software, algorithm | Wikidata | https://www.wikidata.org | QID:Q2013 RRID:SCR_018492 | Project page https://www.wikidata.org/wiki/Wikidata:WikiProject_Chemistry/Natural_products |
| Other | Lotus custom dictionaries | This work | | Archived at https://doi.org/10.5281/zenodo.5801798 |
| Other | Chemical identifier resolver | https://cactus.nci.nih.gov/chemical/structure | | See Chemical structures |
| Other | CrossRef | https://www.crossref.org | QID:Q5188229 RRID:SCR_003217 | See References |
| Other | PubChem | https://pubchem.ncbi.nlm.nih.gov | QID:Q278487 RRID:SCR_004284 | LOTUS data https://pubchem.ncbi.nlm.nih.gov/source/25132 |
| Other | PubMed | https://pubmed.ncbi.nlm.nih.gov | QID:Q180686 RRID:SCR_004846 | See References |
| Other | Taxonomic data sources | https://resolver.globalnames.org/data_sources | | See Translation |
| Other | Natural Products data sources | | | See Appendix 1 |

## Data gathering

Before their inclusion, the overall quality of the source was manually assessed to estimate, both, the quality of referenced structure-organism pairs and the lack of ambiguities in the links between data and references. This led to the identification of thirty-six electronic NP resources as valuable LOTUS input. Data from the proprietary Dictionary of Natural Products (DNP v 29.2) was also used for comparison purposes only and is not publicly disseminated. FooDB was also curated but not publicly disseminated since its license proscribed sharing in Wikidata. Appendix 1 gives all necessary details regarding electronic NP resources access and characteristics.

Manual inspection of each electronic NP resource revealed that the structure, organism, and reference fields were widely variable in format and contents, thus requiring standardization to be comparable. The initial stage consisted of writing tailored scripts that are capable of harmonizing and categorizing knowledge from each source (*Figure 1*). This transformative process led to three categories: fields relevant to the chemical structure described, to the producing biological organism, and the reference describing the occurrence of the chemical structure in the producing biological organism. This process resulted in categorized columns for each source, providing an initial harmonized format for each table.

For all thirty-eight sources, if a single file or multiple files were accessible via a download option including FTP, data was gathered that way. For some sources, data was scraped (cf. Appendix 1). All scraping scripts can be found in the lotus-processor repository in the src/1_gathering directory (under each respective subdirectory). Data extraction scripts for the DNP are available and should allow users with a DNP license only to further exploit the data (src/1_gathering/db/dnp). The chemical structure fields, organism fields, and reference fields were manually categorized into three, two, and ten subcategories, respectively. For chemical structures, "InChI", "SMILES", and "chemical name" (not necessarily IUPAC). For organisms, "clean" and "dirty", meaning lot text not referred to the canonical name was present or the organism was not described by its canonical name (e.g. "Compound isolated from the fresh leaves of *Citrus* spp."). For the references, the original reference was kept in the "original" field. When the format allowed it, references were divided into: "authors", "doi", "external", "isbn", "journal", "original", "publishing details", "pubmed", "title", "split". The generic "external" field was used for all external cross-references to other websites or electronic NP resources (e.g. "also in knapsack"). The last subcategory, "split", corresponds to a still non-atomic field after the removal of parts of the original reference. Other field titles are self-explanatory. The producing organism field was kept as a single field.

## Data harmonization

To perform the harmonization of all previously gathered sources, sixteen columns were chosen as described above. Upon electronic NP resources harmonization, resulting subcategories were divided and subject to further processing. The 'chemical structure' fields were divided into files according to their subcategories ("InChI", "names" and "SMILES"). A file containing all initial structures from all three subcategories was also generated. The same procedure was followed for organisms and references.

## Data processing

To obtain an unambiguously referenced structure-organism pair for Wikidata dissemination, the initial sixteen columns were translated and processed into three fields: the reported structure, the organism canonical name, and the reference. The structure was reported as InChI, together with its SMILES and InChIKey translation. The biological organism field was reported as three minimal necessary and sufficient fields, namely its canonical name and the taxonID and taxonomic DB corresponding to the latter. The reference was reported as four minimal fields, namely reference title, DOI, PMCID, and PMID, one being sufficient. For the forthcoming translation processes, automated solutions were used when available. However, for specific cases (common or vernacular names of the biological organisms, Traditional Chinese Medicine (TCM) names, and conversion between digital reference identifiers), no solution existed, thus requiring the use of tailored dictionaries. Their construction is detailed in the Dictionaries section. The initial entries (containing one or multiple producing organisms per structure, with one or multiple accepted names per organism) were processed into 2M+ referenced structure-organism pairs.

## Chemical structures

To retrieve as much information as possible from the original structure field(s) of each of the sources, the following procedure was followed. Allowed structural fields for the sources were divided into two types: structural (InChI, SMILES) or nominal (chemical name, not necessarily IUPAC). If multiple fields were present, structural identifiers were preferred over structure names. Among structural identifiers, when both identifiers were present, SMILES was preferred over InChI. InChI were translated to SMILES using the *RDKit, 2021* implementation in Python 3.8 (src/2_curating/2_editing/structure/1_translating/inchi.py). They were first converted to ROMol objects which were then converted to SMILES. When no structural identifier was available, the nominal identifier was translated to InChI first thanks to OPSIN (*Lowe et al., 2011*), a fast Java-based translation open-source solution. If no translation was obtained, chemical names were then submitted to the PUG-REST, the interface for programmatic access to PubChem (*Kim et al., 2018*; *Kim et al., 2015b*). If again no translation was obtained, candidates were then submitted to the Chemical Identifier Resolver. Before the translation process, some typical chemical structure-related greek characters (such as $\alpha$, $\beta$) were replaced by their textual equivalents (alpha, beta) to obtain better results. All pre-translation steps are included in the preparing_name function and are available in src/r/preparing_name.R.

The chemical sanitization step sought to standardize the representation of chemical structures coming from different sources. It consisted of three main stages (standardizing, fragment removal, and uncharging) achieved *via* the MolVS package. The initial standardizer function consists of six stages (RDKit Sanitization, RDKit Hs removal, Metals Disconnection, Normalization, Acids Reionization, and Stereochemistry recalculation) detailed in the molvs documentation. In a second step, the FragmentRemover functionality was applied using a list of SMARTS to detect and remove common counterions and crystallization reagents sometimes occurring in the input DB. Finally, the Uncharger function was employed to neutralize molecules when appropriate.

Molconvert function of the MarvinSuite was used for traditional and IUPAC names translation, Marvin 20.19, ChemAxon. When stereochemistry was not fully defined, (+) and (-) symbols were removed from names. All details are available in the following script: src/2_curating/2_editing/structure/4_enriching/naming.R. Chemical classification of all resulting structures was done using classyfireR (*Djoumbou Feunang et al., 2016*) and NPClassifier API.

After manual evaluation, structures remaining as dimers were discarded (all structures containing a "." in their SMILES were removed).

From the 283,267 initial InChI, 242,068 (85%) sanitized structures were obtained, of which 185,929 (77%) had complete stereochemistry defined. A total of 203,718 (72%) were uploaded to Wikidata. From the 248,185 initial SMILES, 207,658 (84%) sanitized structures were obtained, of which 98,685 (48%) had complete stereochemistry defined. 174,091 (70%) were uploaded to Wikidata. From the 49,675 initial chemical names, 27,932 (56%) sanitized structures were obtained, of which 17,460 (63%) had complete stereochemistry defined. 23,036 (46%) were uploaded to Wikidata. In total, 163,800 structures with fully defined stereochemistry were uploaded as "chemical compounds" (Q11173), and 106,669 structures without fully defined stereochemistry were uploaded as "group of stereoisomers" (Q59199015).

## Biological organisms

The processing at the biological organism's level had three objectives: convert the original organism string to (a) taxon name(s), atomize fields containing multiple taxon names, and deduplicate synonyms. The original organism strings were treated with Global Names Finder (GNF) and Global Names Verifier (GNV), both tools coming from the Global Names Architecture (GNA) a system of web services that helps people to register, find, index, check, and organize biological scientific names and interconnect on-line information about species. GNF allows scientific name recognition within raw text blocks and searches for found scientific names among public taxonomic DB. GNV takes names or lists of names and verifies them against various biodiversity data sources. Canonical names, their taxonID, and the taxonomic DB they were found in were retrieved. When a single entry led to multiple canonical names (accepted synonyms), all of them were kept. Because both GNF and GNV recognize scientific names and not common ones, common names were translated before a second resubmission.

## Dictionaries

To perform the translations from common biological organism name to latin scientific name, specialized dictionaries included in DrDuke, FooDB, PhenolExplorer were aggregated together with the translation dictionary of GBIF Backbone Taxonomy. The script used for this was src/1_gathering/translation/common.R. When the canonical translation of a common name contained a specific epithet that was not initially present, the translation pair was discarded (for example, "Aloe" translated in "*Aloe vera*" was discarded). Common names corresponding to a generic name were also discarded (for example "Kiwi" corresponding to the synonym of an *Apteryx* spp. (https://www.gbif.org/species/4849989)). When multiple translations were given for a single common name, the following procedure was followed: the canonical name was split into species name, genus name, and possible subnames. For each common name, genus names and species names were counted. If both the species and genus names were consistent at more than 50%, they were considered consistent overall and, therefore, kept (for example, "Aberrant Bush Warbler" had "*Horornis flavolivaceus*" and "*Horornis flavolivaceus intricatus*" as translation; as both the generic ("*Horornis*") and the specific ("*flavolivaceus*") epithets were consistent at 100%, both ("*Horornis flavolivaceus*") were kept). When only the generic epithet had more than 50% consistency, it was kept (for example, "Angelshark" had "*Squatina australis*" and "*Squatina squatina*" as translation, so only "*Squatina*" was kept). Some unspecific common names were removed (see https://doi.org/10.5281/zenodo.5801816 *Rutz, 2021*) and only common names with more than three characters were kept. This resulted in 181,891 translation pairs further used for the conversion from common names to scientific names. For TCM names, translation dictionaries from TCMID, TMMC, and coming from the Chinese Medicine Board of Australia were aggregated. The script used for this was src/1_gathering/translation/tcm.R. Some unspecific common names were removed (see https://doi.org/10.5281/zenodo.5801816 *Rutz, 2021*). Careful attention was given to the Latin genitive translations and custom dictionaries were written (see https://doi.org/10.5281/zenodo.5801816 *Rutz, 2021*). Organ names of the producing organism were removed to avoid wrong translation (see https://doi.org/10.5281/zenodo.5801816 *Rutz, 2021*). This resulted in 7,070 translation pairs. Both common and TCM translation pairs were then ordered by decreasing string length, first translating the longer names to avoid part of them being translated incorrectly.

## Translation

To ensure compatibility between obtained taxonID with Wikidata, the taxonomic DB 3 (ITIS), 4 (NCBI), 5 (Index Fungorum), 6 (GRIN Taxonomy for Plants), 8 (The Interim Register of Marine and Nonmarine

Genera), 9 (World Register of Marine Species), 11 (GBIF Backbone Taxonomy), 12 (Encyclopedia of Life), 118 (AmphibiaWeb), 128 (ARKive), 132 (ZooBank), 147 (Database of Vascular Plants of Canada (VASCAN)), 148 (Phasmida Species File), 150 (USDA NRCS PLANTS Database), 155 (FishBase), 158 (EUNIS), 163 (IUCN Red List of Threatened Species), 164 (BioLib.cz), 165 (Tropicos - Missouri Botanical Garden), 167 (The International Plant Names Index), 169 (uBio NameBank), 174 (The Mammal Species of The World), 175 (BirdLife International), 179 (Open Tree of Life), 180 (iNaturalist), and 187 (The eBird/Clements Checklist of Birds of the World) were chosen. All other available taxonomic DB are listed at http://index.globalnames.org/datasource. To retrieve as much information as possible from the original organism field of each of the sources, the following procedure was followed: First, a scientific name recognition step, allowing us to retrieve canonical names was carried (src/2_curating/2_editing/organisms/subscripts/1_processingOriginal.R). Then, a subtraction step of the obtained canonical names from the original field was applied, to avoid unwanted translation of parts of canonical names. For example, *Bromus mango* contains "mango" as a specific epithet, which is also the common name for *Mangifera indica*. After this subtraction step, the remaining names were translated from vernacular (common) and TCM names to scientific names, with help of the dictionaries. For performance reasons, this processing step was written in Kotlin and used coroutines to allow efficient parallelization of that process (src/2_curating/2_editing/organisms/2_translating_organism_kotlin/). They were subsequently submitted again to scientific name recognition (src/2_curating/2_editing/organisms/3_processingTranslated.R).

After full resolution of canonical names, all obtained names were submitted to rotl (*Michonneau et al., 2016*) to obtain a unified taxonomy. From the 88,395 initial "clean" organism fields, 43,936 (50%) canonical names were obtained, of which 32,285 (37%) were uploaded to Wikidata. From the 300 initial "dirty" organism fields, 250 (83%) canonical names were obtained, of which 208 (69%) were uploaded to Wikidata.

ReferenceThe Rcrossref package (*Chamberlain et al., 2020*) interfacing with the Crossref API was used to translate references from their original subcategory ("original", "publishingDetails", "split", "title") to a DOI, the title of its corresponding article, the journal it was published in, its date of publication and the name of the first author. The first twenty candidates were kept and ranked according to the score returned by Crossref, which is a tf-idf score. For DOI and PMID, only a single candidate was kept. All DOIs were also translated with this method, to eventually discard any DOI not leading to an object. PMIDs were translated, thanks to the entrez_summary function of the rentrez package (*Winter, 2017*). Scripts used for all subcategories of references are available in the directory src/2_curating/2_editing/reference/1_translating/. Once all translations were made, results coming from each subcategory were integrated, (src/2_curating/2_editing/reference/2_integrating.R) and the producing organism related to the reference was added for further treatment. Because the crossref score was not informative enough, at least one other metric was chosen to complement it. The first metric was related to the presence of the producing organism's generic name in the title of the returned article. If the title contained the generic name of the organism, a score of 1 was given, else 0. Regarding the subcategories "doi", "pubmed" and "title", for which the same subcategory was retrieved *via* crossref or rentrez, distances between the input's string and the candidates' one were calculated. Optimal string alignment (restricted Damerau-Levenshtein distance) was used as a method. Among "publishing details", "original" and "split" categories, three additional metrics were used: If the journal name was present in the original field, a score of 1 was given, else 0. If the name of the first author was present in the original field, a score of 1 was given, else 0. Those three scores were then summed together. All candidates were first ordered according to their crossref score, then by the complement score for related subcategories, then again according to their title-producing organism score, and finally according to their translation distance score. After this re-ranking step, only the first candidate was kept. Finally, the Pubmed PMCID dictionary (PMC-ids.csv.gz) was used to perform the translations between DOI, PMID, and PMCID (src/2_curating/2_editing/reference/3_processing.R).

From the 36,710 initial "original" references, 21,970 (60%) references with sufficient quality were obtained, of which 15,588 (71%) had the organism name in their title. 14,710 (40%) were uploaded to Wikidata. From the 21,953 initial "pubmed" references, 9452 (43%) references with sufficient quality were obtained, of which 6,098 (65%) had the organism name in their title. 5553 (25%) were uploaded to Wikidata. From the 37,371 initial "doi" references, 20,139 (54%) references with sufficient quality were obtained, of which 15,727 (78%) had the organism name in their title. 15,351 (41%)

were uploaded to Wikidata. From the 29,600 initial "title" references, 17,417 (59%) references with sufficient quality were obtained, of which 12,675 (73%) had the organism name in their title. 10,725 (36%) were uploaded to Wikidata. From the 11,325 initial "split" references, 5856 (52%) references with sufficient quality were obtained, of which 3,206 (55%) had the organism name in their title. 2,854 (25%) were uploaded to Wikidata. From the 3,314 initial "publishingDetails" references, 119 (4%) references with sufficient quality were obtained, of which 59 (50%) had the organism name in their title. 58 (2%) were uploaded to Wikidata.

## Data realignment

In order to fetch back the referenced structure-organism pairs links in the original data, the processed structures, processed organisms, and processed references were re-aligned with the initial entries. This resulted in 6.2M+ referenced structure-organism pairs. Those pairs were not unique, with redundancies among electronic NP resources and different original categories leading to the same final pair (for example, entry reporting InChI=1/C21H20O12/c22-6-13-15(27)17(29)18(30)21(32-13)33-20-16(28)14-11(26)4-8(23)5-12(14)31-19(20)7-1-2-9(24)10(25)3-7/h1-5,13,15,17–18,21-27,29–30H,6H2/t13-,15+,17+,18-,21+/m1/s1 in *Crataegus oxyacantha* or InChI=1S/C21H20O12/c22-6-13-15(27)17(29)18(30)21(32-13)33-20-16(28)14-11(26)4-8(23)5-12(14)31-19(20)7-1-2-9(24)10(25)3-7/h1-5,13,15,17–18,21-27,29–30 H,6H2/t13-,15+,17+,18-,21+/m1/s1 in *Crataegus stevenii* both led to OVSQVDMCBVZWGM-DTGCRPNFSA-N in *Crataegus monogyna*). After deduplication, 2M+ unique structure-organism pairs were obtained.

After the curation of all three objects, all of them were put together again. Therefore, the original aligned table containing the original pairs was joined with each curation result. Only entries containing a structure, an organism, and a reference after curation were kept. Each curated object was divided into minimal data (for Wikidata upload) and metadata. A dictionary containing original and curated object translations was written for each object to avoid those translations being made again during the next curation step (src/2_curating/3_integrating.R).

## Data validation

The pairs obtained after curation were of different quality. Globally, structure and organism translation was satisfactory whereas reference translation was not. Therefore, to assess the validity of the obtained results, a randomized set of 420 referenced structure-organism pairs was sampled in each reference subcategory and validated or rejected manually. Entries were sampled with at least 55 of each reference subcategory present (to get a representative idea of each subcategory) (src/3_analyzing/1_sampling.R). An entry was only validated if: (*i*) the structure (as any structural descriptor that could be linked to the final sanitized InChIKey) was described in the reference (*ii*) the producing organism (as any organism descriptor that could be linked to the accepted canonical name) was described in the reference and (*iii*) the reference was describing the occurrence of the chemical structure in the biological organism. Results obtained on the manually analyzed set were categorized according to the initial reference subcategory and are detailed in Appendix 2. To improve these results, further processing of the references was needed. This was done by accepting entries whose reference was coming from a DOI, a PMID, or from a title which restricted Damerau-Levenshtein distance between original and translated was lower than ten or if it was coming from one of the three main journals where NP occurrences are commonly expected to be published (i.e. *Journal of Natural Products*, *Phytochemistry*, or *Journal of Agricultural and Food Chemistry*). For "split", "publishingDetails" and "original" subcategories, the year of publication of the obtained reference, its journal, and the name of the first author were searched in the original entry and if at least two of them were present, the entry was kept. Entries were then further filtered to keep the ones where the reference title contained the first element of the detected canonical name. Except for COCONUT, exceptions to this filter were made for all DOI-based references. The function resulting from those rules is (filter_dirty.R). To validate those filtering criteria, an additional set of 100 structure-organism pairs were manually analyzed. F0.5 score was used as a metric. F0.5 score is a modified F1 score where precision has twice more weight than recall. The F-score was calculated with ß = 0.5, as in *Equation 1*:

$$F_\beta = \left(1 + \beta^2\right) \cdot \frac{precision \cdot recall}{(\beta^2 \cdot precision) + recall} \tag{1}$$

Based on this first manually validated dataset, filtering criteria (src/r/filter_dirty.R) were established to maximize precision and recall. Another 100 entries were sampled, this time respecting the whole set ratios. After manual validation, 97% of true positives were reached on the second set. A summary of the validation results is given in Appendix 2. Once validated, the filtering criteria were established to the whole curated set to filter entries chosen for dissemination (src/3_analyzing/2_validating.R).

## Unit testing

To provide robustness of the whole process and code, unit tests and partial data full-tests were written. They can run on the developer machine but also on the CI/CD system (GitHub) upon each commit to the codebase.

Those tests assess that the functions are providing results coherent with what is expected especially for edge cases detected during the development. The Kotlin code has tests based on JUnit and code quality control checks based on Ktlint, Detekt and Ben Mane's version plugin.

## Data dissemination

### Wikidata

All the data produced for this work has been made available on Wikidata under a Creative Commons 0 license according to Wikidata:Licensing. This is a "No-rights-reserved" license that places no restrictions on reuse.

### Lotus.NaturalProducts.Net (LNPN)

The web interface is implemented following the same protocol as described in the COCONUT publication (*Sorokina et al., 2021a*) that is the data are stored in a MongoDB repository, the backend runs with Kotlin and Java, using the Spring framework, and the frontend is written in React.js, and completely Dockerized. In addition to the diverse search functions available through this web interface, an API is also implemented, allowing programmatic LNPN querying. The complete API usage is described on the "Documentation" page of the website. LNPN is part of the NaturalProducts.net portal, an initiative aimed at gathering diverse open NP resources in one place.

## Data Interaction

### Data retrieval

Bulk retrieval of a frozen (2021-12-20) version of LOTUS data is also available at https://doi.org/10.5281/zenodo.5794106 (*Rutz et al., 2021a*).

The download lotus part of lotus-wikidata-interact allows the download of all chemical compounds with a "found in taxon" property. That way, it does not only get the data produced by this work, but any that would have existed beforehand or that would have been added directly on Wikidata by our users. It makes a copy of all the entities (compounds, taxa, references) into a local triplestore that can be queried with SPARQL as is or converted to a TSV file for inclusion in other projects. It is currently adapted to export directly into the SSOT thus allowing direct reuse by the processing/curation pipeline.

## Data addition

### Wikidata

Data is loaded by the Kotlin importer available in the upload lotus part of lotus-wikidata-interact repository under a GPL V3 license and imported into Wikidata. The importer processes the curated outputs grouping references, organisms, and compounds together. It then checks if they already exist in Wikidata (using SPARQL or a direct connection to Wikidata depending on the kind of data). It then uses *update* or *insert*, also called *upsert*, the entities as needed. The script currently takes the tabular file of the referenced structure-organism pairs resulting from the LOTUS curation process as input. Before upload, a filtering step is performed in order to avoid re-uploading entries we already uploaded. This way, if modifications occur in Wikidata, it will not be erased by the next iteration of the importer. The importer is currently being adapted to use directly the SSOT and avoid an unnecessary conversion step. To import references, it first double checks for the presence of duplicated DOIs and utilizes the Crossref REST API to retrieve metadata associated with the DOI, the support for other

citation sources such as Europe PMC is in progress. The structure-related fields are only subject to limited processing: basic formatting of the molecular formula by subscripting of the numbers. Due to limitations in Wikidata, the molecule names are dropped if they are longer than 250 characters and likewise the InChI strings cannot be stored if they are longer than 1500 characters.

Uploaded taxonomical DB identifiers are currently restricted to ITIS, GBIF, NCBI Taxon, Index Fungorum, IRMNG, WORMS, VASCAN, and iNaturalist, and newly OTL. The taxa levels are currently limited to family, subfamily, tribe, subtribe, genus, species, variety. The importer checks for the existence of each item based on their InChIKey and upserts the compound with the *found in taxon* statement and the associated organisms and references.

## LNPN

From the onset, LNPN has been importing data directly from the frozen tabular data of the LOTUS dataset (https://doi.org/10.5281/zenodo.5794106 *Rutz et al., 2021a*). In future versions, LNPN will directly feed on the SSOT.

## Data edition

The bot framework lotus-wikidata-interact was adapted such that, in addition to batch upload capabilities, it can also edit erroneously created entries on Wikidata. As massive edits have a large potential to disrupt otherwise good data, progressive deployment of this script is used, starting by editing progressively 1, 10, then 100 entries that are manually checked. Upon validation of 100 entries, the full script is run and check its behavior checked at regular intervals. An example of a corrected entry is as follows: https://www.wikidata.org/w/index.php?title=Q105349871&type=revision&diff=1365519277&oldid=1356145998.

## Curation interface

A web-based (Kotlin, Spring Boot for the back-end, and TypeScript with Vue for the front-end) curation interface is currently in construction. It will allow mass-editing of entries and navigate quick navigation in the SSOT for the curation of new and existing entries. This new interface is intended to become open to the public to foster the curation of entries by further means, driven by the users. In line with the overall LOTUS approach, any modification made in this curation interface will be mirrored after validation on Wikidata and LNPN.

## **Code availability**

### General repository

All programs written for this work can be found in the following group: https://github.com/lotusnprod.

### Processing

The source data curation system is available at https://github.com/lotusnprod/lotus-processor (copy archived at swh:1:rev:78e6065d8eb9d0b0d11c2ea8de6ac66b445bca0e, *Rutz, 2022a*). This program takes the source data as input and outputs curated data, ready for dissemination. The first step involves checking if the source data has already been processed. If not, all three elements (biological organism, chemical structures, and references) are submitted to various steps of translation and curation, before validation for dissemination.

### Wikidata

The programs to interact with Wikidata are available at https://github.com/lotusnprod/lotus-wikidata-interact, (copy archived at swh:1:rev:92d19b8995a69f5bba39f438172ba425fdcc0f28, *Rutz, 2022c*). On the upload side, the program takes the processed data resulting from the lotusProcessor subprocess as input and uploads it to Wikidata. It performs a SPARQL query to check which objects already exist. If needed, it creates the missing objects. It then updates the content of each object. Finally, it updates the chemical compound page with a "found in taxon" statement complemented with a "stated in" reference. A publication importer creating an article page from a DOI is also available.

On the download side, the program takes the structured data in Wikidata corresponding to chemical compounds found in taxa with a reference associated as input and exports it in both RDF and

tabular formats for further use. Two subsequent options are (a) that the end-user can directly use the exported data.; or (b) that the exported data, which can be new or modified since the last iteration, is used as new source data in lotusProcessor.

## LNPN

The LNPN website and processing system is available at https://github.com/lotusnprod/lotus-web (copy archived at swh:1:rev:278a5ab82389ebd5df720b1876a1724d15937644, *Rutz, 2022b*). This system takes the processed data resulting from the lotusProcessor as input and uploads it on https://lotus.naturalproducts.net. The repository is not part of the main GitHub group as it benefits from already established pipelines developed by CS and MS. The website allows searches from different points of view, complemented with taxonomies for both the chemical and biological sides. Many chemical molecular properties and molecular descriptors that otherwise are unavailable in Wikidata are also provided.

## Code freezing

All repository hyperlinks in the manuscript point to the main branches by default. The links contain all programs and code and will eventually be updated to a publication branch using modifications resulting from the peer-reviewing process. As the code evolves, readers are invited to refer to the main branch of each repository for the most up-to-date code. A frozen version (2021-12-23) of all programs and code is also available in the LOTUS Zenodo community (5802107 (*Rutz et al., 2021f*), 5802113 (*Bisson et al., 2021*), and 5802120 *Sorokina et al., 2021b*).

## Programs and packages

### R

The R versions used for the project were 4.0.2 up to 4.1.2, and R-packages used were, in alphabetical order: ChemmineR (3.42.1) (*Cao et al., 2008*), chorddiag (0.1.3) (*Flor, 2020*), ClassyfireR (0.3.6) (*Djoumbou Feunang et al., 2016*), data.table (1.14.2) (*Dowle and Srinivasan, 2020*), DBI (1.1.1) (*Wickham and Müller, 2021*), gdata (2.18.0) (*Warnes et al., 2017*), ggalluvial (0.12.3) (*Brunson, 2020*), ggfittext (0.9.1) (*Wilkins, 2020*), ggnewscale (0.4.5) (*Campitelli, 2021*), ggraph (2.0.5) (*Pedersen, 2020*), ggstar (1.0.2) (*Xu, 2021*), ggtree (3.2.0) (*Yu, 2017*), ggtreeExtra (1.4.0) (*Xu et al., 2021*), jsonlite (1.7.2) (*Ooms, 2014*), pbmcapply (1.5.0) (*Kuang et al., 2019*), plotly (4.10.0) (*Sievert, 2020*), rcrossref(1.1.0.99) (*Chamberlain et al., 2020*), readxl (1.3.1) (*Wickham, 2018*), rentrez (1.2.3) (*Winter, 2017*), rotl (3.0.11) (*Michonneau et al., 2016*), rvest (1.0.2) (*Wickham, 2020*), splitstackshape (1.4.8) (*Mahto, 2019*), RSQLite (2.2.8) (*Müller et al., 2021*), stringdist (0.9.8) (*Loo, 2014*), stringi (1.7.6) (*Gagolewski, 2020*), tidyverse (1.3.1) (*Wickham et al., 2019*), treeio (1.18.0) (*Wang et al., 2020*), UpSetR (1.4.0) (*Gehlenborg, 2019*), webchem (1.1.1) (*Szöcs et al., 2020*), XML (3.99–0.8) (*Lang, 2020*), xml2 (1.3.3) (*Wickham and Hester, 2020*).

### Python

The Python version used was 3.7.12 up to 3.9.7, and the Python packages utilized were, in alphabetical order: cmcrameri (1.4) (*Crameri, 2021*; *Crameri et al., 2020*), faerun (0.3.20) (*Probst and Reymond, 2018a*), map4 (1.0) (*Capecchi et al., 2020*), matplotlib (3.5.0) (*Hunter, 2007*), Molvs (0.1.1), pandas (1.3.4) (*Reback et al., 2020*), rdkit (2021.09.2) (*RDKit, 2021*), scipy (1.7.3) (*Virtanen et al., 2020*), tmap (1.0.4) (*Probst and Reymond, 2020*).

### Kotlin

Kotlin packages used were as follows: Common: Kotlin 1.4.21 up to 1.6.0, Univocity 2.9.1, OpenJDK 15, Kotlin serialization 1.3.1, konnector 0.1.34, Log4J 2.14.1 Wikidata Importer Bot:, wdkt 0.12.1, CDK 2.5 (*Willighagen et al., 2017*), RDF4J 3.7.4, Ktor 1.6.5, KotlinXCli 0.3.3, Wikidata data processing: Shadow 5.0.0 Quality control and testing: Ktlint 10.2.0, Kotlinter 3.3.0, Detekt 1.15.0, Ben Mane's version plugin 0.36.0, Junit 5.8.1.

### Additional executable files

GNFinder v.0.16.3, GNVerifier v.0.6.1, OPSIN v.2.5.0 (*Lowe et al., 2011*).

## Data availability

A snapshot of the obtained data at the time of re-submission (2021-12-20) is available at the following Zenodo community: https://zenodo.org/communities/the-lotus-initiative and related record: 5793224 (*Rutz et al., 2021e*), 5794107 (*Rutz et al., 2021c*), 5794597 (*Rutz et al., 2021b*), 5801816 (*Rutz, 2021*). The https://lotus.nprod.net website is intended to gather news and features related to the LOTUS initiative in the future.

## Acknowledgements

AR, JLW, and PMA are thankful to the Swiss National Science Foundation for supporting part of this project through the SNF Sinergia grant CRSII5_189921. JB and AR are really thankful to JetBrains for the Free educational license of IntelliJ and the excellent support received on Youtrack. JB, JGG, and GFP gratefully acknowledge the support of this work by grant U41 AT008706 and supplemental funding to P50 AT000155 from NCCIH and ODS of the NIH. MS and CS are supported by the German Research Foundation within the framework ChemBioSys (Project-ID 239748522, SFB 1127). EW and DM acknowledge the Scholia grant from the Alfred P Sloan Foundation under grant number G-2019–11458. The work on the Wikidata IDSM/Sachem endpoint was supported by an ELIXIR CZ research infrastructure project grant (MEYS Grant No: LM2018131) including access to computing and storage facilities. The authors would like to thank Dmitry Mozzherin for his work done for the Global Names Architecture and related improvements. They would like to acknowledge help from the PubChem team in integrating LOTUS, especially Tiejun Cheng and Evan Bolton. The authors would also like to thank Layla Michán for starting to add pigment information on Wikidata. The authors would also like to thank the team behind Manubot (*Himmelstein et al., 2019*), used to write this manuscript. The authors would also like to thank contributors of all electronic NP resources used in this work and the NP community at large.

## Additional information

### Funding

| Funder | Grant reference number | Author |
| --- | --- | --- |
| Schweizerischer Nationalfonds zur Förderung der Wissenschaftlichen Forschung | CRSII5_189921 | Adriano Rutz<br>Jean-Luc Wolfender<br>Pierre-Marie Allard |
| Office of Dietary Supplements | P50 AT000155 | James G Graham<br>Guido F Pauli<br>Jonathan Bisson |
| Deutsche Forschungsgemeinschaft | 239748522 | Maria Sorokina<br>Christoph Steinbeck |
| Alfred P. Sloan Foundation | G-2019-11458 | Daniel Mietchen<br>Egon Willighagen |
| National Center for Complementary and Integrative Health | U41 AT008706 | James G Graham<br>Jonathan Bisson<br>Guido F Pauli |

The funders had no role in study design, data collection and interpretation, or the decision to submit the work for publication.

### Author contributions

Adriano Rutz, Conceptualization, Data curation, Formal analysis, Investigation, Methodology, Project administration, Software, Validation, Visualization, Wikidata, Writing – original draft, Writing – review and editing; Maria Sorokina, LNPN Website, Software, Writing – review and editing; Jakub Galgonek, Sachem, IDSM, Software, Writing – review and editing; Daniel Mietchen, Egon Willighagen, Ralf Stephan, Wikidata, Writing – review and editing; Arnaud Gaudry, Visualization, Writing – review and

editing; James G Graham, NAPRALERT, Writing – review and editing; Roderic Page, Wikidata; Jiří Vondrášek, Funding acquisition, Resources, Sachem, IDSM; Christoph Steinbeck, Funding acquisition, LNPN Website, Resources, Writing – review and editing; Guido F Pauli, Funding acquisition, NAPRA- LERT, Resources, Writing – review and editing; Jean-Luc Wolfender, Funding acquisition, Resources, Writing – review and editing; Jonathan Bisson, Conceptualization, Data curation, Formal analysis, Investigation, Methodology, NAPRALERT, Project administration, Resources, Software, Supervision, Validation, Writing – review and editing; Pierre-Marie Allard, Conceptualization, Data curation, Formal analysis, Funding acquisition, Investigation, Methodology, Project administration, Resources, Soft- ware, Supervision, Validation, Wikidata, Writing – original draft, Writing – review and editing

**Author ORCIDs**
Adriano Rutz http://orcid.org/0000-0003-0443-9902
Maria Sorokina http://orcid.org/0000-0001-9359-7149
Jakub Galgonek http://orcid.org/0000-0002-7038-544X
Daniel Mietchen http://orcid.org/0000-0001-9488-1870
Egon Willighagen http://orcid.org/0000-0001-7542-0286
Arnaud Gaudry http://orcid.org/0000-0002-3648-7362
Ralf Stephan http://orcid.org/0000-0002-4650-631X
Roderic Page http://orcid.org/0000-0002-7101-9767
Jiří Vondrášek http://orcid.org/0000-0002-6066-973X
Christoph Steinbeck http://orcid.org/0000-0001-6966-0814
Guido F Pauli http://orcid.org/0000-0003-1022-4326
Jean-Luc Wolfender http://orcid.org/0000-0002-0125-952X
Jonathan Bisson http://orcid.org/0000-0003-1640-9989
Pierre-Marie Allard http://orcid.org/0000-0003-3389-2191

**Decision letter and Author response**
Decision letter https://doi.org/10.7554/eLife.70780.sa1
Author response https://doi.org/10.7554/eLife.70780.sa2

# Additional files

**Supplementary files**
• Transparent reporting form
• Supplementary file 1. Wikidata Queries.
• Supplementary file 2. Wikidata Entry Creation Tutorial.

**Data availability**
A snapshot of the obtained data at the time of re-submission (2021-12-20) is available at the following Zenodo community: https://zenodo.org/communities/the-lotus-initiative and related records: https://zenodo.org/record/5793224, https://zenodo.org/record/5794107, https://zenodo.org/record/5794597 and https://zenodo.org/record/5801816. The https://lotus.nprod.net website is intended to gather news and features related to the LOTUS initiative in the future.

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

# Appendix 1

## Data sources list

**Appendix 1—table 1.** Data sources list.

| Database | Type | Initial retrieved unique entries | Cleaned referenced structure-organism pairs | Pairs validated for Wikidata export | Actual validated pairs on Wikidata | Website | Article | Retrieval | License status | Contact | Dump | Status |
|---|---|---|---|---|---|---|---|---|---|---|---|---|
| afrotryp | open | 313 | 93 | 55 | 54 | - | article (Ibezim et al., 2017) | download | license_copyright | Fidele Ntie-Kang or Ngozi Justina Nwodo | YES | unmaintained |
| alkamid | open | 4,416 | 2,639 | 2,309 | 2,160 | website | article (Boonen et al., 2012) | script | license_copyright | Bart De Spiegeleer | NO | maintained |
| antibase | commercial | 46,956 | 45,221 | - | - | - | - | - | - | - | NO | unmaintained |
| antimarin | commercial | 73,017 | 67,559 | - | - | - | - | - | - | - | NO | unmaintained |
| biofacquim | open | 531 | 519 | 519 | 511 | website (old version) | article_old; article_new (Pilón-Jiménez et al., 2019) | download | license_CCBY_4.0 | José Medina-Franco | YES | maintained |
| biophytmol | open | 543 | 558 | 322 | 308 | website | article (Sharma et al., 2014) | script | license_CCBY | Anshu Bhardwaj | NO | unmaintained |
| carotenoiddb | open | 2,922 | 639 | 530 | 485 | website | article (Yabuzaki, 2017) | script | license_copyright | yzjunko@gmail.com | NO | maintained |
| coconut | open | 5,757,872 | 5,723,691 | 153,981 | 140,877 | website | article (Sorokina and Steinbeck, 2020b) | download | license_CCBY_4.0 | Maria Sorokina | YES | maintained |
| cyanometdb | open | 1,905 | 1,631 | 1,621 | 1,605 | - | article (Jones et al., 2021) | download | license_CCBY_4.0 | elisabeth.janssen@eawag.ch | YES | maintained |
| datawarrior | open | 589 | 541 | 71 | 60 | website | article (Sander et al., 2015) | download | no_license | thomas.sander@idorsia.com | YES | retired |
| dianatdb | open | 290 | 323 | 115 | 111 | website | article (Madariaga-Mazón et al., 2021) | download | license_CCBY_NC | amadariaga@iquimica.unam.mx or kmtzm@unam.mx | YES | maintained |
| dnp | commercial | 205,072 | 254,573 | - | - | website | - | script | - | support@taylorfrancis.com | NO | maintained |
| drduke | open | 90,675 | 9,660 | 6,184 | 5,222 | website | - | download | license_CC0 | agref@usda.gov | YES | maintained |
| foodb | restricted | 81,941 | 39,662 | - | - | website | - | download | license_CCBY_NC | jreid3@ualberta.ca (Jennifer) | YES | unmaintained |
| inflamnat | open | 665 | 632 | 306 | 268 | - | article (Zhang et al., 2019) | download | license_copyright | xiaoweilie@ynu.edu.cn | YES | unmaintained |
| knapsack | open | 132,127 | 139,336 | 59,945 | 55,186 | website | article (Shinbo et al., 2006) | script | license_copyright | skanaya@gtc.naist.jp | NO | maintained |
| metabolights | open | 38,208 | 37,704 | 6,241 | 5,687 | website | article (Haug et al., 2020) | download | license_copyright | - | YES | maintained |
| mibig | open | 1,310 | 1,139 | 638 | 535 | website | article (Kautsar et al., 2020) | download | license_CCBY_4.0 | Tilmann Weber or Marnix Medema | YES | unmaintained |
| mitishamba | open | 1,071 | 534 | 294 | 291 | website | article (Derese et al., 2019) | script | license_copyright | - | NO | defunct |
| nanpdb | open | 5,752 | 6,383 | 5,937 | 5,283 | website | article (Ntie-Kang et al., 2017) | script | license_copyright | ntiekfidele@gmail.com stefan.guenther@pharmazie.uni-freiburg.de | NO | maintained |
| napralert | commercial | 681,401 | 392,498 | 294,818 | 270,743 | website | article (Graham and Farnsworth, 2010) | - | license_copyright | napralert@uic.edu | NO | defunct |

*Appendix 1—table 1 Continued on next page*

*Appendix 1—table 1 Continued*

| Database | Type | Initial retrieved unique entries | Cleaned referenced structure-organism pairs | Pairs validated for Wikidata export | Actual validated pairs on Wikidata | Website | Article | Retrieval | License status | Contact | Dump | Status |
|---|---|---|---|---|---|---|---|---|---|---|---|---|
| npass | open | 290,535 | 30,185 | 25,429 | 23,612 | website | article (*Zeng et al., 2018*) | download | license_CCBY_NC | phacyz@nus.edu.sg jiangyy@sz.tsinghua.edu.cn iaochen@163.com | YES | unmaintained |
| npatlas | open | 32,539 | 34,726 | 34,548 | 33,087 | website | article (*van Santen et al., 2019*; ) | download | license_CCBY_4.0 | rliningt@sfu.ca | YES | maintained |
| npcare | open | 7,763 | 5,878 | 3,790 | 3,538 | website | article (*Choi et al., 2017*) | download | license_CCBY_4.0 | choihwanho@gmail.com | YES | unmaintained |
| npedia | open | 82 | 99 | 28 | 28 | website | article (*Tomiki et al., 2006*) | script | no_license | hisyo@riken.jp npd@riken.jp | NO | defunct |
| nubbe | open | 2,189 | 2,340 | 2,340 | 2,119 | website | article (*Pilon et al., 2017*) - | | license_copyright | Vanderlan S. Bolzani | NO | maintained |
| pamdb | open | 3,046 | 2,820 | 24 | 24 | website | article (*Huang et al., 2018*) | download | license_CCBY_NC | awilks@rx.umaryland.edu aoglesby@rx.umaryland.edu mkane@rx.umaryland.edu | YES | unmaintained |
| phenolexplorer | open | 8,077 | 8,700 | 7,123 | 5,721 | website | article (*Rothwell et al., 2013*) | download | license_copyright | scalberta@iarc.fr | YES | unmaintained |
| phytohub | open | 2,349 | 1,145 | 132 | 94 | website | article (*Giacomoni et al., 2017*) | script | no_license | claudine.manach@inra.fr | YES | unmaintained |
| procardb | open | 6,556 | 6,278 | 60 | 55 | website | article (*Nupur et al., 2016*) | script | license_CCBY_4.0 | Anil Kumar PinnakaAshwani Kumar | NO | unmaintained |
| respect | open | 2,759 | 1,064 | 634 | 547 | website | article (*Sawada et al., 2012*) | download | license_CCBY_NC_2.1_Japan | ksaito@psc.riken.jp | YES | unmaintained |
| sancdb | open | 860 | 925 | 747 | 732 | website | article (*Hatherley et al., 2015*) | script | license_CCBY_4.0 | Özlem Tastan Bishop | NO | unmaintained |
| streptomedb | open | 71,638 | 33,217 | 20,715 | 18,395 | website | article (*Klementz et al., 2016*) | download | license_copyright | stefan.guenther@pharmazie.uni-freiburg.de | YES | maintained |
| swmd | open | 1,075 | 1,751 | 1,597 | 1,479 | website | article (*Davis and Vasanthi, 2011*) | script | license_CCBY_4.0 | Dicky.John@gmail.com | NO | unmaintained |
| tmdb | open | 2,116 | 533 | 26 | 24 | website | article (*Yue et al., 2014*) | script | license_copyright | Xiao-Chun WanGuan-Hu Bao | NO | unmaintained |
| tmmc | open | 15,033 | 7,833 | 5,826 | 4,015 | website | article (*Kim et al., 2015a*) | download | license_copyright | Jeong-Ju Lee | YES | unmaintained |
| tppt | open | 27,182 | 23,872 | 684 | 641 | website | article (*Günthardt et al., 2018*) | download | license_copyright | thomas.bucheli@agroscope.admin.ch | YES | unmaintained |
| unpd | open | 331,242 | 304,683 | 211,158 | 197,710 | website | article (*Gu et al., 2013*) - | | license_CCBY_4.0 | lirongc@pku.edu.cn xiaojxu@pku.edu.cn | NO | defunct |
| wakankensaku | open | 367 | 224 | 208 | 202 | website | - | script | - | - | NO | defunct |
| Wikidata | open | 951,268 | 960,611 | 959,747 | 919,752 | website | - | download | license_CC0 | - | YES | maintained |

# Appendix 2

## Summary of the validation statistics

**Appendix 2—table 1.** Summary of the Validation Statistics.

| Reference Type | First validation dataset (n = 420) | | | | | | | | Second validation dataset (n = 100) | |
|---|---|---|---|---|---|---|---|---|---|---|
| | True positive | False positive | False negative | True negative | Ratio | Precision | Recall | $F_{0.5}$ score | True positive | False negative |
| Original | 80 | 6 | 7 | 11 | 0.31 | 0.93 | 0.92 | 0.92 | 38 | 1 |
| Pubmed | 37 | 1 | 5 | 6 | 0.30 | 0.97 | 0.88 | 0.92 | 5 | 1 |
| DOI | 115 | 6 | 0 | 6 | 0.19 | 0.95 | 1.00 | 0.97 | 43 | 1 |
| Title | 38 | 2 | 0 | 16 | 0.12 | 0.95 | 1.00 | 0.97 | 7 | 0 |
| Split | 8 | 0 | 15 | 27 | 0.08 | 1.00 | 0.35 | 0.52 | 4 | 0 |
| Publishing details | 1 | 0 | 1 | 32 | 0.01 | 1.00 | 0.50 | 0.67 | 0 | 0 |
| Total | 279 | 15 | 28 | 98 | 1.00 | - | - | - | 97 | 3 |
| Corrected total | - | - | - | - | - | 0.96 | 0.89 | 0.91 | - | - |

