## [Editor Report]

Rutz et al. describe the LOTUS initiative, an open science database that contains over 750,000 referenced structure-organism pairs. Present both the data that they have made available in Wikidata, as well as an interactive web portal, LOTUS provides a powerful platform for mining literature for published data on structure-organism pairs. The strength of this initiative lies in the effort the authors had put in creating a database that is both reproducible and usable. The result is thus a complete and user-friendly product that will respond to people's needs.

---

## [Decision Letter]

**Decision letter after peer review:**

Thank you for submitting your article "The LOTUS Initiative for Open Natural Products Research: Knowledge Management through Wikidata" for consideration by *eLife*. Your article has been reviewed by 3 peer reviewers, and the evaluation has been overseen by a Reviewing Editor and Anna Akhmanova as the Senior Editor. The following individual involved in review of your submission has agreed to reveal their identity: Charles Tapley Hoyt (Reviewer #1).

The reviewers have discussed their reviews with one another, and the Reviewing Editor has drafted this to help you prepare a revised submission. Please notice you count with detailed comments from three reviewers. While all of them see potential in your ms, we encourage you to provide a meaningful revision and detailed point-by-point response to criticism.

Essential revisions:

1) Reviewers would like to see the authors clarify the scope of their work and address to what extent this database will provide the community with a comprehensive database of structure organisms pairs, that is actually useable to answer research questions. This could be achieved by adding concrete examples of how these data could be used to help readers/reviewers understand the scope.

2) Reviewers think the manuscript provides missing or incomplete documentation on the LOTUS processes and software, especially in terms of reproducibility of results. Providing open access to source code and data is an important first step, and additional (challenging!) steps are needed to help others independently build, and use, the provided software/data.

3) The authors have yet to provide compelling evidence of how their continuously managed (and updated) resources can be reliably cited in scholarly literature and incorporated in scientific workflows. In order to study and reference Wikidata, a versioned copy needs to be provided: Wikidata is updated constantly, and these constant stream of changes make it hard for others to verify results extracted from some older version of the Wikidata corpus unless a versioned copy is provided.

*Reviewer #1 (Recommendations for the authors):*

“A third fundamental element of a structure-organism pair is a reference to the experimental evidence that establishes the linkages between a chemical structure and a biologicl organism and a future-oriented electronic NP resource should contain only fully-referenced structure-organism pairs.”

Typo in "biologicl"

“Currently, no open, cross-kingdom, comprehensive, computer-interpretable electronic NP resource links NP and their producing organisms, along with referral to the underlying experimental work”.

missing "that"

"KNApSAck currently contains 50,000+ structures and 100,000+ structure-organism pairs. However, the organism field is not standardized and access to the data is not straightforward".

This is the first opportunity in the manuscript to describe in more detail the perils of previous databases, especially the mess that is KNApSAck, which you have no choice but to work on because of its ubiquity.

“NAPRALERT is not an open platform, employing an access model that provides only limited free searches of the dataset”.

There's an awful lot of praise for this database given that it is directly antithetical to the manuscript. Please provide further commentary contrasting the work in NAPRALERT (particularly, about its shortcomings as a closed resource) to LOTUS.

“FAIR and TRUST (Transparency, Responsibility, User focus, Sustainability and Technology) principles”.

If you really want to go down the buzzword bingo, I'd suggest making a table or going in depth into each point. These acronyms are, in my opinion, effectively meaningless from a technical perspective, so it falls on the authors of the paper who want to use them (as you are no doubt pressured to do in the modern publishing landscape) to define them and qualify their relevance to your more practical goals.

“…any researcher to contribute, edit and reuse the data with a clear and open CC0 license (Creative Commons 0).”

This is a really interesting point considering you have taken information from several unlicensed databases and several that have more permissive licenses. How do you justify this?

“The SSOT approach consists of a PostgreSQL DB that structures links and data schemes such that every data element has a single place”.

Why is Wikidata not the single source of truth? This means that the curation and generation of this dataset can never really be decentralized- it will always have to be maintained by someone who is the maintainer of the PostgreSQL database. What are the pros/cons for this?

“The LOTUS processing pipeline is tailored to efficiently include and diffuse novel or curated data directly from new sources or at the Wikidata level.”

This sentence is confusing

“All stages of the workflow are described on the git sites of the LOTUS initiative at https://gitlab.com/lotus7 and https://github.com/mSorok/LOTUSweb”

Later this became a problem for code review. Why are the code all in different places? There are "organizations" both on GitLab and GitHub to keep related code together. Further, putting code in a personal user's namespace makes it difficult to for potential community involvement, especially if the user becomes inactive. Also a matter of opinion, but most science is being done on GitHub. Using GitLab will likely limit the number of people who will interact with the repository. Highly suggested to move it to GitHub.

“All necessary scripts for data gathering and harmonization can be found in the lotus-processor repository in the src/1_gathering directory."

The reference to "SI 1 Data Sources List" does not include the actual license information in the table, but rather links to READMEs (which may not stay stable for the life of the manuscript). This should explicitly state which license each database has. Further, it seems a bit disingenuous since some of these links point to README pages that state there is no license given, such as for Biofacquim (and several others)

https://gitlab.com/lotus7/lotus-processor/-/blob/main/docs/licenses/biofacquim.md

How can you justify taking this content and redistributing it under a more permissive license?

Further, this table should contain versioning information for each database and a flag as to whether it is still being maintained, whether data can be accessed as a dump, and if there is a dump, if it is structured. Right now, the retrieval column is not sufficient for describing how the data is actually procured.

Can you cross-reference these databases to Wikidata pages? FAIRSharing pages? I'm sure following publication in the near- or mid-term, more of these databases will go down permanently, so there should be as much information about what they were available with this publication.

Small suggestion: right align all numbers in tables.

“This process allowed us to establish rules for automatic filtering and validation of the entries. The filtering was then applied to all entries.”

Please explain this process, in detail (and also link to the exact code that does it as you have done in other sections)

“Table 1: Example of a referenced structure-organism pair before and after curation”.

Would like some discussion of the importance of anatomical region in addition to organism. Obviously, there is importance in the production of the NP in a given region. This is highly granular and makes the problem more difficult, but even if you don't tackle it, it has to be mentioned.

Further, why is the organism not standardized to a database identifier? Why are the prefixes to go along with these ID spaces not mentioned (InChI-key for structure, DOI for reference, and what for organism?) Maybe one of the confounders is that there are many taxonomical databases for organisms with varying coverage – this should be addressed. On second pass through the review, I noticed this was described a bit more in the methods section, so this should be linked. Further, it's actually not clear even by the end of the methods section what identifiers are used in the end. Are Wikidata Q identifiers the single unifying identifier in the end?

“Figure 2: Alluvial plot of the data transformation flow within LOTUS during the automated curation and validation processes.”

Looks like more things are rejected than kept. What about having a human curator in the loop? Or should poorly curated stuff be thrown way forever? There was a small mention about building a human curation system, but I don't think this had the same focus.

“The figure highlights, for example, the essential contribution of the DOI category of references contained in NAPRALERT towards the current set of validated references in LOTUS”.

Is this to say LOTUS is constructed with the closed data in NAPRALERT? Or the small portion that was made open was sufficient?

“…any researcher has a means of providing new data for LOTUS, keeping in mind the inevitable delay between data addition and subsequent inclusion into LOTUS.”

How likely do you think users will be to contribute their data to LOTUS, when there are many competing interests, like publishing their own database on their own website, which will support their own publication (and then likely go into obscurity)?

Ultimately, this means there should be a long-term commitment from the LOTUS initiative to continue to find new databases as they're published, to do outreach to authors (if possible/necessary), and to incorporate the new databases into the LOTUS pipeline as a third party. This should be described in detail – which organizations will do this/how will they fund it?

“For this, biological organisms were grouped into four artificial taxonomic levels (plants, fungi, animals and bacteria).”

How? Be more specific about methodology including code/dependencies

“Figure 6: TMAP visualizations of the chemical diversity present in LOTUS. Each dot”.

This color scheme is hard to interpret. Please consider using figure 6 in https://www.nature.com/articles/s41467-020-19160-7 to help pick a better color scheme.

“The biological specificity score at a given taxonomic level for a given chemical class is calculated as the number of structure-organism pairs within the taxon where the chemical class occurs the most, divided by the total number of pairs.”

Have you considered the Kullback-Leibler divergence (https://en.wikipedia.org/wiki/Kullback%E2%80%93Leibler_divergence) which is often used in text mining to compare the likelihood of a query in a given subset of a full corpus? I think this would also be appropriate for this setting.

“This specificity score was calculated as in Equation 2:”

This is a very similar look to the classic Jaccard Index (https://en.wikipedia.org/wiki/Jaccard_index; which is skewed by large differences in the size of the two sets) and the overlap coefficient (https://en.wikipedia.org/wiki/Overlap_coefficient). I think the multiplication of the sizes of the sets will have the same issue as Jaccard index. What is the justification for using this non-standard formulation? Please comment if the conclusions change from using the overlap coefficient.

Since this is supporting Figure 7, a new way to explore the chemical and biological diversity, and you previously wrote the huge skew in organisms and taxa over which information is available, it would be important to comment on the possible bias in this figure. Were you trying to make the point that the NPClassifier system helps reduce this bias? If so, it was unclear.

“As LOTUS follows the guidelines of FAIRness and TRUSTworthiness, all researchers across disciplines can benefit from this opportunity, whether the interest is in ecology and evolution, chemical ecology, drug discovery, biosynthesis pathway elucidation, chemotaxonomy, or other research fields that connect with NP.”

This was emphatically not qualified within the manuscript. The expansions for FAIR and TRUST were given, but with no explanation for what they are. Given that these are buzzwords that have very little meaning, it would either be necessary to re-motivate their importance in the introduction when you mention them, then refer to them throughout, or to skip them. That being said, I see the need for authors to use buzzwords like this to interest a large audience (who themselves do not necessarily understand the purpose of the concepts).

“This project paves the way for the establishment of an open and expandable electronic NP resource.”

By the end of the conclusion, there was no discussion of some of the drawbacks and roadblocks that might get in the way of the success of your initiative (or broadly how you would define success). I mentioned a few times already in this review places where there were meaningful discussions that were overlooked. This would be a good place to reinvestigate those.

Methods

I'm a huge opponent to the methods section coming after the results. There were a few places where the cross-linking worked well, but when reading the manuscript from top to bottom, I had many questions that made it difficult to proceed. The alternative might not be so good either, so I understand that this likely can't be addressed.

“Before their inclusion, the overall quality of the source was manually assessed to estimate, both, the quality of referenced structure-organism pairs and the lack of ambiguities in the links between data and references.”

How was quality assessed? Do you have a flowchart describing the process?

“Traditional Chinese Medicine (TCM) names, and conversion between digital reference identifiers, no solution existed, thus requiring the use of tailored dictionaries.”

How were these made? Where are the artifacts? Is it the same as the artifacts linked in the later section? Please cross link to them if so.

“All pre-translation steps are included in the preparing_name function and are available in src/r/preparing_name.R.”

For this whole section: why were no NER methods applied?

“When stereochemistry was not fully defined, (+) and (-) symbols were removed from names.”

Given the domain, this seems like a huge loss of information. Can you reflect on what the potential negative impacts of this will be? This also comes back to my other suggestion to better motivate the downstream work that uses databases like LOTUS – how could it impact one of those specific examples?

“After full resolution of canonical names, all obtained names were submitted to rotl (Michonneau et al., 2016)”.

missing reference.

Overall

Thanks for writing such a nice manuscript. I very much enjoyed reading it.

In direct conflict with all of the other things that could improve this manuscript, it's also quite long already. Because it reads easily, this wasn't too much of a problem, but use your best judgement if/when making updates.

*Reviewer #2 (Recommendations for the authors):*

Overall well written with few errors.

page 14:

"Targeted queries allowing to interrogate LOTUS data from the perspective of one of the three objects forming the referenced structure-organism pairs can be also built."

Should be "can also be built"

Some of the figures after page 68 appear to be corrupted.

*Reviewer #3 (Recommendations for the authors):*

Thank you for submitting your manuscript. Your approach to re-using an existing platform (e.g., Wikidata) in combination with carefully designed integration and harmonization workflows has great potential to help make better use of existing openly available datasets.

However, I was unable to complete my review due to my inability to reproduce (even partially) the described LOTUS workflows. I did, however, appreciate the attempt that the authors made to work towards meeting the *eLife* data access policies:

from https://submit.elifesciences.org/html/eLife_author_instructions.html#policies accessed on 5 Aug 2021 :

"[…] Availability of Data, Software, and Research Materials

Data, methods used in the analysis, and materials used to conduct the research must be clearly and precisely documented, and be maximally available to any researcher for purposes of reproducing the results or replicating the procedure.

Regardless of whether authors use original data or are reusing data available from public repositories, they must provide program code, scripts for statistical packages, and other documentation sufficient to allow an informed researcher to precisely reproduce all published results. […]"

I've included my review notes below for your consideration and I am looking forward to your comments.

re: "LOTUS employs a Single Source of Truth (SSOT, Single_source_of_truth) to ensure data reliability and continuous availability of the latest curated version of LOTUS data in both Wikidata and LNPN (Figure 1, stage 4). The SSOT approach consists of a PostgreSQL DB that structures links and data schemes such that every data element has a single place."

Single source of truth is advocated by authors, but not explicitly referenced in statements upserted into Wikidata. How can you trace the origin of LOTUS Wikidata statements to a specific version of the SSOT LOTUS postgres db?

re: Figure 1: Blueprint of the LOTUS initiative.

Data cleaning is a subjective statement: what may be "clean" data to some, may be considered incomplete or incorrectly transformed by others. Suggest to use less subjective statement like "process", "filter", "transform", or "translate".

re: "The contacts of the electronic NP resources not explicitly licensed as open were individually reached for permission to access and reuse data. A detailed list of data sources and related information is available as SI-1."

The provided dataset references provide no mechanism for verifying that the referenced data was in fact the data used in the LOTUS initiative. A method is provided for versioning the various data products across different stages (e.g., external, interim, processed validation), but README was insufficient to access these data (for example see below).

re: "All necessary scripts for data gathering and harmonization can be found in the lotus-processor repository in the src/1_gathering directory. All subsequent and future iterations that include additional data sources, either updated information from the same data sources or new data, will involve a comparison of the new with previously gathered data at the SSOT level to ensure that the data is only curated once."

re: "These tests allow a continuous revalidation of any change made to the code, ensuring that corrected errors will not reappear."

How do you continuously validate the code? Periodically, or only when changes occur? In my experience, claiming continuous validation and testing is supported by active continuous integration / automated testing loops. However, I was unable to find links to continuous testing logs on platforms like travis-ci.org, github actions, or similar.

re: "LOTUS uses Wikidata as a repository for referenced structure-organism pairs, as this allows documented research data to be integrated with a large, pre-existing and extensible body of chemical and biological knowledge. The dynamic nature of Wikidata fosters the continuous curation of deposited data through the user community. Independence from individual and institutional funding represents another major advantage of Wikidata."

How is Wikidata independent from individual and institutional funding? Assuming that it takes major funding to keep Wikidata up and running, what other funding source is used by Wikidata beyond individual donations and institutional grants/support?

re: "The openness of Wikidata also o ers unprecedented opportunities for community curation, which will support, if not guarantee, a dynamic and evolving data repository."

Can you provide example in which community curation happened? Did any non-LOTUS contributor curate data provided by LOTUS initiative?

"As the taxonomy of living organisms is a complex and constantly evolving eld, all the taxon identi ers from all accepted taxonomic DB for a given taxon name were kept. Initiatives such as the Open Tree of Life (OTL) (Rees and Cranston, 2017) will help to gradually reduce these discrepancies, the Wikidata platform should support such developments."

Discrepancies, conflicts, taxonomic revisions, and disagreements are common between taxonomies due to scientific differences, misaligned update schedules, and transcription errors. However, Wikidata does not support organized dissent, instead forcing a single (artificial) taxonomic view onto commonly used names. In my mind, this oversimplifies taxonomic realities, favors one-size-fits-none taxonomies, and leaves behind the specialized taxonomic authorities that are less tech savvy, but highly accurate in their systematics. How do you imagine dealing with conflicting, or outdated, taxonomic interpretations related to published names?

"The currently favored approach to add new data to LOTUS is to edit Wikidata entries directly. Newly edited data will then be imported into the SSOT repository. There are several ways to interact with Wikidata which depend on the technical skills of the user and the volume of data to be imported/modi ed."

How do you deal with edit conflicts? E.g., source data is updated around the same time as an expert updates the compound-organism-reference triple through Wikidata. Which edit remains?

"Even if correct at a given time point, scientic advances can invalidate or update previously uploaded data. Thus, the possibility to continuously edit the data is desirable and guarantees data quality and sustainability. Community-maintained knowledge bases such as Wikidata encourage such a process."

On using LOTUS in scientific publications, how can you retrieve the exact version of Wikidata or LOTUS resource referenced in a specific publication? In my understanding, specific Wikidata versions are not easy to access and discarded periodically. Please provide an example of how you imagine LOTUS users citing claims provided by a specific version of LOTUS SSOT to Wikidata.

"The LOTUS initiative provides a framework for rigorous review and incorporation of new records and already presents a valuable overview of the distribution of NP occurrences studied to date."

How can a non-LOTUS contributor dissent with a claim made by LOTUS, and make sure that their dissent in recorded in Wikidata or other available platforms? Some claims are expected to be disputed and resolved over time. Please provide an example of how you imagine dealing with unresolved disputes.

"Community participation is the most e cient means of achieving more comprehensive documentation of NP occurrences, and the comprehensive editing opportunities provided within LOTUS and through the associated Wikidata distribution platform open new opportunities for collaborative engagement."

Do you have any evidence to suggest that community participation is the most effective way to achieve more comprehensive documentation of NP occurrences? Please provide references or data to support this claim.

"In addition to facilitating the introduction of new data, it also provides a forum for critical review of existing data, as well as harmonization and veri cation of existing NP datasets as they come online."

Please provide example of cases in which a review led to a documented dispute that allowed two or more conflicting sources to co-exist. Show example of how to annotate disputed claims and claims with suggested corrections.

How do imagine using Wikidata to document evidence for a refuted "compound-found-in-organism claimed-by-reference" claim?

"Researchers worldwide uniformly acknowledge the limitations caused by the intrinsic unavailability of essential (raw) data (Bisson et al., 2016)."

Note that FAIR does not mandate open access of data, only suggests vague guidelines to find, access, integrate and re-use possibly resources that may or may not be access controlled.

If you'd like to emphasize open access in addition to FAIR, please separately mention Open Data / Open Access references.

re: "We believe that the LOTUS initiative has the potential to fuel a virtuous cycle of research habits and, as a result, contribute to a better understanding of Life and its chemistry."

LOTUS relies on the (hard) work of underlying data sources to capture the relationships of chemical compounds with organisms along with their citation reference. However, the data source are not credited in the Wikidata interface. How does *not* crediting your data sources contribute to a virtuous cycle of research habits?

re: Wikidata activity of NPImporterBot

Associated bot can be found at:

https://www.wikidata.org/wiki/User:NPImporterBot

On 5 Aug 2021, the page https://www.wikidata.org/wiki/Special:Contributions/NPImporterBot

most recent entries included a single edit from 4 June 2021

https://www.wikidata.org/w/index.php?title=Q107038883&oldid=1434969377

followed by many edits from 1 March 2021

e.g.,

https://www.wikidata.org/w/index.php?title=Q27139831&oldid=1373345585

with many entries added/upserted in early Dec 2020

with earliest entries from 28 Aug 2020

https://www.wikidata.org/w/index.php?title=User:NPImporterBot&oldid=1267066716

It appears that the bot has been inactive since June 2021. Is this expected? When does the bot become active?

re: "The Wikidata importer is available at https://gitlab.com/lotus7/lotus-wikidata-importer. This program takes the processed data resulting from the lotusProcessor subprocess as input and uploads it to Wikidata. It performs a SPARQL query to check which objects already exist. If needed, it creates the missing objects. It then updates the content of each object. Finally, it updates the chemical compound page with a "found in taxon" statement complemented with a "stated in" reference."

How does the Wikidata upserter take manual edits into account? Also, if an entry is manually deleted, how does the upserter know the statement was explicitly removed to make sure to now re-add a potentially erroneous entry.

In attempting to review and reproduce (parts of) workflow using provided documentation, https://gitlab.com/lotus7/lotus-processor/-/blob/7484cf6c1505542e493d3c27c33e9beebacfd63a/README.adoc#user-content-pull-the-repository was found to suggest to run:

git pull https://gitlab.unige.ch/Adriano.Rutz/opennaturalproductsdb.git

However, this command failed with error:

fatal: not a git repository (or any parent up to mount point /home)

Stopping at filesystem boundary (GIT_DISCOVERY_ACROSS_FILESYSTEM not set).

Assuming that the documentation meant to suggest to run:

git clone https://gitlab.com/lotus7/lotus-processor,

the subsequent command to retrieve all data also failed:

$ dvc pull

WARNING: No file hash info found for 'data/processed'. It won't be created.

WARNING: No file hash info found for 'data/interim'. It won't be created.

WARNING: No file hash info found for 'data/external'. It won't be created.

WARNING: No file hash info found for 'data/validation'. It won't be created.

4 files failed

ERROR: failed to pull data from the cloud – Checkout failed for following targets:

data/processed

data/interim

data/external

data/validation

Is your cache up to date?

Without a reliable way to retrieve the data products, I cannot review your work and verify your claims of reproducability and versioned data products.

Also, in https://gitlab.com/lotus7/lotus-processor/-/blob/7484cf6c1505542e493d3c27c33e9beebacfd63a/README.adoc#user-content-dataset-list, you claim that the dataset list was captured in xref:docs/dataset.tsv. However, no such file was found. Suspect a broken link with correct version pointing to xref:docs/dataset.csv.

[Editors' note: further revisions were suggested prior to acceptance, as described below.]

Thank you for resubmitting your work entitled "The LOTUS Initiative for Open Knowledge Management in Natural Products Research" for further consideration by *eLife*. Your revised article has been evaluated by Anna Akhmanova (Senior Editor) and a Reviewing Editor.

The manuscript has been improved but there are some remaining issues (particularly those of Reviewer 2) that need to be addressed, as outlined below:

*Reviewer #1 (Recommendations for the authors):*

I'm quite happy with the authors' responses to my previous review. Thank you for addressing mine and the other reviewers' points carefully.

*Reviewer #2 (Recommendations for the authors):*

Rutz et al. describe the LOTUS initiative, a database which contains over 750,000 referenced structure-organism pairs. Present both the data which they have made available in Wikidata, as well as their interactive web portal (https://lotus.naturalproducts.net/). provides a powerful platform for mining literature for published data on structure-organism pairs.

The clarity and completeness of the manuscript has improved significantly from the first round to the second round of reviews. Specifically, the authors clarified the scope of the study and provided concrete, and clear examples of how they think the tool should/could be used to advance np research.

I think the authors adequately addressed reviewer concerns regarding the documentation of the work and provide ample documentation on how the LOTUS initiative was built, as well as useful documentation on how it can be used by researchers.

Strengths:

The Lotus initiative is a completely open source, database built on the principles of FAIRness and TRUSTworthiness. Moreover, the authors have laid out their vision of how they hope LOTUS will evolve and grow.

The authors make a significant effort to consider the LOTUS end-user. This is evidenced by the clarity of their manuscript and the completeness of their documentation and tutorials for how to use, cite and add to the LOTUS database.

Weaknesses/Questions:

1) The authors largely addressed my primary concern about the previous version of the manuscript, which was that the scope and the completeness of the LOTUS initiative was not clearly defined. Moreover, by providing concrete examples of how the resource can be used both clarifies the scope of the project and increases the chances that it will be adopted by the great scientific community. In the previous round of reviews, I asked "if LOTUS represents a comprehensive database". In their response, the authors raise the important point that the database will always be limited by the available data in the literature. While I agree with the authors on this point and do not think it takes away from the value of the manuscript/initiative, I think a more nuanced discussion of this point is merited in the paper.

While the authors say that the example queries provided illustrate that the LOTUS can be used to answer research questions, i.e. how many compounds are found in Arabipsis. However, interpreting the results of the query are far more nuanced. For example, can one quantitively compare the number of compounds observed in different species, or families based on the database (e.g. do legumes produce more nitrogen containing compounds than other plant families)? Or do the inherent biases in literature inhibit our ability to draw conclusions such as this? I like that the authors showcase the potential value of using the outlined queries as a starting place for a systematic review, as they highlight in the text. However, I would like to see the authors discuss the inherent limitations with using/interpreting the database queries given the incompleteness of NP exploration.

2) On page (16?) the authors discuss how the exploration of NP is rather limited, citing the fact that each structure is found on average in only 3 organisms, and that only eleven structures per organism. How do we know that these low numbers are a result of incomplete literature or research efforts and not due to the biology of natural products? For example, plant secondary metabolites are extremely diverse and untargeted metabolomics studies have shown that any given compound is found in only a few species. There is likely no way of knowing the answer to this question, but this section could benefit from a more in-depth discussion.

3) I do not understand Figure 4. I would like to see more details in the figure legend and an explanation of what an "individual" represents? Are these individual publications? Individual structure-organism pairs? Also, what do the box plots represent? Are they connected to the specific query of Arbidosis thaliana/B-sitosterol? If they are there should only be a value returned for the number of structures in *Arabidopsis thaliana*, rather than a distribution. Thus, I assume that the accumulation curve and the box plots are unrelated, and this should be spelt out in the figure legend or separated into two different panels.

4) On page 5 the authors introduce/discuss COCONUT. I am confused about how exactly COCONUT is related to or different from LOTUS. Can you provide some more context, similar to how the other databases were discussed in the preceding paragraph?

---

## [Author Response]

Essential revisions:1) Reviewers would like to see the authors clarify the scope of their work and address to what extent this database will provide the community with a comprehensive database of structure organisms pairs, that is actually useable to answer research questions. This could be achieved by adding concrete examples of how these data could be used to help readers/reviewers understand the scope.

We thank the editors and reviewers for these suggestions. The scope of our work is dual: LOTUS was designed both to help the gathering and exploitation of past NP research output (structure-organism pairs) and to facilitate future formatting and reuse. LOTUS doesn’t only provide a wide set of curated and documented structure-organisms pairs, but also a set of *tools* to gather, organize, and interrogate them. The results of this first output of the LOTUS initiative can thus be exploited in multiple ways by a wide range of researchers. We agree with the reviewers that mentioning concrete application examples is of importance.

Previous to this work, efficient access to specialized metabolites occurrences information was a complicated task. Since we implemented LOTUS, through Wikidata, anyone can easily access high-quality information about natural products occurrences in different ways (which organisms contain a given chemical compound? which compounds are found in a given organism?).

For example, with a simple click and without programming skills, anyone can look up the metabolome of *Arabidopsis thaliana*, on Scholia (https://scholia.toolforge.org/taxon/Q158695). This page and specifically the “Metabolome” section yields a table with relevant information related to the chemical structures (Wikidata identifiers, SMILE codes, etc.) and can be easily downloadable. Sharing such public data in an open and correctly formatted form is the main scope of this first project of the LOTUS initiative and a first concrete step to engage the NP community in such a data-sharing model.

To concretely illustrate the scope of our work, we listed multiple concrete examples of how these data could help answer relevant research questions in Table 2, which we further updated (see below) in our manuscript (https://lotusnprod.github.io/lotus-manuscript/#tbl:queries). This table lists a selected set of SPARQL queries that offer the reader concrete illustrations to exploit the LOTUS data available at Wikidata. Following the reviewer’s suggestions, we adapted the manuscript to advertise this central table better and earlier in the manuscript (See https://github.com/lotusnprod/lotus-manuscript/commit/1456cc82d6cb7a79a57f1403a053f9f0348fb351). Furthermore, readers are invited to have a look at the project page (https://www.wikidata.org/wiki/Wikidata:WikiProject_Chemistry/Natural_products) on Wikidata to fetch the most recent and updated SPARQL queries examples. This project page is also indicated to ask for help or suggest improvements.

To further showcase the possibilities opened by LOTUS, and also answer the remark on the comprehensiveness of our resource, we established two additional queries (https://w.wiki/4VGC and https://w.wiki/4VGC). Both queries were inspired by recent literature review works (https://doi.org/10.1016/j.micres.2021.126708 and https://doi.org/10.1016/j.phytochem.2021.113011). The first work describes compounds found in Actinobacteria, with a focus on compounds with reported bioactivity. The second one describes compounds found in *Aspergillus* spp., with a chemical focus on terpenoids. In both cases, within seconds, the queries allow retrieving a table similar to the ones in the mentioned literature reviews. Such queries are not meant to fully replace the valuable work behind the establishment of such review papers, however, they now offer a solid basis to start such works and liberate precious time to analyze and discuss the results of these queries.

Overall, the LOTUS initiative aims to support the NP community’s information retrieval and to free researchers' valuable time for tasks that cannot be automated.

In the manuscript, we precisely outlined our vision of the overall scope of the LOTUS initiative both with concrete application cases and more generic objectives (see https://github.com/lotusnprod/lotus-manuscript/commit/67c4fbdae130b737dfc6acc3da0b6099d5ac2fb9).

With this, we hope that the scope of this first project of the LOTUS Initiative has been clarified and properly exemplified and we remain at your disposal for any further illustrations.

2) Reviewers think the manuscript provides missing or incomplete documentation on the LOTUS processes and software, especially in terms of reproducibility of results. Providing open access to source code and data is an important first step, and additional (challenging!) steps are needed to help others independently build, and use, the provided software/data.

Reviewers were right. At the initial submission stage, multiple parts of the LOTUS pipeline were not fully accessible and/or poorly documented.

Taking this into account, we adapted and reformatted multiple parts of our different repositories.

First, we kept DVC (a data management tool – https://dvc.org) for our internal use only and added an option to programmatically access easily fetchable external data sources, for anyone to access the input data for workflow reproducibility and testing purposes.

Second, and since the full reproduction of the entire LOTUS processing workflow is time-consuming (> days), we added a minimal working example (https://github.com/lotusnprod/lotus-processor/blob/main/tests/tests_min.tsv) sampled from the original data to illustrate how the process works and anyone to be able to reproduce results from this subset.

Third, as suggested, we moved all repositories to a single place, on GitHub (https://github.com/lotusnprod). The code repositories now contain improved technical documentation, important steps periodically built by Continuous Integration, and additional features to improve user experience (see below). While previously focused on Linux systems only, we extended the portability of our workflow to macOS and Windows under WSL (https://github.com/lotusnprod/lotus-processor/actions/runs/1450330111). This should help to reach a broader audience.

Moreover, a “manual” mode has been implemented for each researcher to be able to upload its own documented structure-organism pairs to Wikidata. See instructions here https://github.com/lotusnprod/lotus-processor/wiki/Processing-your-own-documented-structure-organism-pairs.

As we built our tool for the NP community, additional requests will likely appear once more users start using it. The GitHub issue tracker allows collecting in a single place users’ problems and suggestions. We do not guarantee the absence of hiccups and assure the editors that we did our best for optimal reproducibility following the reviewer’s comment. Again, we are happy to improve and open to additional suggestions.

3) The authors have yet to provide compelling evidence of how their continuously managed (and updated) resources can be reliably cited in scholarly literature and incorporated in scientific workflows. In order to study and reference Wikidata, a versioned copy needs to be provided: Wikidata is updated constantly, and these constant stream of changes make it hard for others to verify results extracted from some older version of the Wikidata corpus unless a versioned copy is provided.

This is indeed a very pertinent point. Wikidata being a dynamic environment, the versioning aspects imply some challenges. However, versioning and tracking of the data dynamics can be achieved in multiple ways and at different levels:

Wikidata level:

The first way to access LOTUS data at a given time point is using the Wikidata images. Indeed, regular dumps (versions) of Wikidata exist and are available here https://dumps.wikimedia.org/wikidatawiki/entities. Querying them is indeed challenging and not the most convenient, this being a direct consequence of the amount of information they hold. However, these are regular snapshots that include the totality of the LOTUS data present on Wikidata. These regular dumps are realized independently of the LOTUS initiative members and thus offer increased durability concerning versioning.

LOTUS level:

A way to get around the problems posed by the amount of data in the versioning strategy detailed above (full Wikidata dumps) is to subset the Wikidata for entries of interest only. In our case the structures, biological organisms, bibliographic references, and the links among them. This can be done by the interested researchers using the https://github.com/lotusnprod/lotus-wikidata-interact bot.

We will also regularly upload results of a global SPARQL query on the LOTUS Initiative Zenodo Community repository (https://zenodo.org/communities/the-lotus-initiative) using this bot or future version of this bot on such datasets https://zenodo.org/record/5668855.

SPARQL level:

When using Wikidata as a source for LOTUS information, we invite users to share their query using the short URL and archive its results on a data repository such as Zenodo, OSF, or similar used and share it when releasing associated results. This will greatly help moving to better reproducible research outputs.

For example, the output of this Wikidata SPARQL query https://w.wiki/4N8G realized on the 2021-11-10T16:56 can be easily archived and shared in a publication https://zenodo.org/record/5668380*.*

The interested user can directly upload the results of their SPARQL queries on our LOTUS Initiative Community repository using the following link: https://zenodo.org/deposit/new?c=the-lotus-initiative.

Individual entry level:

At a more precise scale, each Wikidata entry has a permanent history. Therefore, versioning of single queries is automatically done for each entry of the LOTUS data and can be accessed by everyone. See here for the full history of erythromycin https://www.wikidata.org/w/index.php?title=Q213511&action=history.

To summarize, given its Wikidata-based hosting, LOTUS data is expected to evolve. While versioning is not impossible, it requires different means than the one used for more classical and static databases. It is to be noted that each of the queries of the researchers can be easily versioned and shared with the community and that the individual history of each of LOTUS central objects (chemical structures, biological organisms, and references) is kept and fully accessible on the history page of each Wikidata item entry. Altogether, we hope that the means proposed above provide good ways of tracking and versioning LOTUS entries.

Some versioning mechanism propositions were added to the manuscript: see https://github.com/lotusnprod/lotus-manuscript/commit/92833166375aa9cc29f44cffdd5ad693fa42934c

Reviewer #1 (Recommendations for the authors):“A third fundamental element of a structure-organism pair is a reference to the experimental evidence that establishes the linkages between a chemical structure and a biologicl organism and a future-oriented electronic NP resource should contain only fully-referenced structure-organism pairs.”Typo in "biologicl"

We thank the reviewer for carefully reading, and we corrected the typo.

“Currently, no open, cross-kingdom, comprehensive, computer-interpretable electronic NP resource links NP and their producing organisms, along with referral to the underlying experimental work”.missing "that"

We couldn’t find the missing “that” in this sentence.

"KNApSAck currently contains 50,000+ structures and 100,000+ structure-organism pairs. However, the organism field is not standardized and access to the data is not straightforward".This is the first opportunity in the manuscript to describe in more detail the perils of previous databases, especially the mess that is KNApSAck, which you have no choice but to work on because of its ubiquity.

As answered to Reviewer 1 – question N° 3, it was a deliberate choice not to emphasize the perils/weaknesses of currently available DBs, as we use them, but to focus on better habits for the future. We didn’t want to start devaluating resources or elevate some at the cost of others.

“NAPRALERT is not an open platform, employing an access model that provides only limited free searches of the dataset”.There's an awful lot of praise for this database given that it is directly antithetical to the manuscript. Please provide further commentary contrasting the work in NAPRALERT (particularly, about its shortcomings as a closed resource) to LOTUS.

NAPRALERT maintenance and information input were greatly reduced in the last decade. And as such, its limitations are showing even more. But it proved to be a valuable resource for LOTUS. Its current maintainers, which all contributed to LOTUS and this manuscript decided to donate all its chemical-related data to the project so it could be made available for all. The quality of chemical annotations in NAPRALERT is rather poor as it doesn't have direct structures or structural information but mainly chemical names requiring extensive use of chemical translators (a quite error-prone process see the answer to question 9.) However, the bibliographical data is of much higher quality than other resources and the organism data as well, despite many of the names have changed with recent taxonomic revisions. Overall, and as we answered the previous question, our choice in this manuscript was not to compare or list shortcomings of existing data sources but rather propose new directions for the future.

“FAIR and TRUST (Transparency, Responsibility, User focus, Sustainability and Technology) principles”.If you really want to go down the buzzword bingo, I'd suggest making a table or going in depth into each point. These acronyms are, in my opinion, effectively meaningless from a technical perspective, so it falls on the authors of the paper who want to use them (as you are no doubt pressured to do in the modern publishing landscape) to define them and qualify their relevance to your more practical goals.

The categorization of these acronyms as meaningless buzzwords or valuable and ideal guidelines to be met is indeed, and happily, the responsibility and liberty of each researcher. As long as these terms will be considered as buzzwords and not guidelines there will be work to do for the advancement of open research. After deliberation among the authors, we believed that because LOTUS checks both boxes of FAIR (https://en.wikipedia.org/wiki/FAIR_data) and TRUST (https://www.nature.com/articles/s41597-020-0486-7), we would rather stick to these two terms. As stated in the TRUST white paper “However, to make data FAIR whilst preserving them over time requires trustworthy digital repositories (TDRs) with sustainable governance and organizational frameworks, reliable infrastructure, and comprehensive policies supporting community-agreed practices.” We strongly believe that Wikidata is such a repository. Regarding both terms, we felt that redefining alternative ones would not help the reader/user.

“…any researcher to contribute, edit and reuse the data with a clear and open CC0 license (Creative Commons 0).”This is a really interesting point considering you have taken information from several unlicensed databases and several that have more permissive licenses. How do you justify this?

We thank the reviewer for this comment, it is indeed central to the work. These are our justifications:

– First, we never strictly copied a DB. We carefully took relevant pieces of information as allowed per the *Right to quote*.

– Second, there was significant work of harmonization and curation including manual validation steps. This alone should justify the possibility of dissemination under another license.

– Finally, we contacted all authors of DBs whose licensing status was unclear in our view, showing goodwill, and will retract any data on request.

In the end, who could, in any ethical consistency, legally own the fact that a given natural product is present in a given organism?

“The SSOT approach consists of a PostgreSQL DB that structures links and data schemes such that every data element has a single place”.Why is Wikidata not the single source of truth? This means that the curation and generation of this dataset can never really be decentralized- it will always have to be maintained by someone who is the maintainer of the PostgreSQL database. What are the pros/cons for this?

We thank the reviewer for bringing up this crucial point. Here, there are multiple elements to take into consideration:

First, Wikidata does not currently allow sharing properties of interest for the NP community (e.g. structural depiction, spectra, etc.). Having them in a specialized DB for the moment can help. If the community is happy with the offered resource, it is much more likely it will contribute to the project and push towards better, accepted, Wikidata-compatible standards.

Second, even if sad, intentional, or unintentional vandalism exists on Wikidata. Having a decoupled PostgreSQL can therefore be seen as additional security.

Of course, this makes the maintainability of the minimal “non-Wikidata” part of LOTUS as weak as the current criticized system, but the very large part being already on Wikidata, we consider it as a mere backup system We further consider it as a necessary transition step for the community to adhere to our initiative.

Now that the initial release of the LOTUS dataset is on Wikidata it is indeed decentralized as data addition/correction/deletion can come from any of the Wikidata users. We will be happy to retire the non-Wikidata part of LOTUS as soon as it will not serve any purpose anymore.

“The LOTUS processing pipeline is tailored to efficiently include and diffuse novel or curated data directly from new sources or at the Wikidata level.”This sentence is confusing

We thank the reviewer and correct the sentence as follows: “The LOTUS processing pipeline is tailored to efficiently include and diffuse novel curated data from new sources. These sources can be new open external resources or additions made directly to Wikidata.”

“All stages of the workflow are described on the git sites of the LOTUS initiative at https://gitlab.com/lotus7 and https://github.com/mSorok/LOTUSweb”Later this became a problem for code review. Why are the code all in different places? There are "organizations" both on GitLab and GitHub to keep related code together. Further, putting code in a personal user's namespace makes it difficult to for potential community involvement, especially if the user becomes inactive. Also a matter of opinion, but most science is being done on GitHub. Using GitLab will likely limit the number of people who will interact with the repository. Highly suggested to move it to GitHub.

This is a good remark. Code was initially split because of work repartition among the team. We tried to centralize everything on GitLab, which had our favor at the time. Taking the reviewer considerations into account, we moved all repositories to GitHub under the lotusnprod organization https://github.com/lotusnprod. As we want users to interact as much as possible with LOTUS, if this can increase, as the reviewer suggests, the number of interactions, we will be happy.

“All necessary scripts for data gathering and harmonization can be found in the lotus-processor repository in the src/1_gathering directory."The reference to "SI 1 Data Sources List" does not include the actual license information in the table, but rather links to READMEs (which may not stay stable for the life of the manuscript). This should explicitly state which license each database has. Further, it seems a bit disingenuous since some of these links point to README pages that state there is no license given, such as for Biofacquim (and several others)
https://gitlab.com/lotus7/lotus-processor/-/blob/main/docs/licenses/biofacquim.md
How can you justify taking this content and redistributing it under a more permissive license?

We thank the reviewer for this comment and updated the table accordingly. Having all this information is challenging. We are sorry for the questionable format of the table but strongly disagree with “disingenuous”. All searches were made ingenuously. Regarding the biofacquim example, it is a really good one since multiple versions were published along the road of the manuscript. As mentioned in the license.md file (“*But the [article](https://www.mdpi.com/2218-273X/9/1/31) is under [Creative Commons Attribution License] (https://creativecommons.org/licenses/by/4.0/)*”), MDPI clearly states:

“No special permission is required to reuse all or part of article published by MDPI, including figures and tables. For articles published under an open access Creative Common CC BY license, any part of the article may be reused without permission provided that the original article is clearly cited. Reuse of an article does not imply endorsement by the authors or MDPI.“

We cited the original article clearly and we do not see any restriction issue since the distributed content is not copied from Biofacquim but parts of it that have undergone significant processing steps.

Further, this table should contain versioning information for each database and a flag as to whether it is still being maintained, whether data can be accessed as a dump, and if there is a dump, if it is structured. Right now, the retrieval column is not sufficient for describing how the data is actually procured.

We added the requested flag. The table (https://github.com/lotusnprod/lotus-processor/blob/main/docs/dataset.csv) now contains a boolean “dump” column, together with a column defining the status of the resource (maintained, unmaintained, retired or defunct), and a column with the timestamp of the last input modification. As the majority of the resources do not provide versioning, the timestamp corresponds to when the query was manually performed last. The input might of course have changed in the meantime however the git history of this table will allow tracking these evolutions.

Can you cross-reference these databases to Wikidata pages? FAIRSharing pages? I'm sure following publication in the near- or mid-term, more of these databases will go down permanently, so there should be as much information about what they were available with this publication.

After deliberation, we decided to link the structure-organisms pairs on Wikidata pages only to the original experimental work documenting the occurrence (the scientific publication). The goal of this LOTUS initiative project was to move away from the classical natural products databases system. Just like we don’t refer to LOTUS for a documented pair, we don’t refer to any other DB (what is the scientific interest of listing the x DB documenting quercetin in Quercus sp ?) but simply the original experimental work.

Another interesting point raised is the fact that the resources might go down, permanently or not. Since the beginning of our initiative, many of them are not openly accessible anymore (wakankensaku, mitishamba), or their licensing status changed (alkamid, tcmid translation files). This is an additional reason to refer to the original work directly.

Small suggestion: right align all numbers in tables.

We appreciate the suggestion, we right-aligned the numbers in tables.

“This process allowed us to establish rules for automatic filtering and validation of the entries. The filtering was then applied to all entries.”Please explain this process, in detail (and also link to the exact code that does it as you have done in other sections)

The process in detail is described in the methods section. We added the link to the exact code and a sentence linking the results to the corresponding method.

“Table 1: Example of a referenced structure-organism pair before and after curation”.Would like some discussion of the importance of anatomical region in addition to organism. Obviously, there is importance in the production of the NP in a given region. This is highly granular and makes the problem more difficult, but even if you don't tackle it, it has to be mentioned.

The discussion opened here is of great interest, but probably out of scope for the paper. We aim to establish the basis for better reporting, not solving the problem until the finest granulometry. This could lead to endless discussions, not only about organs but also open the door to many other topics. Saying “quercetin found in Quercus sp.” is anyway a proxy. It was not found in all *Quercus* sp. but in a specific extract of a specific organism (if lucky with a voucher ID), growing in a specific location, with specific endophytes (probably responsible for the production of many compounds, for an estimation: 10.1186/s40793-021-00375-0), specific climate, etc. This also explains why we deliberately chose the found in taxon property (https://www.wikidata.org/wiki/Property:P703) and not the natural product of taxon one (https://www.wikidata.org/wiki/Property:P1582). While those aspects certainly are very interesting challenges for the coming years, we cannot address such precision levels in the current work.

Further, why is the organism not standardized to a database identifier? Why are the prefixes to go along with these ID spaces not mentioned (InChI-key for structure, DOI for reference, and what for organism?) Maybe one of the confounders is that there are many taxonomical databases for organisms with varying coverage – this should be addressed. On second pass through the review, I noticed this was described a bit more in the methods section, so this should be linked. Further, it's actually not clear even by the end of the methods section what identifiers are used in the end. Are Wikidata Q identifiers the single unifying identifier in the end?

Indeed, taxa are more complex to handle than inchikeys and dois. As mentioned, all taxon names are linked to (1) a taxonomic database (2) the ID corresponding to this DB. And because of the known disagreements between taxonomic DBs, this pair constitutes the “ID”. A taxon name could correspond to different taxa in different DBs. As Wikidata still uses taxa names as IDs to link them to taxonomic DBs, we did the same and uploaded each identifier we found through GNames. So in the end, the 3 identifiers used to decide whether to create or update a QID are inchikey, doi, taxon name (accepted in a taxonomic db accepted within Wikidata taxonomy community). As the reviewer states at the end of its comment, Wikidata id of a taxon name can indeed be seen as efficient means to identify a biological organism as Wikidata in this case acts as a taxonomical aggregator and allows to moderate disagreements between taxonomists (who often disagree …). See taxa graph on the scholia page of Lotus halophilus for example https://scholia.toolforge.org/taxon/Q15435646.

“Figure 2: Alluvial plot of the data transformation flow within LOTUS during the automated curation and validation processes.”Looks like more things are rejected than kept. What about having a human curator in the loop? Or should poorly curated stuff be thrown way forever? There was a small mention about building a human curation system, but I don't think this had the same focus.

Indeed, we favored quality over quantity, as highlighted by the F-scores obtained after automated curation. As mentioned, human curation will take place later on, entries are not thrown away forever but are not of a sufficient quality to be sent as-is. We adapted the text accordingly.

A curation interface is actually in development, with access to part of the SSOT, and technical details for access, logging, etc. are still in discussion. This curation interface will, as mentioned, allow us to upload many entries we did not upload at the moment. Following the reviewer question, we added the non-uploaded part of our processed entries to Zenodo under the following link: 10.5281/zenodo.5794596.

“The figure highlights, for example, the essential contribution of the DOI category of references contained in NAPRALERT towards the current set of validated references in LOTUS”.Is this to say LOTUS is constructed with the closed data in NAPRALERT? Or the small portion that was made open was sufficient?

As previously mentioned, part of the closed data of NAPRALERT was donated to LOTUS. This part only consists of the minimally needed triplet: structure-organism-reference. The sentence was written to reinforce the fact that identifiers, such as DOI, lead to better results than other types of data. We removed the sentence concerning NAPRALERT and adapted the text.

“…any researcher has a means of providing new data for LOTUS, keeping in mind the inevitable delay between data addition and subsequent inclusion into LOTUS.”How likely do you think users will be to contribute their data to LOTUS, when there are many competing interests, like publishing their own database on their own website, which will support their own publication (and then likely go into obscurity)?Ultimately, this means there should be a long-term commitment from the LOTUS initiative to continue to find new databases as they're published, to do outreach to authors (if possible/necessary), and to incorporate the new databases into the LOTUS pipeline as a third party. This should be described in detail – which organizations will do this/how will they fund it?

We thank the reviewer for pointing this out. The sentence was indeed wrongly formulated. What the authors wanted to express was “new data for the community, via Wikidata (for LOTUS)”. As written in the text, the NP community suffers from sub-optimal data-sharing habits and the authors would like to participate in positive changes on these aspects. Without the help of publishers, it might take some time, but spectral repositories clearly show it is feasible. We are convinced users and maintainers will adopt this change, as it also clearly lowers the fundings needed.

“For this, biological organisms were grouped into four artificial taxonomic levels (plants, fungi, animals and bacteria).”How? Be more specific about methodology including code/dependencies

An additional sentence linking to the related method has been added. More details are also given in the main text.

“Figure 6: TMAP visualizations of the chemical diversity present in LOTUS. Each dot”.This color scheme is hard to interpret. Please consider using figure 6 in https://www.nature.com/articles/s41467-020-19160-7 to help pick a better color scheme.

We thank the reviewer for this suggestion. We tried our best with color palettes on all figures, some of them being challenging. While mainly aware of categorical color-blind friendly color palettes, we did not know the batlow palette. We implemented it and revised all TMAPs coloring.

“The biological specificity score at a given taxonomic level for a given chemical class is calculated as the number of structure-organism pairs within the taxon where the chemical class occurs the most, divided by the total number of pairs.”Have you considered the Kullback-Leibler divergence (https://en.wikipedia.org/wiki/Kullback%E2%80%93Leibler_divergence) which is often used in text mining to compare the likelihood of a query in a given subset of a full corpus? I think this would also be appropriate for this setting.

We thank the reviewer very much for this suggestion. After having looked at Kullback-Leibler divergence and its limitations, we found it more appropriate to use Jensen-Shannon divergence as it is both symmetric and bounded. We compared this divergence to our initial heuristic equation and chose to implement it.

“This specificity score was calculated as in Equation 2:”This is a very similar look to the classic Jaccard Index (https://en.wikipedia.org/wiki/Jaccard_index; which is skewed by large differences in the size of the two sets) and the overlap coefficient (https://en.wikipedia.org/wiki/Overlap_coefficient). I think the multiplication of the sizes of the sets will have the same issue as Jaccard index. What is the justification for using this non-standard formulation? Please comment if the conclusions change from using the overlap coefficient.

As for the previous equation, we thank the reviewer for pointing this out. Again, our equation was purely heuristic and its initial form was:

Structuresinchemicalclass∩StructuresintaxonStructuresinchemicalclass(chemicalpart) ∗Structuresinchemicalclass∩StructuresintaxonStructuresintaxon(taxonpart)

We did not realize it was so close to a Jaccard Index, so we did compute both the Jaccard index and the overlap coefficient.

**Author response image 1. sa2fig1:** Left heuristic vs right Overlap index.

While Jaccard is a good alternative to our initial score, with an interesting increase in small values, the overlap index flattens differences too much in our view, as shown in Figure This is due to the min() in the equation, given classes being too small. In all three investigated metrics, even if the height of the bars changes, the color pattern on the biological tree remains almost unchanged.

Since this is supporting Figure 7, a new way to explore the chemical and biological diversity, and you previously wrote the huge skew in organisms and taxa over which information is available, it would be important to comment on the possible bias in this figure. Were you trying to make the point that the NPClassifier system helps reduce this bias? If so, it was unclear.

We thank the reviewer for pointing out this unclarity. Our goal was not to make the point that NPClassifier helps to reduce this bias. We favored it over Classyfire simply because it was tailored to describe NPs and thus has the most interesting classes when describing the chemistry of Life. We adapted the text.

With the mentioned limitations concerning studied organisms, LOTUS still appears as the most comprehensive resource to display such chemical and biological diversity.

“As LOTUS follows the guidelines of FAIRness and TRUSTworthiness, all researchers across disciplines can benefit from this opportunity, whether the interest is in ecology and evolution, chemical ecology, drug discovery, biosynthesis pathway elucidation, chemotaxonomy, or other research fields that connect with NP.”This was emphatically not qualified within the manuscript. The expansions for FAIR and TRUST were given, but with no explanation for what they are. Given that these are buzzwords that have very little meaning, it would either be necessary to re-motivate their importance in the introduction when you mention them, then refer to them throughout, or to skip them. That being said, I see the need for authors to use buzzwords like this to interest a large audience (who themselves do not necessarily understand the purpose of the concepts).

We addressed this point in a previous remark, it should be clearer now.

“This project paves the way for the establishment of an open and expandable electronic NP resource.”By the end of the conclusion, there was no discussion of some of the drawbacks and roadblocks that might get in the way of the success of your initiative (or broadly how you would define success). I mentioned a few times already in this review places where there were meaningful discussions that were overlooked. This would be a good place to reinvestigate those.

Thanks for the suggestion, we updated the conclusion part of the manuscript with a shortcomings and challenges section. See https://github.com/lotusnprod/lotus-manuscript/commit/a866a01bad10dfd8b3af90e2f30bb3ae51dd7b9e.

MethodsI'm a huge opponent to the methods section coming after the results. There were a few places where the cross-linking worked well, but when reading the manuscript from top to bottom, I had many questions that made it difficult to proceed. The alternative might not be so good either, so I understand that this likely can't be addressed.

Thank you for your suggestions here. We thought about the remodeling implied by shifting the methods section and decided to conserve the document structure as is. We understand it is not optimal. We have, however, added crosslinks between results and methods sections.

“Before their inclusion, the overall quality of the source was manually assessed to estimate, both, the quality of referenced structure-organism pairs and the lack of ambiguities in the links between data and references.”How was quality assessed? Do you have a flowchart describing the process?

This question is highly interesting. The quality assessment relies mainly on human experience. Basically, some random documented pairs were chosen and their plausibility evaluated. Even if some projects (https://doi.org/10.33774/chemrxiv-2021-gxjgc-v2) aim to automatically link biological sources to chemical structures, it is currently difficult to automate.

Where, for trained NP chemists, it might be easier to quickly discard some DBs reporting aberrant structures in some organisms. Intentionally, we have not focused on the rejected in order not to "point the finger" at them too much, as this is not the primary aim of our study. Regarding the second part, some DBs simply document all pairs to all references, in a way it is almost impossible to find the reference documenting the pair. This also needs human evaluation before computational rules.

“Traditional Chinese Medicine (TCM) names, and conversion between digital reference identifiers, no solution existed, thus requiring the use of tailored dictionaries.”How were these made? Where are the artifacts? Is it the same as the artifacts linked in the later section? Please cross link to them if so.

Yes, they are the ones mentioned later in the methods. We added a linking sentence.

“All pre-translation steps are included in the preparing_name function and are available in src/r/preparing_name.R.”For this whole section: why were no NER methods applied?

This section and other ones would highly benefit from NER. However, NER would have to be applied to the original publications, and not on already preformatted fields, which make NER less powerful. We believe some work has to be done on the original articles but this implies other (also legal access) restrictions. Projects such as DECIMER (https://doi.org/10.1186/s13321-020-00469-w) will greatly help in this direction.

“When stereochemistry was not fully defined, (+) and (-) symbols were removed from names.”Given the domain, this seems like a huge loss of information. Can you reflect on what the potential negative impacts of this will be? This also comes back to my other suggestion to better motivate the downstream work that uses databases like LOTUS – how could it impact one of those specific examples?

It is actually no loss of information, but simply a curation process of the structure name field. This was carried to avoid non-existing information from being invented. Maybe the sentence was not clear, but, sometimes, with an isomeric SMILES, the generated name contained (+)/(-), which is not true. To be on the safe side, and not attribute a random stereochemistry, we removed it.

Many structures reported with fully defined stereochemistry in the original articles were then sadly degraded when imported into the source DBs of this work. This is an additional reason we want to push towards direct reporting of the pairs from the article to Wikidata. So, many of the structures we added have no stereochemistry information while it might exist but we opted for the safe solution. In our view, the negative impact would have been to try doing the opposite.

Having non-fully defined structures is already of great interest to many communities, such as the Mass Spec community, where, for example, the dereplication of structures can be carried without stereochemistry.

“After full resolution of canonical names, all obtained names were submitted to rotl (Michonneau et al., 2016)”.missing reference

We thank the reviewer for carefully reading, we added the correct reference.

Reviewer #2 (Recommendations for the authors):Overall well written with few errors.page 14:"Targeted queries allowing to interrogate LOTUS data from the perspective of one of the three objects forming the referenced structure-organism pairs can be also built."Should be "can also be built"

We thank the reviewer for careful reading and corrected the sentence

Some of the figures after page 68 appear to be corrupted.

We saw it and contacted the *eLife* editorial office about it. It might be an internal error only, all figures being displayed well in https://lotusnprod.github.io/lotus-manuscript.

Reviewer #3 (Recommendations for the authors):[…]re: "LOTUS employs a Single Source of Truth (SSOT, Single_source_of_truth) to ensure data reliability and continuous availability of the latest curated version of LOTUS data in both Wikidata and LNPN (Figure 1, stage 4). The SSOT approach consists of a PostgreSQL DB that structures links and data schemes such that every data element has a single place."Single source of truth is advocated by authors, but not explicitly referenced in statements upserted into Wikidata. How can you trace the origin of LOTUS Wikidata statements to a specific version of the SSOT LOTUS postgres db?

Concerning the Wikidata statements, we clearly want to avoid any attribution to sources DB (LOTUS or another one). The documentation of the structure-organism must be the original publication, without intermediate. While it is true that, actually, a lot of information is retrieved through specialized databases, we argue this practice should stop and move towards what we are suggesting, direct access to the original work on Wikidata. If, in the future data regarding structure-organisms pairs will be directly edited/uploaded to Wikidata, this whole issue would be solved. Regarding the SSOT, it is versioned internally, and we match each Wikidata statement through its 3 identifiers: organism name, inchikey and doi.

re: Figure 1: Blueprint of the LOTUS initiative.Data cleaning is a subjective statement: what may be "clean" data to some, may be considered incomplete or incorrectly transformed by others. Suggest to use less subjective statement like "process", "filter", "transform", or "translate".

We thank the reviewer for the suggestion. We adopted a more neutral wording. We adapted the figure, the text and the repository architecture, replacing all “clean-” with “process-”.

re: "The contacts of the electronic NP resources not explicitly licensed as open were individually reached for permission to access and reuse data. A detailed list of data sources and related information is available as SI-1."The provided dataset references provide no mechanism for verifying that the referenced data was in fact the data used in the LOTUS initiative. A method is provided for versioning the various data products across different stages (e.g., external, interim, processed validation), but README was insufficient to access these data (for example see below).

While we agree such a mechanism would be helpful, the first part of the remark applies to all published science, not only to our initiative. As previously explained , we do not want to redistribute original content. We made all our distributable data accessible, and the totality of our processing pipeline, even for not distributed data accessible.

We improved the documentation and data access in order to facilitate reproducibility of all steps that can be.

re: "All necessary scripts for data gathering and harmonization can be found in the lotus-processor repository in the src/1_gathering directory. All subsequent and future iterations that include additional data sources, either updated information from the same data sources or new data, will involve a comparison of the new with previously gathered data at the SSOT level to ensure that the data is only curated once."re: "These tests allow a continuous revalidation of any change made to the code, ensuring that corrected errors will not reappear."How do you continuously validate the code? Periodically, or only when changes occur? In my experience, claiming continuous validation and testing is supported by active continuous integration / automated testing loops. However, I was unable to find links to continuous testing logs on platforms like travis-ci.org, github actions, or similar.

Code is continuously validated each time changes are pushed to the repository through the CI/CD. It was previously implemented in gitlab (https://gitlab.com/lotus7/lotus-processor/-/pipelines). Since we moved the repository to GitHub as requested by reviewer 1, we made a new CI pipeline available at: https://github.com/lotusnprod/lotus-processor/actions. The pipeline goes through each step of the processing with a minimal test example. As detailed previously, we also implemented new features, such as the “custom” mode (allowing the user to give its own list of documented structure-organism pairs) and this one is also tested through the CI.

re: "LOTUS uses Wikidata as a repository for referenced structure-organism pairs, as this allows documented research data to be integrated with a large, pre-existing and extensible body of chemical and biological knowledge. The dynamic nature of Wikidata fosters the continuous curation of deposited data through the user community. Independence from individual and institutional funding represents another major advantage of Wikidata."How is Wikidata independent from individual and institutional funding? Assuming that it takes major funding to keep Wikidata up and running, what other funding source is used by Wikidata beyond individual donations and institutional grants/support?

We thank the reviewer for pointing this out, our sentence was indeed unclear. We adapted it in the manuscript.

What we wanted to express is the precarity of many academic tools, relying on a single funding, or PhD student/ postdoctoral student only. Wikidata benefits from a vast portfolio of fundings, as described in our answer to reviewer 1, thus making it less vulnerable. As long as it will be useful to the community there is no doubt it will be funded.

re: "The openness of Wikidata also o ers unprecedented opportunities for community curation, which will support, if not guarantee, a dynamic and evolving data repository."Can you provide example in which community curation happened? Did any non-LOTUS contributor curate data provided by LOTUS initiative?

Here is an example of cholesterol entry dynamic evolution: https://www.wikidata.org/w/index.php?title=Q43656&action=history.

The KrBot removed some duplicated entries and redirected some other ones. *Clerodendrum fragrans (Q15622830)*, for example, became *Clerodendrum chinense (Q10475331)* and our statement was updated by this non-LOTUS contributor.

"As the taxonomy of living organisms is a complex and constantly evolving eld, all the taxon identi ers from all accepted taxonomic DB for a given taxon name were kept. Initiatives such as the Open Tree of Life (OTL) (Rees and Cranston, 2017) will help to gradually reduce these discrepancies, the Wikidata platform should support such developments."Discrepancies, conflicts, taxonomic revisions, and disagreements are common between taxonomies due to scientific differences, misaligned update schedules, and transcription errors. However, Wikidata does not support organized dissent, instead forcing a single (artificial) taxonomic view onto commonly used names. In my mind, this oversimplifies taxonomic realities, favors one-size-fits-none taxonomies, and leaves behind the specialized taxonomic authorities that are less tech savvy, but highly accurate in their systematics. How do you imagine dealing with conflicting, or outdated, taxonomic interpretations related to published names?

The reviewer is right about the complexity of taxonomy. However, Wikidata does not favor a “one size fits all” approach and in fact actually supports organized dissent. Please see https://www.wikidata.org/wiki/Wikidata:WikiProject_Taxonomy for additional background on the matter.

For example, https://www.wikidata.org/wiki/Q161265 supports two parent taxa, leading to a divergent graph https://w.wiki/453a.

One of the weak points might be in the distinction between a taxon and a taxon name, but this problem is far from being addressed in current natural products literature.

Within our pipeline, outdated taxon names should be linked to their corresponding accepted ones, and therefore we add structures reported in old taxon names to their currently accepted ones. If these become outdated, and a new accepted name is linked to it within Wikidata, the data will continue to follow.

"The currently favored approach to add new data to LOTUS is to edit Wikidata entries directly. Newly edited data will then be imported into the SSOT repository. There are several ways to interact with Wikidata which depend on the technical skills of the user and the volume of data to be imported/modi ed."How do you deal with edit conflicts? E.g., source data is updated around the same time as an expert updates the compound-organism-reference triple through Wikidata. Which edit remains?

Rule: Robot has a memory of what it sent and will never try to send it again, unless manually overridden. As such, the robot trusts humans and other robots. If someone does an edit (suppression, modification) the robot will not try to send it again. In case of vandalism three approaches can be used:

– manual undo (for small cases)

– Wikidata admins undo (for large abuses)

– robot reupload (in case Wikidata admins cannot deal with the vandalism)

So, in case of manual addition by an expert on Wikidata, the expert addition remains. In case of manual deletion by an expert, the expert deletion remains.

We clarified this in the manuscript also (see https://github.com/lotusnprod/lotus-manuscript/commit/6f39237d23980cf3af7c78e4ff7c2a2b4cc64629).

"Even if correct at a given time point, scientic advances can invalidate or update previously uploaded data. Thus, the possibility to continuously edit the data is desirable and guarantees data quality and sustainability. Community-maintained knowledge bases such as Wikidata encourage such a process."On using LOTUS in scientific publications, how can you retrieve the exact version of Wikidata or LOTUS resource referenced in a specific publication? In my understanding, specific Wikidata versions are not easy to access and discarded periodically.

Regarding the versions, in our view, it is the responsibility of the authors of the scientific publication to freeze the version and provide it, with possible modifications, to the reviewers and to the public. We provide the query to export all LOTUS pairs from Wikidata, so the authors of a scientific publication just have to run the query, timestamp it, use it as they want and provide the version they used. This allows full reproducibility while keeping the “living” aspect of Wikidata. See also comments to editors regarding possible versioning mechanisms. E.g. “We will also regularly upload results of a global SPARQL query on the zenodo repository using the https://github.com/lotusnprod/lotus-wikidata-interact or future version of this bot on such datasets https://zenodo.org/record/5668855” or “For example the output of this Wikidata SPARQL query https://w.wiki/4N8G realized on the 2021-11-10T16 :56 can be easily archived and shared in a publication https://zenodo.org/record/5668380.”

On the other hand, on lotus.naturalproducts.net, we implemented a versioning for users. So they can simply refer to the version they downloaded from the website.

We agree with the reviewer, Wikidata versions are difficult to query for specific natural products information. Wikidata Toolkit (https://www.mediawiki.org/wiki/Wikidata_Toolkit) might offer options to query them. However, nothing is discarded.

Some versioning mechanism proposition were added to the manuscript: see https://github.com/lotusnprod/lotus-manuscript/commit/92833166375aa9cc29f44cffdd5ad693fa42934c**.**

Please provide an example of how you imagine LOTUS users citing claims provided by a specific version of LOTUS SSOT to Wikidata.

LOTUS does not directly claim occurrences of chemical structures within biological organisms. The claims are directly supported by the bibliographic reference.

"The LOTUS initiative provides a framework for rigorous review and incorporation of new records and already presents a valuable overview of the distribution of NP occurrences studied to date."How can a non-LOTUS contributor dissent with a claim made by LOTUS, and make sure that their dissent in recorded in Wikidata or other available platforms? Some claims are expected to be disputed and resolved over time. Please provide an example of how you imagine dealing with unresolved disputes.

The easiest way is to contribute directly to Wikidata. Some tutorials are given in the SI to create new statements, also showing the way on how to modify one. Wikidata, thanks to its ever updated content, will always be “one step ahead” of LOTUS. We will periodically re-import Wikidata and we will be able to see:

– new statements corresponding to our criteria

– statements we made that were modified

In the case of modification of statements we previously uploaded, Wikidata always has the priority. The only way to deal with unresolved disputes is manually.

We take this opportunity to link this entry we created, which raised words of caution discussed openly, with details on the reason why: https://www.wikidata.org/wiki/Talk:Q104916955

"Community participation is the most e cient means of achieving more comprehensive documentation of NP occurrences, and the comprehensive editing opportunities provided within LOTUS and through the associated Wikidata distribution platform open new opportunities for collaborative engagement."Do you have any evidence to suggest that community participation is the most effective way to achieve more comprehensive documentation of NP occurrences? Please provide references or data to support this claim.

Our phrasing was not accurate and we modified this section. We thank the reviewer again for careful reading.

"In addition to facilitating the introduction of new data, it also provides a forum for critical review of existing data, as well as harmonization and veri cation of existing NP datasets as they come online."Please provide example of cases in which a review led to a documented dispute that allowed two or more conflicting sources to co-exist. Show example of how to annotate disputed claims and claims with suggested corrections.

As LOTUS data is still “young”, it is difficult to find such a discussion, although we already found an example: https://www.wikidata.org/wiki/Talk:Q104916955. More advanced examples exist, such as for some *Aloe* sp. (https://www.wikidata.org/wiki/Talk:Q145534). One of the most developed discussions concerning taxa is about *Boletus erythropus*. Under https://www.wikidata.org/wiki/Q728666, users can read about how useful those discussions can be. (A query listing all organisms, linked to compounds, with an open discussion page is available at https://petscan.wmflabs.org/?psid=20892472).

How do imagine using Wikidata to document evidence for a refuted "compound-found-in-organism claimed-by-reference" claim?

While it is still difficult to imagine the whole community discussing claims on Wikidata, it offers the right infrastructure to do it. As an example, the place of burial of Albert Einstein has been discussed on: https://www.wikidata.org/wiki/Talk:Q937. This would also be possible for natural products occurrences, with experts discussing the statements made on the compound page. Just like item pages, the talk pages are versioned. See also previously mentioned examples.

Here also the implementation of the Evidence Ontology (http://obofoundry.org/ontology/eco.html) might be a good direction to look at and further characterize and discuss the pertinence of LOTUS data. Such statements could for example complement the “stated in” reference for a natural product occurrence and document more formally the type of evidence (eg. isolated, NMR, crystals etc.)

"Researchers worldwide uniformly acknowledge the limitations caused by the intrinsic unavailability of essential (raw) data (Bisson et al., 2016)."Note that FAIR does not mandate open access of data, only suggests vague guidelines to find, access, integrate and re-use possibly resources that may or may not be access controlled.If you'd like to emphasize open access in addition to FAIR, please separately mention Open Data / Open Access references.

We thank the reviewer for pointing this out, we added Open Data in addition to FAIR.

re: "We believe that the LOTUS initiative has the potential to fuel a virtuous cycle of research habits and, as a result, contribute to a better understanding of Life and its chemistry."LOTUS relies on the (hard) work of underlying data sources to capture the relationships of chemical compounds with organisms along with their citation reference. However, the data source are not credited in the Wikidata interface. How does *not* crediting your data sources contribute to a virtuous cycle of research habits?

We understand the reviewer comment, and this is related to our wish to break with previous research habits cycles. We argue that data sources must shift from the old DB model to the original publication, which is the only source allowing to (in)validate the statement. We do not want to fall in an ever ending circle of a pair originally described in article X but then extracted from Knapsack, then from NPAtlas and COCONUT. The credit goes to the author who did the hard work of experimentally describing the pair. Just like when citing a reference in a scientific paper the idea is to cite the original work and not citing articles. Hopefully, in the end, authors will make it directly available to Wikidata, maybe with the help of publishers.

re: Wikidata activity of NPImporterBotAssociated bot can be found at:
https://www.wikidata.org/wiki/User:NPImporterBot
On 5 Aug 2021, the page https://www.wikidata.org/wiki/Special:Contributions/NPImporterBotmost recent entries included a single edit from 4 June 2021
https://www.wikidata.org/w/index.php?title=Q107038883&oldid=1434969377
followed by many edits from 1 March 2021e.g.,
https://www.wikidata.org/w/index.php?title=Q27139831&oldid=1373345585
with many entries added/upserted in early Dec 2020with earliest entries from 28 Aug 2020
https://www.wikidata.org/w/index.php?title=User:NPImporterBot&oldid=1267066716
It appears that the bot has been inactive since June 2021. Is this expected? When does the bot become active?

This behaviour is expected. The bot becomes active periodically, when we subjectively find there is enough change to run it. We will probably do it on a regular basis or if major (updates of) open databases are (still) released. We waited for the review before running the bot again, so we could easily highlight differences. This has now been done and the difference can be observed between 10.5281/zenodo.5665295 and 10.5281/zenodo.5793668.

re: "The Wikidata importer is available at https://gitlab.com/lotus7/lotus-wikidata-importer. This program takes the processed data resulting from the lotusProcessor subprocess as input and uploads it to Wikidata. It performs a SPARQL query to check which objects already exist. If needed, it creates the missing objects. It then updates the content of each object. Finally, it updates the chemical compound page with a "found in taxon" statement complemented with a "stated in" reference."How does the wikidata upserter take manual edits into account? Also, if an entry is manually deleted, how does the upserter know the statement was explicitly removed to make sure to now re-add a potentially erroneous entry.

This was answered to another question above.

In attempting to review and reproduce (parts of) workflow using provided documentation, https://gitlab.com/lotus7/lotus-processor/-/blob/7484cf6c1505542e493d3c27c33e9beebacfd63a/README.adoc#user-content-pull-the-repository was found to suggest to run:git pull https://gitlab.unige.ch/Adriano.Rutz/opennaturalproductsdb.gitHowever, this command failed with error:fatal: not a git repository (or any parent up to mount point /home)Stopping at filesystem boundary (GIT_DISCOVERY_ACROSS_FILESYSTEM not set).Assuming that the documentation meant to suggest to run:git clone https://gitlab.com/lotus7/lotus-processor,the subsequent command to retrieve all data also failed:$ dvc pullWARNING: No file hash info found for 'data/processed'. It won't be created.WARNING: No file hash info found for 'data/interim'. It won't be created.WARNING: No file hash info found for 'data/external'. It won't be created.WARNING: No file hash info found for 'data/validation'. It won't be created.4 files failedERROR: failed to pull data from the cloud – Checkout failed for following targets:data/processeddata/interimdata/externaldata/validationIs your cache up to date?Without a reliable way to retrieve the data products, I cannot review your work and verify your claims of reproducability and versioned data products.Also, in https://gitlab.com/lotus7/lotus-processor/-/blob/7484cf6c1505542e493d3c27c33e9beebacfd63a/README.adoc#user-content-dataset-list, you claim that the dataset list was captured in xref:docs/dataset.tsv. However, no such file was found. Suspect a broken link with correct version pointing to xref:docs/dataset.csv.

We thank the reviewer for carefully reading, we corrected the broken links. The DVC access was actually an internal one, with no means for non-members to access the data. We removed DVC for the public and made the whole pipeline available with all accessible data. We wrote additional programs to gather all data in a programmatic way: acessible at https://github.com/lotusnprod/lotus-processor/tree/main/src/1_gathering (related commits: https://github.com/lotusnprod/lotus-processor/commit/6cdf56b65b9296eb6fa4f466857dd753c145898c, https://github.com/lotusnprod/lotus-processor/commit/335e83434d20fdacd25505dd0973980756940570, and https://github.com/lotusnprod/lotus-processor/commit/9e93224ca138213bbe24b25f01648fc9be3cd739).

[Editors' note: further revisions were suggested prior to acceptance, as described below.]

The manuscript has been improved but there are some remaining issues (particularly those of Reviewer 2) that need to be addressed, as outlined below:Reviewer #2 (Recommendations for the authors):Rutz et al. describe the LOTUS initiative, a database which contains over 750,000 referenced structure-organism pairs. Present both the data which they have made available in Wikidata, as well as their interactive web portal (https://lotus.naturalproducts.net/). provides a powerful platform for mining literature for published data on structure-organism pairs.The clarity and completeness of the manuscript has improved significantly from the first round to the second round of reviews. Specifically, the authors clarified the scope of the study and provided concrete, and clear examples of how they think the tool should/could be used to advance np research.I think the authors adequately addressed reviewer concerns regarding the documentation of the work and provide ample documentation on how the LOTUS initiative was built, as well as useful documentation on how it can be used by researchers.Strengths:The Lotus initiative is a completely open source, database built on the principles of FAIRness and TRUSTworthiness. Moreover, the authors have laid out their vision of how they hope LOTUS will evolve and grow.The authors make a significant effort to consider the LOTUS end-user. This is evidenced by the clarity of their manuscript and the completeness of their documentation and tutorials for how to use, cite and add to the LOTUS database.Weaknesses/Questions:1) The authors largely addressed my primary concern about the previous version of the manuscript, which was that the scope and the completeness of the LOTUS initiative was not clearly defined. Moreover, by providing concrete examples of how the resource can be used both clarifies the scope of the project and increases the chances that it will be adopted by the great scientific community. In the previous round of reviews, I asked "if LOTUS represents a comprehensive database". In their response, the authors raise the important point that the database will always be limited by the available data in the literature. While I agree with the authors on this point and do not think it takes away from the value of the manuscript/initiative, I think a more nuanced discussion of this point is merited in the paper.While the authors say that the example queries provided illustrate that the LOTUS can be used to answer research questions, i.e. how many compounds are found in Arabipsis. However, interpreting the results of the query are far more nuanced. For example, can one quantitively compare the number of compounds observed in different species, or families based on the database (e.g. do legumes produce more nitrogen containing compounds than other plant families)? Or do the inherent biases in literature inhibit our ability to draw conclusions such as this? I like that the authors showcase the potential value of using the outlined queries as a starting place for a systematic review, as they highlight in the text. However, I would like to see the authors discuss the inherent limitations with using/interpreting the database queries given the incompleteness of NP exploration.

We thank the reviewer for his appreciation of our response to his first comments regarding the coverage of the database.

We also thank him for pointing again to the limitations of the LOTUS Initiative resource completeness at the moment. We added an in-depth discussion of the possible causes of the incompleteness of these (and other existing) NP resources and also propose some perspectives to address them in the future. See the Data Interpretation paragraph:

https://lotusnprod.github.io/lotus-manuscript/#data-interpretation

2) On page (16?) the authors discuss how the exploration of NP is rather limited, citing the fact that each structure is found on average in only 3 organisms, and that only eleven structures per organism. How do we know that these low numbers are a result of incomplete literature or research efforts and not due to the biology of natural products? For example, plant secondary metabolites are extremely diverse and untargeted metabolomics studies have shown that any given compound is found in only a few species. There is likely no way of knowing the answer to this question, but this section could benefit from a more in-depth discussion.

We thank the reviewer for opening the door for this discussion which is in fact closely related to the previous point.

While we agree on the fact that a wide range of the overall metabolism is heavily specialized, thus that some compounds might indeed be found only in a few species, the untargeted metabolomics studies mentioned by the reviewer also highlight how much remains to be discovered (much more than eleven compounds per organism). This is for now confirmed with heavily studied organisms, where the number of compounds reported is sometimes orders of magnitudes higher.

We thus do not necessarily expect the number of organisms per structure to grow much (except for basal metabolism not much will be shared indeed), but rather the number of structures per organism. Again, we would like to highlight the fact that the habit of only reporting few, new, compounds is highly influenced by editorial guidelines of journals publishing natural products research and is one of the causes of this bias.

One of our next research projects actually stems from these observations. We want to explore in more depth the distribution of core and specialized metabolism across the tree of life using LOTUS data as a starting point. By modeling the distribution of chemical classes across the taxonomy and taking into account priors such as the research effort on given taxa or bias inherent to the chemical classes (alkaloids are easy to fetch through acid/base extraction and sought after for the often potent bioactivities) we expect to disentangle part of the tree of life which are lacking research efforts to be better described from those actually not producing the chemicals.

Here again, see our additional paragraph in the Data Interpretation section to answer the reviewer's justified comment:

https://lotusnprod.github.io/lotus-manuscript/#data-interpretation

3) I do not understand Figure 4. I would like to see more details in the figure legend and an explanation of what an "individual" represents? Are these individual publications? Individual structure-organism pairs? Also, what do the box plots represent? Are they connected to the specific query of Arbidosis thaliana/B-sitosterol? If they are there should only be a value returned for the number of structures in *Arabidopsis thaliana*, rather than a distribution. Thus, I assume that the accumulation curve and the box plots are unrelated, and this should be spelt out in the figure legend or separated into two different panels.

We improved the figure caption taking the reviewer’s comments into account.

The “number of individuals” (now renamed “number of entries”) represents either a chemical structure or a biological organism (a species), according to the color. Both box plots and curves take all organisms and structures present in LOTUS into account. *A. thaliana* and KZJWDPNRJALLNS are taken as two notable examples to guide the reader. *A. thaliana* contains 687 different short inchikeys and KZJWDPNRJALLNS is reported in 3981 distinct organisms. These two examples are thus just a specific case of both the box plot and the curves. They were shifted up in the new figure to avoid confusion.

The accumulation curves and boxplots are related. They are two different visualizations illustrating the same data in different ways, as both have their limitations. The accumulation curves show, for example, that around 80 percent of the structures present in LOTUS are covered with less than 10 percent of the organisms. This could not be seen on the boxplot. On the other hand, it is not possible to see that the median of reported structures per organism is 5 on the accumulation curves. We thank the reviewer again for pointing out the lack of clarity in our caption which should now allow the reader an easier understanding of Figure 4.

4) On page 5 the authors introduce/discuss COCONUT. I am confused about how exactly COCONUT is related to or different from LOTUS. Can you provide some more context, similar to how the other databases were discussed in the preceding paragraph?

We thank the reviewer for this interesting question, and we are happy to bring some clarifications both here and in the manuscript. In LOTUS, every NP structure is strictly associated with at least one organism known to produce it, and this association is always documented, which makes LOTUS a highly curated resource. On the opposite, COCONUT contains all known natural product structures, regardless of their documentation status or producer information, and with variable annotation quality. These annotation aspects make both databases complementary, but different, in the same way as SwissProt and TrEMBL are for protein data. Figure 5 offers further information regarding the differences and complementarity in terms of biological occurrences coverage between COCONUT and LOTUS

Additionally, to avoid needless maintenance costs, both websites (https://coconut.naturalproducts.net/, https://lotus.naturalproducts.net/) are built on the same domain, using the same framework, thus increasing the visual similarity.

We rephrased and clarified the paragraph defining the role of COCONUT in the LOTUS Initiative in the Introduction section.